# Giant optical polarisation rotations induced by a single quantum dot spin

E. Mehdi [1,2], M. Gundín [1], C. Millet[1], N. Somaschi[3], A. Lemaître [1], I. Sagnes [1], L. Le Gratiet[1], D. A. Fioretto[1,3], N. Belabas[1], O. Krebs [1], P. Senellart[1] & L. Lanco [1,2,4] ✉

In the framework of optical quantum computing and communications, a major objective consists in building receiving nodes implementing conditional operations on incoming photons, using a single stationary qubit. In particular, the quest for scalable nodes motivated the development of cavity-enhanced spin-photon interfaces with solid-state emitters. An important challenge remains, however, to produce a stable, controllable, spin-dependent photon state, in a deterministic way. Here we use an electrically-contacted pillar-based cavity, embedding a single InGaAs quantum dot, to demonstrate giant polarisation rotations induced on reflected photons by a single electron spin. A complete tomography approach is introduced to extrapolate the output polarisation Stokes vector, conditioned by a specific spin state, in presence of spin and charge fluctuations. We experimentally approach polarisation states conditionally rotated by $\frac{\pi}{2}$, $\pi$, and $-\frac{\pi}{2}$ in the Poincaré sphere with extrapolated fidelities of $(97 \pm 1)$ %, $(84 \pm 7)$ %, and $(90 \pm 8)$ %, respectively. We find that an enhanced light-matter coupling, together with limited cavity birefringence and reduced spectral fluctuations, allow targeting most conditional rotations in the Poincaré sphere, with a control both in longitude and latitude. Such polarisation control may prove crucial to adapt spin-photon interfaces to various configurations and protocols for quantum information.

A major challenge for optical quantum information is the development of deterministic light-matter interfaces, used as stationary nodes communicating through photons[1]. Potentially, loss-resistant quantum communication and quantum computing could be performed with only a few nodes[2], or even a single one[3–5], used to both emit and receive photons.

In the last decade, important efforts have thus been devoted to building efficient receiving nodes, performing conditional operations on incoming photons. This led, for instance, to the demonstration of various quantum gates between incoming photons and stationary qubits, using, e.g., atoms[6–9], solid-state spins[10–14], and superconducting qubits[15]. The developed interfaces could be used in demonstrations of photon-photon gates[16,17], single-photon transistors[8,18,19], quantum memories[20], and quantum non-demolition detectors[21]. In the optical domain, in particular, potential spin-photon interfaces have been explored with a number of solid-state emitters and cavities[22,23], as well as various encodings, including polarisation, path, and time-bin[10–14].

Polarisation encoding, in this respect, has the advantage of providing straightforward 1-qubit gates and measurements, as well as conceptually simple protocols for various spin-photon and multi-photon gates[24–28]. A key objective is to produce a perfect spin-polarisation mapping: starting from a fixed incoming photon state, $|\Psi_{in}\rangle$, and depending upon a spin state $|\uparrow\rangle$ or $|\downarrow\rangle$, an ideal device would deterministically produce states of orthogonal polarisations, namely

[1]Université Paris-Saclay, CNRS, Centre de Nanosciences et de Nanotechnologies, 91120 Palaiseau, France. [2]Université Paris Cité, Centre de Nanosciences et de Nanotechnologies, 91120 Palaiseau, France. [3]Quandela, 7 rue Leonard de Vinci, 91300 Massy, France. [4]Institut Universitaire de France (IUF), 75005 Paris, France. ✉e-mail: loic.lanco@u-paris.fr

$|\Psi_{out}\rangle = |\Psi_\uparrow\rangle$ or $|\Psi_\downarrow\rangle$ with $\langle\Psi_\uparrow|\Psi_\downarrow\rangle = 0$. This in turn would allow producing maximally entangled spin-photon states of the form $(|\uparrow\rangle \otimes |\Psi_\uparrow\rangle + |\downarrow\rangle \otimes |\Psi_\downarrow\rangle)/\sqrt{2}$, through the interaction between an incoming photon and a coherent spin superposition, such as $(|\uparrow\rangle + |\downarrow\rangle)/\sqrt{2}$.

In this respect, most realisations have been pioneered using both high magnetic fields and highly birefringent cavities[7,10,11], i.e., suppressing any spectral overlap between optical transitions and orthogonally polarised cavity modes. In such a case only one transition, and one cavity eigenaxis, can be excited by a given input. For a perfect device, this allows exploiting the $\pi$ phase-shift induced by the excited transition to implement, ideally, a conditional $\pi$ polarisation rotation[29].

Alternatively, a promising strategy is to use cavity-QED devices with moderate birefringence, i.e., exhibiting a spectral overlap between orthogonally polarised cavity modes[30–32]. In such a configuration, perfect spin-polarisation mapping can also be obtained at zero magnetic fields, through opposite, $\pm\frac{\pi}{2}$ rotations in the Poincaré sphere for the states $|\Psi_\uparrow\rangle$ and $|\Psi_\downarrow\rangle$[33]. This ensures compatibility with a variety of protocols based on deterministic quantum gates[24,27] and deterministic entanglement of multiple photons[25,26].

Spin-photon interfaces with moderate birefringence have already been implemented using semiconductor quantum dots (QDs) in pillar-based structures[30,33–36]. Such cavities allow a robust and deterministic light-matter coupling compatible with electrical control[37,38], together with efficient injection[31] and extraction[39] of photons into and from the cavity mode. Yet, until now, conditional spin-induced rotations have remained limited in angle, due to optical losses[33] and/or inhomogeneous broadenings[33–36]. In particular, inhomogeneous broadenings much larger than the homogeneous linewidths lead the output states to fluctuate all around the Poincaré sphere, resulting in potentially strong depolarisations, though post-selection can be used to partially mitigate the impact of fluctuations[35].

Interestingly, in the variety of polarisation-based experiments[7,10,11,30–36] spin-induced polarisation rotations have mostly been measured via intensity contrasts on a single polarisation basis. This is equivalent to a single-axis projection in the Poincaré sphere, which can only give limited information. A typical difficulty is

that a measurement axis can be well-adapted to measure some fidelity, with respect to an ideal target[35,36], yet at the same time prevent from distinguishing between depolarisation effects and actual polarisation rotations. In refs. [33,34], conversely, actual rotations could be demonstrated using a well-adapted measurement basis, yet only reaching rotation angles up to 6°.

Here, we report on giant polarisation rotations induced by a single QD-embedded electron spin, deterministically coupled to an electrically contacted pillar cavity (see Methods). Compared to previous works[33–36], the strong reduction of spectral fluctuations provided by the electrical contacts, and the increased Purcell enhancement, are key ingredients allowing us to reach giant polarisation rotations. This includes highly desired configurations such as $\pm\frac{\pi}{2}$ and $\pi$ conditional rotations in the Poincaré sphere, though degradation of the polarisation purity, down to around 70%, is observed at large angles. We use polarisation tomography[40], this time applied to a charged quantum dot-microcavity device, to fully characterise the state of the reflected photons in the Poincaré sphere. The possibility to add or remove the electron from the quantum dot allows us to extrapolate the conditional Stokes vector $\vec{S}_\uparrow$, conditioned to a charged QD in the spin state $|\uparrow\rangle$, even in the presence of detrimental spin and charge fluctuations. We finally show that, by a proper set of detunings, most orientations of the conditional output Stokes vector $\vec{S}_\uparrow$ can be reached. This provides essential degrees of freedom to adapt future devices to a wide range of protocols and experimental conditions.

## Results
### Principle of the experiments

We sketch in Fig. 1a the charged QD energy levels, with two ground states with opposite electron spin $|\uparrow\rangle$ and $|\downarrow\rangle$, and their corresponding trion states $|\uparrow\downarrow\Uparrow\rangle$ and $|\downarrow\uparrow\Downarrow\rangle$, consisting of a pair of electrons and a single hole[39,41]. The electron might escape the QD, as described by the additional empty state denoted $|\varnothing\rangle$[42]. We apply an external longitudinal magnetic field of magnitude $B$ (Faraday configuration), lifting the energy degeneracy between the two transitions[43]. For a magnetic field around $2\,T$, the two transitions have no energy overlap: we, respectively, label $\omega_{QD}^\uparrow$ and $\omega_{QD}^\downarrow$ the $|\uparrow\rangle - |\uparrow\downarrow\Uparrow\rangle$ and $|\downarrow\rangle - |\downarrow\uparrow\Downarrow\rangle$

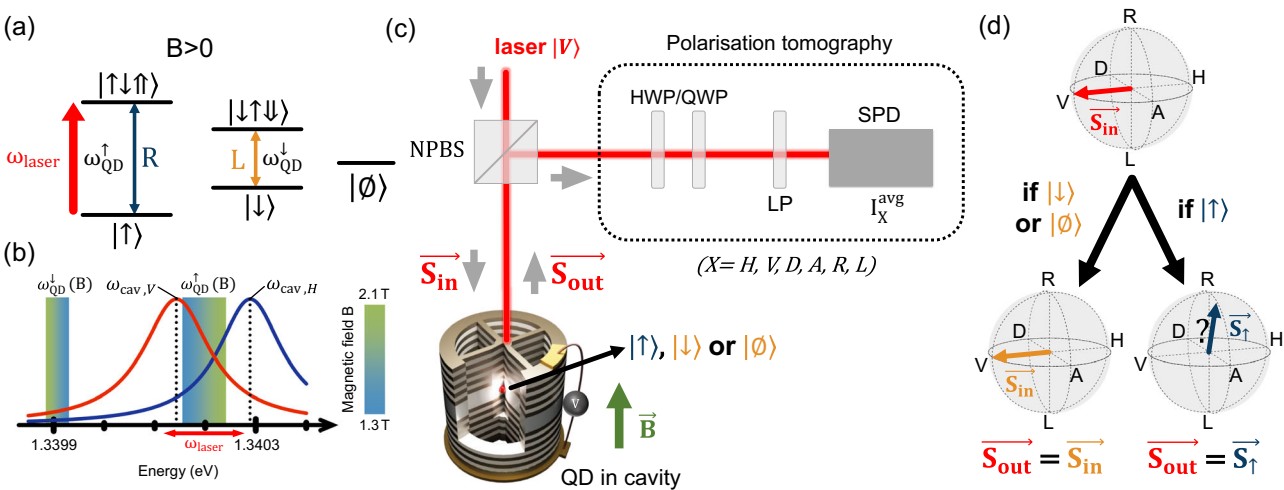

**Fig. 1 | Spin-selective polarisation rotation. a** Energy levels of the negatively charged quantum dot, with an applied magnetic field of magnitude $B$ (Faraday configuration). The system is probed with a laser of energy $\omega_{laser}$ close to $\omega_{QD}^\uparrow$. **b** Scale comparing the two transitions energies, $\omega_{QD}^\uparrow$ and $\omega_{QD}^\downarrow$, to the $V$ and $H$ cavity modes energies, $\omega_{cav,V} = 1.340143$ eV and $\omega_{cav,H} = 1.340289$ eV. The energies $\omega_{QD}^\uparrow$ and $\omega_{QD}^\downarrow$ depend on the value of $B$, as shown by the colorscale, and the two cavity modes spectrally overlap. **c** Experimental scheme. The incoming laser, of Stokes

vector $\vec{S}_{in} \equiv |V\rangle$, is sent on an electrically contacted quantum dot−micropillar device. The reflected polarisation state $\vec{S}_{out}$ is analysed through a polarisation tomography setup (dashed box) measuring the average reflected intensity $I_X^{avg}$ along the six polarisations $X = H, V, D, A, R, L$. NPBS non-polarising beamsplitter, H(Q)WP half (quarter) wave plate, LP linear polariser, SPD single-photon detector. **d** The possible output polarisations of the reflected light in the Poincaré sphere.

transition energies (in $\hbar = 1$ units), with $\omega_{QD}^\uparrow > \omega_{QD}^\downarrow$. A tunable narrow-band laser of energy $\omega_{laser}$ close to $\omega_{QD}^\uparrow$ selectively probes the $|\uparrow\rangle - |\uparrow\downarrow\Uparrow\rangle$ transition.

To provide an efficient spin-photon interface, the QD is determi-nistically coupled to a pillar-based, electrically contacted microcavity[37,38] (see Methods). Furthermore, the transition energy $\omega_{QD}^\uparrow$ is varied, with the applied magnetic field, in the vicinity of the two cavity mode resonances $\omega_{cav,H}$ and $\omega_{cav,V}$, as displayed in Fig. 1b. This ensures that the $|\uparrow\rangle - |\uparrow\downarrow\Uparrow\rangle$ transition, at $\omega_{QD}^\uparrow(B)$, benefits from an efficient Purcell enhancement in both cavity eigenmodes, yet with generally different Purcell factors. These eigenmodes are defined as "horizontally" ($H$) and "vertically" ($V$) polarised.

The principle of the experimental setup is sketched in Fig. 1c (see additional information in the Methods section). The incoming polar-isation, with a Stokes vector denoted $\vec{S}_{in}$, is adjusted to match one of the two cavity eigenaxes, defining state $|V\rangle$. This allows avoiding cavity-induced polarisation rotation[31] while exciting the desired transition, as $|V\rangle = i(|L\rangle - |R\rangle)/\sqrt{2}$, with $R$ and $L$ the right- and left-handed circular polarisations corresponding to the selection rules in Fig. 1a. A non-polarising beamsplitter directs the reflected light, whose polarisation Stokes vector is denoted $\vec{S}_{out}$, to the polarisation tomography setup[31,40] that measures the reflected intensities $I_X$ in various polarisa-tion states $X = H, V, D, A, R, L$ (with $D/A$ the diagonal/antidiagonal polarisations).

As detailed in Supplementary Note 5, the device under study operates in a rapid co-tunnelling regime, where a trapped electron escapes the quantum dot in typically 4 ns, and is directly replaced by another electron from the Fermi sea. Even in such conditions, the radiative transitions can still be considered stable enough to provide a well-defined, state-dependent optical response, since their Purcell-enhanced emission time is around 200 ps. However, when integrating counts for 0.1 s on the single-photon detector, one measures the average intensities $I_X^{avg}$, with contributions from the three possible ground states $|\uparrow\rangle$, $|\downarrow\rangle$ and $|\varnothing\rangle$, with respective probabilities $P_\uparrow$, $P_\downarrow$ and $P_\varnothing$. Notably, the co-tunnelling regime prevents from initialising the spin by optical spin pumping, as each co-tunnelling event implies a loss of spin memory, leading to $P_\uparrow = P_\downarrow = \frac{P_c}{2}$, with $P_c$ the charge occupation probability (see Supplementary Note 4).

When the ground state is $|\uparrow\rangle$, the spin-photon interface can induce large polarisation rotations (Kerr rotation), from $\vec{S}_{in}$ to $\vec{S}_{out} = \vec{S}_\uparrow$. As detailed in Supplementary Note 2, such rotations can be interpreted in the so-called semiclassical approximation, which is valid in the low-power regime (negligibly populated trion states), and when neglecting all sources of fluctuations. In this approximation, one can solve the optical Bloch equations for the cavity operators describing the two eigenmodes ($H$- and $V$-polarised), and for the QD operators describing the two electron-trion transitions ($R$- and $L$-polarised). One shows that $\vec{S}_\uparrow$ corresponds to a pure polarisation $|\Psi_\uparrow\rangle$, with:

$$|\Psi_\uparrow\rangle = \frac{r_{V\to H}^\uparrow |H\rangle + r_{V\to V}^\uparrow |V\rangle}{\sqrt{|r_{V\to H}^\uparrow|^2 + |r_{V\to V}^\uparrow|^2}}, \tag{1}$$

where $r_{V\to H}^\uparrow$ and $r_{V\to V}^\uparrow$ denote complex reflection coefficients (see Supplementary Note 2). They respectively govern the $H$-polarised and $V$-polarised contributions to the reflected output field, in the case where $|\Psi_{in}\rangle = |V\rangle$.

Importantly, both $r_{V\to V}^\uparrow$ and $r_{V\to H}^\uparrow$ depend on two experimentally tunable parameters, $\omega_{laser}$ and $\omega_{QD}^\uparrow$, that can be independently varied. This provides the two required degrees of freedom to control the position of $|\Psi_\uparrow\rangle$ in the Poincaré sphere: the modulus (respectively, the phase) of the ratio $r_{V\to H}^\uparrow / r_{V\to V}^\uparrow$ governs the projection of the Stokes vector on the $HV$ axis (respectively, its relative orientation around the $HV$ axis). As a consequence, a pure state $|\Psi_\uparrow\rangle$ requires stable reflection coefficients, and thus a stable, lifetime-limited transition energy $\omega_{QD}^\uparrow$.

Conversely, inhomogeneous broadenings result in the instability of these coefficients in phase and/or amplitude. This can lead to various degrees of depolarisation (as already observed in ref. 40 with a neutral quantum dot) and potentially severe limitations regarding the aver-aged rotation angle[34].

Finally, if the QD is in state $|\downarrow\rangle$ or $|\varnothing\rangle$, expressions similar to Eq. (1) can also be written, yet with $r_{V\to H}^\downarrow = r_{V\to H}^\varnothing = 0$, the laser being far-detuned from any available transition. This leads to an unchanged Stokes vector ($\vec{S}_{out} = \vec{S}_{in}$), since $|\Psi_\downarrow\rangle = |\Psi_\varnothing\rangle = |V\rangle$.

**Spin-induced polarisation rotation**

We first display in Fig. 2a the reflected light intensities $I_{H/V}^{avg}$, normalised by the input laser intensity, and plotted as a function of the energy detuning $\omega_{laser} - \omega_{cav,V}$, for three different magnetic fields. The applied magnetic field controls the splitting between the two transitions of energy $\omega_{QD}^\uparrow$ and $\omega_{QD}^\downarrow$, bringing $\omega_{QD}^\uparrow$ in and out of resonance with the cavity mode $V$ (see Fig. 1b). The transition at energy $\omega_{QD}^\downarrow$ remains detuned from both cavity modes, and outside the spectral range probed in Fig. 2a.

The incoming intensity being V-polarised, $I_H^{avg}$ corresponds only to the QD resonance fluorescence (RF) emission, cross-polarised to the incoming laser polarisation. By contrast, the reflected intensity $I_V^{avg}$ results from the interference between the empty-cavity reflectivity, contributing to a Lorentzian-shaped reflectivity dip centred at $\omega_{cav,V}$, and the co-polarised part of the QD RF emission, contributing to a deviation from the Lorentzian shape in the vicinity of $\omega_{QD}^\uparrow$.

Focusing on a smaller energy range around $\omega_{QD}^\uparrow$, at 1.7 T, the top panel of Fig. 2b shows the dependence of the reflected intensities, and their sum $I_V^{avg} + I_H^{avg}$, on the energy detuning $\omega_{laser} - \omega_{QD}^\uparrow$. In the lower panels, the Stokes parameters $s_{XX}^{avg}$ of the reflected light, retrieved via full tomography (see Methods), are displayed. Far from the resonance with the $\omega_{QD}^\uparrow$ transition, the reflected polarisation is identical to the input one: $s_{HV}^{avg} = -1$, $s_{DA}^{avg} = s_{RL}^{avg} = 0$, $\vec{S}_{out} = \vec{S}_{in} \equiv |V\rangle$. Conversely, the laser in resonance with $\omega_{QD}^\uparrow$ results in a maximum for $s_{HV}^{avg}$.

The experimental data for both the intensities and Stokes vectors are in good agreement with our theoretical modelling. Numerical simulations have been used, where the main effects result from the polarisation rotation governed by the reflectivity coefficients $r_{V\to H}$ and $r_{V\to V}$, additionally taking into account slow fluctuations of the excited transition energy, and the averaging over the ground states, $|\uparrow\rangle$, $|\downarrow\rangle$ and $|\varnothing\rangle$. The discussion of the extracted parameters is summarised below, details being given in the Supplementary Notes 2 and 3.

First, all cavity parameters have been unambiguously extracted by fitting experimental data obtained in the empty-cavity regime (see extended data in Supplementary Note 7). The cavity is described by its two Lorentzian modes with energies $\omega_{cav,V}$ and $\omega_{cav,H}$, and damping rates $\kappa_V = (162 \pm 6) \, \mu eV$ and $\kappa_H = (155 \pm 6) \, \mu eV$, deduced from the central energies and widths of the reflectivity spectra. The splitting between the two cavity modes is $\Delta = \omega_{cav,H} - \omega_{cav,V} = (146 \pm 1) \, \mu eV$, of the order of $\kappa_H$ and $\kappa_V$. This characterises a moderate birefringence, i.e., a partial overlap between the modes, as schematized in Fig. 1b. The probability that an intracavity photon escapes from the top mirror of the micro-pillar is given by the effective top mirror output coupling $\eta_{top,H/V} = (0.635 \pm 0.01)$ for both modes.

Then, most of the quantum dot parameters are unambiguously extracted, by fitting the average intensity measurements $I_{H/V}^{avg}$ in var-ious conditions. This includes the parameters governing the magnetic field response, and thus $\omega_{QD}^\uparrow$, deduced from the resonance frequencies measured at various magnetic fields. This also includes the QD-cavity coupling constant $g = (15.0 \pm 1) \, \mu eV$ and the spontaneous decay rate $\gamma_{sp} = (0.35 \pm 0.05) \, \mu eV$, describing the emission in all spatial modes other than the two fundamental cavity modes $H$ and $V$. These two parameters are quite precisely estimated thanks to the fact that they directly influence the amplitude, shape, and width of the QD-induced optical response.

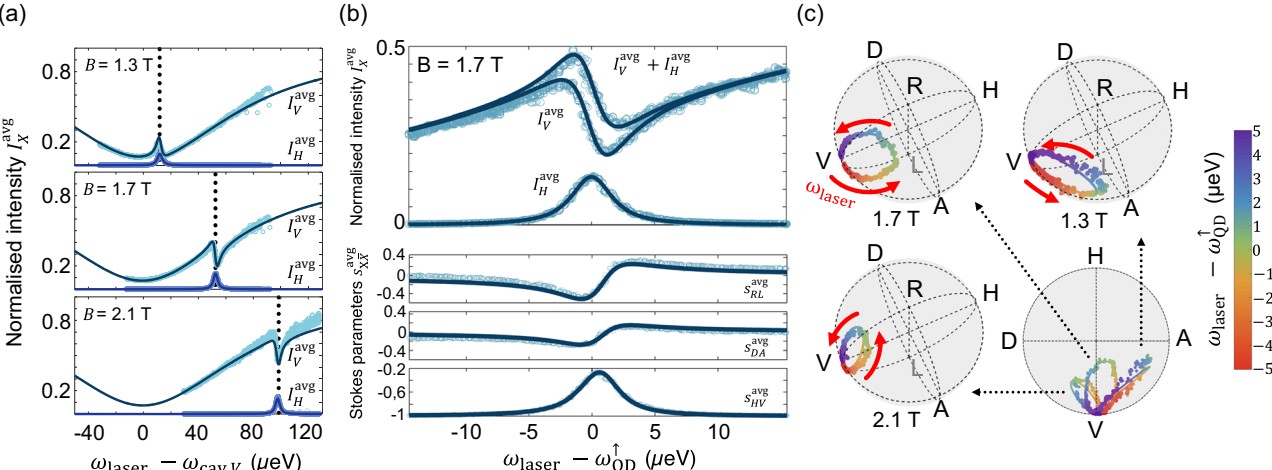

**Fig. 2 | Averaged intensities and polarisations for a non-initialised QD state.** (circles: experimental data; lines: numerical simulations). **a** Normalised average intensity $I_X^{avg}$ (with X=H, V) as a function of the energy detuning between the laser and the V cavity mode for three different magnetic fields 1.3 T, 1.7 T and 2.1 T. The measured signal results from the averaging over QD states ($|\uparrow\rangle$, $|\downarrow\rangle$ and $|\varnothing\rangle$), inherent to the measurement timescale. In each panel, the vertical dashed line highlights the resonance condition for which $\omega_{laser}$ equals $\omega_{QD}^\uparrow$, the energy of the $|\uparrow\rangle$-$|\uparrow\downarrow\Uparrow\rangle$ transition, which is shifted by the applied magnetic field (see Fig. 1b). **b** Normalised average intensity $I_X^{avg}$ (with X=H, V) and Stokes parameters $s_{X\bar{X}}^{avg}$ (with $X\bar{X}$=HV, DA, RL) as a function of the energy detuning between the laser and the $|\uparrow\rangle - |\uparrow\downarrow\Uparrow\rangle$ transition at 1.7 T. **c** Rotation of the average output polarisation in the Poincaré sphere as a function of the detuning (see colorscale) between the laser and the $|\uparrow\rangle - |\uparrow\downarrow\Uparrow\rangle$ transition, for the three different magnetic fields. The bottom-right panel displays the output polarisation in a top view of the Poincaré sphere for the three magnetic fields (see also Supplementary Movies 1–3 for complete 3D visualisations). The red arrows describe increasing laser-QD detunings.

Finally, the two remaining parameters are related to the expected sources of noise. First, spectral fluctuations are considered, and described by a Gaussian distribution of $\omega_{QD}^\uparrow$, with standard deviation $\sigma_{SF}$. In addition, a non-unity value of the charge occupation probability $P_c$ is also considered. As discussed in Supplementary Notes 3 and 5, the best fit is obtained for $P_c = 0.94 \pm 0.03$, together with $\sigma_{SF} = (0.5 \pm 0.2)\,\mu eV$, where the uncertainties correspond to the standard error. This set of parameters is the one that allows fitting all the measured data in Fig. 2, but also additional data, in the form of two-photon coincidences measured as a function of delay. Such measurements (see Supplementary Note 5) show that, for the optimal applied voltage of −0.63 V, a good fit of all available data is obtained with respective escape and capture times of 4 ns and 250 ps, indeed corresponding to $P_c = 0.94$.

In general, spectral fluctuations can have various causes, including electrostatically induced fluctuations[44]. Yet, the standard deviation $\sigma_{SF} = 0.5\,\mu eV$ can be almost entirely explained by hyperfine interaction. If that were strictly the case, one would have $\sigma_{HI}^{(e)} = 2\sigma_{SF} = 1\,\mu eV$, with $\sigma_{HI}^{(e)}$ the standard deviation of the Zeeman splitting, mainly induced by the electron-nuclei interaction. Such a value of $\sigma_{HI}^{(e)}$ is in agreement with the ones observed in similar quantum dots[45], and only slightly larger than the value $\sigma_{HI}^{(e)} = 0.8\,\mu eV$ used to fit experiments in a different device from our group[46].

The polarisation rotation induced by the device is shown in Fig. 2c, in which the average Stokes vectors $\vec{S}_{out}$ are plotted as a function of the detuning between the laser and $\omega_{QD}^\uparrow$, for the different magnetic fields. The output polarisation $\vec{S}_{out}$ is deduced from the six average reflected intensities $I_X^{avg}$. Each point is colour-coded to a specific detuning $\omega_{laser} - \omega_{QD}^\uparrow$. The top and bottom-left panels show the polarisation rotation for the three magnetic fields 1.3 T, 1.7 T, and 2.1 T, while the bottom-right panel aggregates the three cases as viewed from a different angle. On resonance with the $|\uparrow\rangle - |\uparrow\downarrow\Uparrow\rangle$ transition (green data points), the output polarisation is the farthest from the $|V\rangle$ input polarisation. Far from resonance, $\vec{S}_{out}$ remains $|V\rangle$. The trajectory of $\vec{S}_{out}$, i.e., the ensemble of points in the Poincaré sphere as the laser wavelength is scanned, depends directly on the detuning between the $|\uparrow\rangle - |\uparrow\downarrow\Uparrow\rangle$ transition and the cavity modes H and V. Such a detuning is controlled by the applied magnetic field as shown in

Fig. 2a. The numerical simulations in solid lines matches the measured trajectory of the average output polarisation $\vec{S}_{out}$, for all three magnetic fields, with the above-described set of parameters.

## Conditional Stokes vector extrapolation

We now deduce the behaviour of the output polarisation $\vec{S}_\uparrow$, conditioned to a spin being in state $|\uparrow\rangle$, even though the electron spin is not experimentally initialised. It can indeed be extrapolated from the measured average intensities and complementary measurements with the empty-cavity. When the QD is in the state $|\uparrow\rangle$ with a probability $P_\uparrow$, the reflected light is described by the set of intensities $I_X^\uparrow$, corresponding to the Stokes vector $\vec{S}_\uparrow$. Whereas, when the QD is in the state $|\downarrow\rangle$ or $|\varnothing\rangle$, with a probability $(1 - P_\uparrow)$, it is transparent for the laser at the studied energy range (see Fig. 1b and d). The corresponding reflected light polarisation is then described by the empty-cavity intensities $I_X^{cav}$, that can be experimentally measured by forcing the absence of an electron in the QD (applied voltage of 0 V). For each polarisation X, $I_X^{avg}$ is thus the weighted sum of the conditional intensities in the two previous cases:

$$I_X^{avg} = (1 - P_\uparrow)I_X^{cav} + P_\uparrow I_X^\uparrow. \tag{2}$$

Figure 3a illustrates the extrapolation process at 1.7 T. The measured normalised intensities $I_X^{avg}$ and $I_X^{cav}$ (for X = H and V), are plotted as a function of the energy detuning between the laser and the cavity mode V in the first two panels. The third panel shows the extrapolated intensities $I_X^\uparrow$, as deduced from the measured intensities $I_X^{avg}$ and $I_X^{cav}$ with Eq. (2), using $P_\uparrow = \frac{P_c}{2} = 0.47 \pm 0.015$. The same extrapolation process is performed for X = D, A, R and L.

The extrapolated behaviour of $\vec{S}_\uparrow$ is plotted in the Poincaré sphere as a function of the detuning $\omega_{laser} - \omega_{QD}^\uparrow$, as shown in Fig. 3b where the left and right spheres represent two views of the same trajectory at B=1.7 T. $\vec{S}_\uparrow$ is experiencing a giant rotation as shown in the first view. The second view confirms that $\vec{S}_\uparrow$ starts from the polarisation $|V\rangle$, in the off-resonance case, and rotates close to $|H\rangle$ on resonance with the $|\uparrow\rangle$ - $|\uparrow\downarrow\Uparrow\rangle$ transition. Here, the deterministic giant polarisation rotation almost fully reverses the state of the reflected photon, conditioned by the $|\uparrow\rangle$ state of the QD spin.

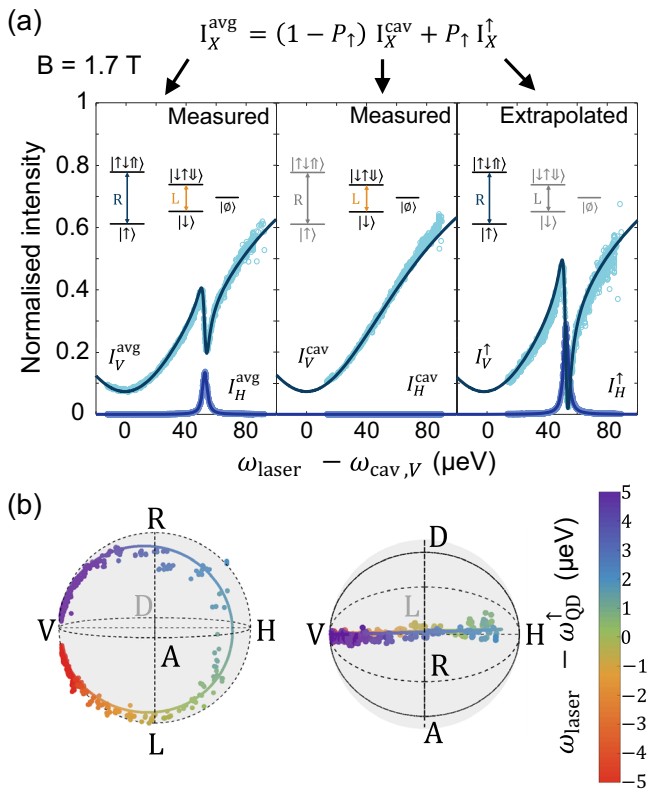

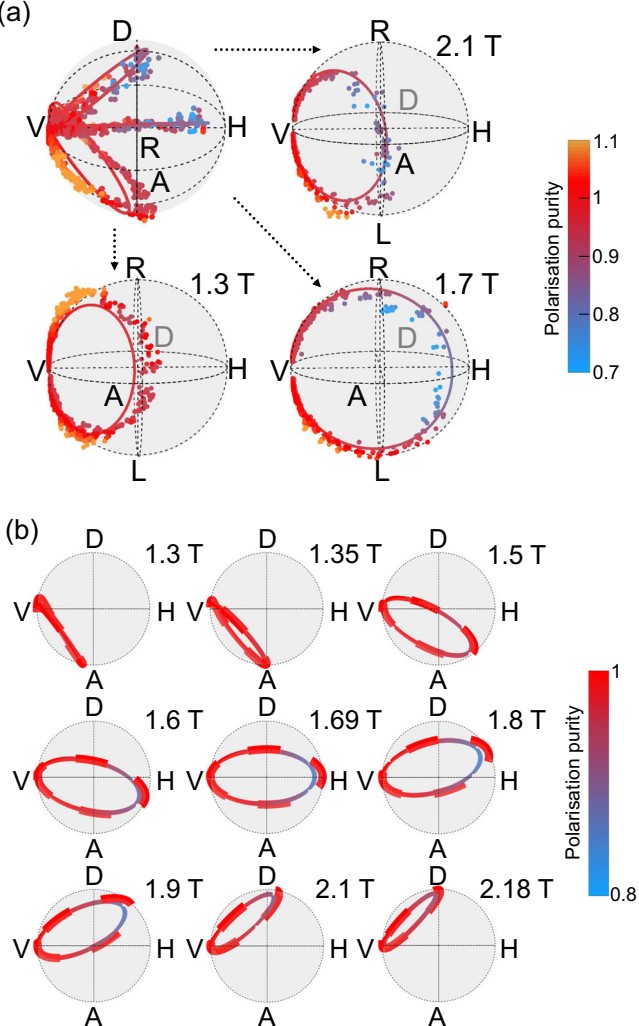

**Fig. 3 | Extrapolating the Stokes vector for a pure QD spin state. a** Normalised intensity $I_X^{avg}$ (with X=H, V) as a function of the detuning between the laser and the cavity V-mode at 1.7T (circles: measured or extrapolated data; lines: numerical simulations). First panel: experimentally measured intensity $I_X^{avg}$ averaged over the non-initialised QD state ($|\uparrow\rangle$, $|\downarrow\rangle$ and $|\varnothing\rangle$), as in Fig. 2a middle panel. Second panel: experimentally measured intensity $I_X^{cav}$ for the empty-cavity, providing information on the system response when it is not in state $|\uparrow\rangle$. Third panel: extrapolated intensity $I_X^{\uparrow}$ describing the system response when the QD is in state $|\uparrow\rangle$.
**b** Extrapolated Stokes vector $\vec{S}_{\uparrow}$ as a function of $\omega_{laser} - \omega_{QD}^{\uparrow}$ (see colorscale), viewed from two different angles in the Poincaré sphere (circles: extrapolated from experimental data; lines: numerical simulations). See also Supplementary Movie 4 for a complete 3D visualisation.

The output polarisation purity, described by the norm of the Stokes vector $\vec{S}_{\uparrow}$, is slightly deteriorated on resonance with the QD, as the corresponding points are not at the surface of the sphere. This effect is due to the environmental noise, in particular hyperfine interaction, leading to fluctuations of $\omega_{QD}^{\uparrow}$. The behaviour of $\vec{S}_{\uparrow}$ given by the extrapolation is well predicted by our numerical simulation, with the exact same parameters as those used in Fig. 2. In Fig. 3b, we note that some points display an unphysical purity above unity. This would not arise with other estimation techniques such as maximum likelihood[47], yet we chose here the most direct method, displaying $\vec{S}_{\uparrow}$ from the values of $s_{HV}$, $s_{DA}$ and $s_{RL}$, to visualise the result of the extrapolation technique. The comparison between the extrapolated data and the numerical simulation confirms the viability of the polarisation tomography in spite of the experimental errors, which include detection noise and nonlinearity, as well as polarisation basis miscalibration. In the results of Fig. 3b, in particular, such errors are amplified by the extrapolation process.

Thus far, the extrapolation at 1.7 T demonstrated the ability to produce different outputs in the Poincaré sphere by controlling $\omega_{laser}$. In addition, the trajectory of $\vec{S}_{\uparrow}$ in the Poincaré sphere is also controlled by the value of $\omega_{QD}^{\uparrow}$, experimentally tunable with the applied magnetic field. The results of the extrapolation processes are

**Fig. 4 | Generating different output states in the Poincaré sphere. a** Extrapolated Stokes vector $\vec{S}_{\uparrow}$, for different magnetic fields (circles: extrapolated from experimental data; lines: numerical simulations). In this figure, the trajectories correspond to scan of detunings $\omega_{laser} - \omega_{QD}^{\uparrow}$ similarly to Fig. 2c, but the colorscale indicates the polarisation purity. See also Supplementary Movies 5–7 for complete 3D visualisations. **b** Numerical simulation of the reflected Stokes vector $\vec{S}_{\uparrow}$ for different magnetic fields without and with environmental noise (resp. dashed lines and solid lines). These simulations are viewed from the top of the Poincaré sphere, i.e., from a different perspective than in previous figures. See also Supplementary Movies 8–9 for complete 3D visualisations.

displayed in the Poincaré sphere, in Fig. 4a, for the three magnetic fields previously explored (see also the corresponding Stokes components in Supplementary Note 7). The top-left panel aggregates the trajectories of $\vec{S}_{\uparrow}$ for all magnetic fields. Each other panel isolates $\vec{S}_{\uparrow}$ for a single magnetic field, under a second viewing angle. In these panels, though the points correspond to different laser-QD detunings, each point is colour-mapped with respect to its polarisation purity. Here again, the non-physical points with a purity above unity are explained by the experimental uncertainties on $I_X^{avg}$ and $I_X^{cav}$. This is especially visible at 1.3 T, where $\omega_{QD}^{\uparrow}$ is close to $\omega_{cav,V}$, leading to low reflectivities, and a higher sensitivity to detector noise and residual cavity-induced polarisation rotations. This discrepancy between theory and experiments is amplified by the extrapolation process.

## Discussion

These experimental results indicate the possibility of generating most of the polarisation states in the Poincaré sphere through a proper

setting of the experimental parameters. This is possible because the rotated polarisation state, given by Eq. (1) in the absence of fluctuations, can be controlled by varying both coefficients $r^{\uparrow}_{V \to V}$ and $r^{\uparrow}_{V \to H}$, through the experimentally controlled parameters $\omega_{laser}$ and $\omega^{\uparrow}_{QD}$. By using magnetic fields of 1.3 T, 1.7 T and 2.1 T, we respectively targeted to reach, at specific QD-laser detunings, an ideal polarisation state $|A\rangle$ (i.e., $r^{\uparrow}_{V \to V} = -r^{\uparrow}_{V \to H}$), $|H\rangle$ (i.e., $r^{\uparrow}_{V \to V} = 0$) and $|D\rangle$ (i.e., $r^{\uparrow}_{V \to V} = r^{\uparrow}_{V \to H}$). These targets were experimentally approached with fidelities of $(97 \pm 1)\%$, $(84 \pm 7)\%$, and $(90 \pm 8)\%$, respectively (see Methods).

To illustrate the diversity of possible output states, we also show in Fig. 4b simulations for the trajectory of $\vec{S}_{\uparrow}$, with spectral fluctuations ($\sigma_{SF} = 0.5\,\mu eV$, solid lines) and without (dashed lines), for other magnetic fields. For each trajectory, the simulated points are obtained for different QD-laser detunings, yet here also the colorscale represents polarisation purity. In the absence of spectral fluctuations, the polarisation of the reflected photons can reach most states at the surface of the Poincaré sphere, through a wide range of combinations in latitude and longitude (see Supplementary Note 7).

Conversely, in the case where spectral fluctuations are introduced, the expected polarisation is slightly degraded. This is directly related to the averaging of the Stokes vector over various orientations, induced by the fluctuations of $\omega^{\uparrow}_{QD}$. As an example, when $\omega^{\uparrow}_{QD} \approx \omega_{cav,V} + 51.4\,\mu eV$ (obtained here at 1.69 T), the QD-laser detuning can be set such that the Stokes vector points towards $|H\rangle$, with $s^{\uparrow}_{HV} = 0.81$ and $s^{\uparrow}_{DA} = s^{\uparrow}_{RL} = 0$. In such a case the Stokes vector has experienced a $\pi$ rotation in longitude while keeping its latitude at 0, the limitation being entirely due to the polarisation purity degraded down to 0.81 (see Methods).

Interestingly, a good theoretical purity of 98% is obtained when $\omega^{\uparrow}_{QD} = \omega_{cav,V} + 14.1\,\mu eV$ (obtained here at 1.35T), which translates into a fidelity of 99% with respect to the desired state $|A\rangle$. This is especially promising for reaching perfect spin-polarisation mapping, i.e., $\langle\Psi_{\uparrow}|\Psi_{\downarrow}\rangle = 0$, at zero magnetic field. Indeed, the polarisation-converting coefficients carry a spin-dependant sign, while the co-polarising coefficients do not (see Supplementary Note 2). Thus, when $\omega^{\uparrow}_{QD} = \omega^{\downarrow}_{QD}$ (at zero magnetic field), $r^{\uparrow}_{V \to H} = -r^{\downarrow}_{V \to H}$, while $r^{\uparrow}_{V \to V} = r^{\downarrow}_{V \to V}$. With our device, it was impossible to tune the QD-transition energy at zero field so that $\omega^{\uparrow}_{QD} = \omega^{\downarrow}_{QD} = \omega_{cav,V} + 14.1\,\mu eV$. Had it been the case, the spin states $|\uparrow\rangle$ and $|\downarrow\rangle$ would almost have been mapped to the opposite polarisation states $|\Psi_{\uparrow}\rangle = |A\rangle$ and $|\Psi_{\downarrow}\rangle = |D\rangle$. In a highly-birefringent device, such a feature would be out of reach in any case due to $r^{\uparrow/\downarrow}_{V \to H} = r^{\uparrow/\downarrow}_{H \to V} = 0$ (see Supplementary Notes 2 and 6).

Though being imperfect, the cavity-QED device presented here actually meets the four crucial conditions to be able to generate most polarisation states in the Poincaré sphere: (i) a large-enough output-coupling efficiency $\eta_{top}$, to allow the QD resonance fluorescence to strongly interfere with the directly reflected light[33] (ii) a drastic reduction of spectral fluctuations, to limit the QD inhomogeneous broadening and thus preserve the purity of the output polarisation states (iii) a cooperativity large-enough to broaden the QD-transition homogeneous linewidth significantly above the residual spectral fluctuations, and (iv) a moderate birefringence, allowing Purcell-enhanced emission in both polarisations. The condition of moderate birefringence, in particular, is essential to allow converting light polarisation, i.e., reach significant values for $r_{V \to H}$ or $r_{H \to V}$. Still, the other three conditions explain why giant and stable rotations had never been observed in low-birefringence devices, either due to output-coupling efficiencies below 50%[33] and/or inhomogeneous broadenings[33–36]. As an example, for the two devices respectively used in refs. 34,35 and in ref. 36, the inhomogeneous full-widths at half-maximum are respectively around 4.5 $\mu eV$ and 10 $\mu eV$, to be compared with respective homogeneous linewidths of 0.8 $\mu eV$

and 1.4 $\mu eV$. In our work, the inhomogeneous broadening remains limited since $\sigma_{SF} = 0.5\,\mu eV$, thanks to the efficient evacuation of fluctuating carriers through the electrical contacts (see Methods and refs. 37,38 for details on the device structure). In addition, the homogeneous linewidth is increased to 3.5 $\mu eV$, thanks to an improved quality factor due to a higher number of Bragg mirror pairs, yet at the cost of a sub-optimal output-coupling efficiency $\eta_{top} = 0.635$ (as compared to $\eta_{top} > 0.9$ in refs. 34,35 and [36]). We also note that in the previous references[33–36], as in the present work, the spin has not been initialised.

To mitigate the effects of slow spectral fluctuations and non-initialised spin, a post-selection approach has been introduced in ref. 35, yet such a technique cannot compensate for fast hyperfine-induced fluctuations[44]. A more scalable approach will be to combine electrical control, as performed here, with techniques recently developed that allow drastic reductions of hyperfine-induced fluctuations[48]. To allow implementing spin initialisation, control and superposition, it will also be crucial to prevent co-tunnelling, e.g., by widening the tunnel barrier between the QD layer and the n-doped Fermi sea. GaAs/AlGaAs quantum dots could also be used as the embedded stationary qubit, allowing it to reach spin coherence times above 100 $\mu s$ through dynamical decoupling[49]. Such improvements will allow taking advantage of spin-induced polarisation rotation with the spin prepared in highly coherent superposition states, in order, for instance, to deterministically generate entanglement between a spin and multiple photons[25]. Higher-purity polarisation states will also be obtained through increased Purcell enhancements, for both polarisations, to maximise the robustness of the coefficients $r_{V \to V}$ and $r_{H \to V}$, with respect to remaining fluctuations.

To develop truly optimal devices, one also needs to increase the efficiency of the spin-photon interface, i.e., the probability to successfully reflect the incoming photons, which is equivalent to the total normalised intensity $I_H + I_V$. The latter is limited, in particular, due to the imperfect $\eta_{top} = 0.635$. As seen from the values of $I_H$ and $I_V$ in Fig. 2a, the present device efficiency is around 35% at 1.7 T, and around 60% at 2.1 T, yet dropping down to around 15% at 1.3 T. This dependence is due to the varying detunings $\omega^{\uparrow}_{QD} - \omega_{cav,H}$ and $\omega^{\uparrow}_{QD} - \omega_{cav,V}$, which govern $|r_{V \to H}|^2$ and $|r_{V \to V}|^2$, and thus $I_H$ and $I_V$. We note that improved values of $\eta_{top} = 0.9$ have already been obtained in similar structures[39]: this paves the way towards enhanced efficiencies, and potentially towards nearly deterministic entanglement of incoming photons with a single spin.

Finally, we note that this control of the polarisation states, in moderately birefringent devices, provides a general approach to implement perfect spin-polarisation mapping at any magnetic field. Starting from a trivial input state such as $|\Psi_{in}\rangle = |V\rangle$, one can indeed maximise the distance between $|\Psi_{\uparrow}\rangle$ and $|\Psi_{\downarrow}\rangle$, and eventually reach the desired condition $\langle\Psi_{\uparrow}|\Psi_{\downarrow}\rangle = 0$, equivalent to $(r^{\uparrow}_{V \to H})^* r^{\downarrow}_{V \to H} + (r^{\uparrow}_{V \to V})^* r^{\downarrow}_{V \to V} = 0$. This strongly encourages pursuing the efforts towards new cavity-QED devices with low birefrigence, also including bullseye[50] and open Fabry-Perot cavities[14], or carefully engineered photonic crystals[51]. With high magnetic fields, using highly birefringent cavities[7,10,12,52] may also allow the engineering of perfect spin-polarisation mapping, yet this requires using non-trivial input states to compensate for the negligible coefficients $r_{V \to H}$ and $r_{H \to V}$ (see Supplementary Note 6). This renders it all the more necessary to include the tomography approach in future experiments with polarisation interfaces, to reach high-fidelity operations even with imperfect devices.

## Methods
### Device fabrication
Full details on the device fabrication can be found in refs. 37,38, though a different QD-pillar device is used in the present work. In

particular, the sample is grown by molecular beam epitaxy and consists of a $\lambda$-GaAs cavity, formed by two distributed Bragg reflectors, embedding an annealed InGaAs QD. The Bragg mirrors are made by alternating layers of GaAs and $Al_{0.9}Ga_{0.1}As$, with 20 (30) pairs for the top (bottom) mirror. To electrically contact the structure, the bottom mirror (Si-doped) presents a gradual doping from $2 \times 10^{18}$ cm$^{-3}$ down to $1 \times 10^{18}$ cm$^{-3}$. This level of doping is maintained in the first half of the cavity region and is stopped only 25 nm before the QD layer, which creates a tunnel barrier between the quantum dot and the Fermi sea. The top mirror is C-doped with increasing doping level, from zero to $2 \times 10^{19}$ cm$^{-3}$ at the surface.

The deterministic spatial and spectral matching between the micropillar cavity and a single QD is achieved by in-situ lithography[37,38]. Each device is formed of a central micropillar (around 3 $\mu$m diameter) connected through four ridges to a large circular frame, attached to a gold-plated mesa enabling the electrical control. Applying a bias of $-0.63$ V stabilises best the electrical environment of the QD while ensuring a maximal probability for a single electron to occupy it (see Supplementary Note 5). The micropillar presents a small ellipticity leading to the lift of degeneracy of the cavity modes, which are split into two spectrally overlapping modes corresponding to $H$ and $V$ polarisations. The corresponding quality factor slightly differs for the two polarisations, with $Q_H = 8650$, and $Q_V = 8300$, respectively (see Supplementary Note 7).

### Resonant excitation experiments

A tunable continuous wave laser, in the linear low-power regime ($P_{in} = 4$ pW), is injected in the QD-micropillar device, placed in a liquid Helium cryostat at 4 K. Light is focused and collected using a cold aspheric lens (4 mm focal length) within the cryostat. The diameter of the free space beam incoming on the aspheric lens is adjusted so that the numerical aperture of the focused beam matches that of the pillar device ($N.A. \approx 0.2$). Using an optimisation and characterisation method detailed in ref. 31, near-perfect mode matching can be approached, leading to $\approx 95\%$ coupling of incoming photons into the cavity.

The application of a longitudinal magnetic field is used to vary the $|\uparrow\rangle - |\uparrow\downarrow\uparrow\rangle$ transition between 925.05 nm and 925.20 nm, while the two cavity mode resonances are at 925.156 nm (V-mode) and 925.056 nm ($H$ mode).

### Polarisation characterisation

In addition to the optical elements sketched in Fig. 1c, additional motorised waveplates are used to compensate for the polarisation rotations induced by the optical setup and the cryostat window, both for the incoming and reflected beams. The non-polarising beamsplitter has a reflectivity of around 90% for both polarisations, ensuring that the intensity is divided independently of the polarisation. This does not exclude the possibility of a phase difference between polarisations, inducing a unitary polarisation rotation that is included in the optical compensation.

The polarisation tomography setup successively measures the average intensities $I_X^{avg}$ in the six different polarisation bases $X = H, V, D, A, R, L$, where all polarisations are defined at the entrance of the microcavity. The intensity measurements are performed through single-photon detection, which is also useful for extracting additional data such as photon coincidences and second-order intensity correlations, as in Supplementary Note 5. The Stokes parameters $s_{X\bar{X}}$ are deduced as $s_{X\bar{X}}^{avg} = (I_X^{avg} - I_{\bar{X}}^{avg})/(I_X^{avg} + I_{\bar{X}}^{avg})$, with $X\bar{X} = HV, DA, RL$. A given polarisation in the Poincaré sphere is characterised by its Stokes vector with components ($s_{HV}$, $s_{DA}$, $s_{RL}$), and its polarisation purity by $\rho = \sqrt{s_{HV}^2 + s_{DA}^2 + s_{RL}^2}$. Its latitude $\theta$, defined as the angle between the

Stokes vector and the equatorial ($HDVA$) plane, is then geometrically expressed as $\theta = \arcsin(s_{RL}/\sqrt{s_{HV}^2 + s_{DA}^2 + s_{RL}^2})$. Its longitude $\phi$, defined as the angle between the $|H\rangle$ state and the projection of the Stokes vector in the equatorial plane, is expressed as $\phi = \text{sgn}(s_{DA}) \arccos(s_{HV}/\sqrt{s_{HV}^2 + s_{DA}^2})$. The fidelity of a Stokes vector $\vec{S}$ to an ideal target $\vec{S}_{target}$ is computed through $F = \frac{1}{2}(1 + \vec{S} \cdot \vec{S}_{target})$. For each fidelity the displayed standard error is computed through uncertainty propagation, considering a $\pm 3\%$ relative standard error on each measured intensity, as well as the estimated $\pm 0.015$ standard error on $P_\uparrow$.

## Data availability

The authors declare that the data supporting the findings of this study are available within the paper, its supplementary information files, and the Figshare repository (https://doi.org/10.6084/m9.figshare.24721761).

## Code availability

Codes used to simulate polarisation rotations are available from the corresponding author upon request.

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

## Acknowledgements

This work was partially supported by the Paris Ile-de-France Région in the framework of DIM SIRTEQ, the European Union's Horizon 2020 Research and Innovation Programme QUDOT-TECH under the Marie Sklodowska-Curie Grant Agreement No. 861097, the European Union's Horizon 2020 FET OPEN project QLUSTER (Grant ID 862035), and a public grant overseen by the French National Research Agency (ANR) as part of the "Investissements d'Avenir" programme (Labex NanoSaclay, reference: ANR-10-LABX-0035). This work was done within the C2N micro nano-technologies platforms and partly supported by the RENATECH network and the General Council of Essonne.

## Author contributions

E.M. and M.G. performed the experiments, with the help of C.M. who participated in the experimental developments. E.M. analysed the data with the help of M.G. A.L., I.S., N.S. and L.L.G. fabricated the device based on a design by P.S. Numerical simulations were developed by C.M., E.M. and M.G. D.A.F., N.B., O.K. and P.S. helped with the discussion of scientific methods and data analysis. E.M. prepared all figures and wrote the initial versions of the manuscript. L.L. designed the project and supervised the experiments, data analysis and manuscript writing. All authors discussed the results and helped with the manuscript preparation.

## Competing interests

N.S. is a co-founder and P.S. is a scientific advisor and co-founder of the company Quandela. The other authors declare no competing interests.
