## [Peer Review File · Nature Communications]

Giant optical polarisation rotations induced by a single quantum dot spinREVIEWER COMMENTS

Reviewer #1 (Remarks to the Author):

A weak vertically polarized laser is reflected off a micropillar cavity with an embedded InGaAs quantum dot. The fully polarization resolved reflectivity spectra is measured in the vicinity of the electron spin-up negative trion optical transition. The device is electrically gated, and measurements in two charge states are taken: an empty state, and a statistical mix of up/down and empty. The resulting change in signal is analyzed to infer the rotation of the photon polarization conditioned on the electron spin state. The reported rotations of more than $\pi/2$, and close to π are large – particularly when one considers that a single spin is involved, and the device is a few μm in size. Potentially, such a result is significant as evidence for the technological promise of spin-photon devices, and related quantum optics devices. However, there are some issues, which are discussed below.

1. (abstract) For me, the impression created on first reading the abstract is that an experiment where the electron spin is initialized is reported. Later, when I discovered that this was not the case, I was disappointed. A good abstract should manage expectations, but more importantly should not be open to interpretation. The term “..rotation amplitudes..” lacks precision. Are you referring to a conditional or absolute rotation? As I understand, rotation amplitudes of $\pm\pi/2$ and π are theoretical targets, and in an experimental paper what the reader wants to know is what do you measure? How close to that ideal do you get? Only one input polarization is reported, and the sentence “We find that the polarization state...” needs to be a bit more precisely worded. Are you claiming control over two rotation axes?

2. (abstract, L14) “semiconductor quantum dot” This should be more specific, eg InAs. For the expert this conveys a lot of information about wavelength range etc. The non-expert will still understand the dot is a semiconductor. The generality of the results is independent of the vagueness of the language used.

3. (intro, L46) The point of the third paragraph is unclear.

4. (intro, L53-L65) This is an important paragraph since it explains the desired functionality of the device. It is hard to understand. The first few lines are probably not a sentence. The sentence (L57) needs to be articulated more precisely, and maybe in terms that directly relate to the presented experiments. Sentence beginning, L62 should come before L60. The birefringence, and magnetic fields are high compared to what? The experiments are at 1-2 Tesla which is quite strong. I would consider shifting L60 to another paragraph, see 5.

5. (intro, L66-L82) The discussion of moderate vs high birefringence is fragmented. Sentence beginning L72 is not obviously consistent with previous statement on L63. The advantages of a low birefringence device compared to high birefringence is not explained. I would consider breaking into two paragraphs. The first motivating the use of micropillars as a general spin-photon interface as paragraph 3, paragraph 4 discussing the desired device function, and paragraph 5 explaining why low birefringence is advantageous.

6. (intro L80) "Yet, until now, the spin induced polarization have remained limited by optical losses...noise". Looking at the actual data in figs 2, 3, this is still the case. A data manipulation is needed to remove the noise due to variations in the charge and spin state.

7. (intro, L103) What is the Q-factor of the device?

8. (intro, sentence beginning L102). The sentence should be more precise. Also, "full pi-rotations." This is a bold claim, and probably not defensible. The rotation angle that is inferred should be reported, this is impressive enough.

9. (intro, L114) What is the nature of the "residual noise" considered? Also L189.

10. (intro, L116) "...any orientation..can be reached." Another bold claim. To be continued...

11. (PoE, L176-L191) My feeling is the reader needs a physically intuitive picture of what determines the axis of rotation, so they can understand that in principle one could access the entire Poincare sphere. This may help them understand why the birefringence is important.

12. (results,L332-l341) According to the “here” paragraph the main claims rest on a quantitative measurement of the rotation angle of the photon polarization conditioned on the spin of the QD. This measurement is indirect, and is inferred from a fit to model. To be convinced, the reader needs to understand the key points of the fitting procedure, and to be confident of the validity of the method. As a minimum there needs to be a convincing summary in the main text of this procedure, and this is absent. When I take a look at supplementary section 11, there is no pertinent information provided. It should not be necessary to read an 11-page supplement to understand the full horrors of the underlying argument.

The magnitude of the polarization rotation angle is expressed in the data as the depth of the dip at the QD-resonance. This depends on three fit parameters: P-c, the probability of the QD being occupied in the appropriate spin state, the Cooperativity, and a parameter sigma representing a standard deviation in an unspecified quantity due to “environmental noise”. The other parameters are fixed by the empty cavity measurement. The fit is over-specified, and there is no unique fitting solution, see table 2 of supplement. The fitting solution is a surface in the parameter space, and the range of values is set by placing limits on g (i.e. the Cooperativity) and sigma. These limits are not set by an independent measurement of sigma or the Cooperativity, indeed there is no argument given for the values that are fixed. In effect, this is not a measurement. The values for the rotation angle are a choice, and this is really uncool. Please clarify the data processing.

13. A potential issue is optical pumping of the spin. This is not mentioned in the main text, but is discussed in supplement sec. 2. (p4, “no initialisation”) There is simply an assertion that the system operates in a particular regime where optical pumping is not relevant. The assertion is not supported by either measurements of the relevant timescales, or by references to measurements of typical timescales in the literature.

14. In fig 4(a), extracted data is presented to argue that a phase-shift of $\pm\pi/2$ can be achieved, i.e. a D/A state can be attained. However, whilst 3d plots look nice, it is difficult looking at fig 4(a) , particularly in the case of D, to see that the desired effect is achieved. In (b), a different perspective is shown which is clearer, but in a demonstration the focus is on

the measured results, not the theory curves. Perhaps it would be good to reproduce the data of 3D plots in the supplement, using 2D plots of Stokes-vector components vs the control parameter (detuning?).

15. In fig. 4, it looks as if the polarization rotates about an axis lying in the AHAD plane, which is controlled via the B-field – perhaps by tuning the cavity/optical transition detuning, although this is unclear. It looks as if there is a limit to how close this axis can be tilted towards A or D, limiting access to the H-state, i.e. pi-shift. This kind of undermines the claim of pi-shift made in the “here..” paragraph, see 7.

16. (Methods) Sufficient information to reproduce the experiments should be provided, possibly in the supplement. If the information can be found in refs, then a clear statement such as “Full details of the device, and experimental setup can be found in ref. “ should be made. In particular, quantitative information on the doping, and dimensions of the micropillar are absent. Likewise, the optics is sketchy. The polarization properties will be important here. What NPBS do you use? How do you calibrate/compensate for birefringence of setup? Is the light focused by a room temperature objective through a strained cryostat window, or is a cold objective used? How is the mode-matching achieved? What is the NA of the micropillar?

17. Ref. [32] is in Phys. Rev. applied. Check the refs.

To summarize the report. The device is state of the art, and the raw data of fig. 2, 3(measured) clearly shows a giant polarization rotation conditioned on the QD spin. This is impressive. However, a paper needs to be measured against the claims made. According to the abstract, “We experimentally demonstrate rotation amplitudes such as $\pi/2$ and π in the Poincare sphere, as required for applications based on spin-polarisation mapping and spin-mediated photon-photon gates.” Ideally, such claims would be supported by direct measurements. By inspection, the “extrapolated data” does not pass through H, and thus claim of pi rotation is simply not true. Probably it is not strictly true even in theory. To accept the validity of the claim of $\pi/2$ rotation, the reader needs to understand the data processing/manipulation, including underlying assumptions, and uniqueness of fit. A favourable endorsement will largely be decided by the strength of response to point 12.

Similar questions can be asked of the title. If the electron is not initialized, there is no control over the spin state so the title is not accurate, and needs to change.

Overall, the writing is imprecise, a greater care over the accuracy of statements, and more attention to communicating key points is needed. Whilst the dots are often there, the narrative does not join the dots, and construct full and effective arguments. For example, there is no clear explanation of what determines the angle or axis of polarization rotation, or why weak birefringence is desirable. The paper would benefit from a “brief history” of the device evolution, so the reader can understand where the current device fits. A quantitative benchmarking of rotation angles achieved in prior works would also be beneficial to place the current work in context.

Reviewer #2 (Remarks to the Author):

In this manuscript, E. Mehdi and coauthors employed a structure composed of a singly-charged semiconductor quantum dot embedded in a high-Q pillar-based microcavity, to show spin-selective rotations of the polarization of the reflected photons, done in a deterministic way and possible to operate in configurations with zero or low magnetic field. Building deterministic light-matter interfaces, in particular with solid-state spins as stationary qubits, is an important current topic in the realm of optical quantum information science and hence the focus of the manuscript is of relevance and importance in the field. Before consideration for publication, there are multiple points/questions that need to be clarified:

1- It is crucial to properly present the exact distinction between the current work and the recently published work in the area. In particular, authors need to address the difference between their work and previous works such as [Androvitsaneas, Petros, et al. "Efficient quantum photonic phase shift in a low Q-factor regime." ACS Photonics 6.2 (2019): 429-435] and [Wells, L. M., et al. "Photon phase shift at the few-photon level and optical switching by a quantum dot in a microcavity." Physical Review Applied 11.6 (2019): 061001.] and [Antón, Carlos, et al. "Tomography of the optical polarization rotation induced by a single quantum dot in a cavity." Optica 4.11 (2017): 1326-1332.] which have also employed negatively charged QDs in micropillar cavities to realize spin-photon interfaces, in more detail.

- 2- How critical is the quality factor in achieving the targeted phase shift and what are the consequences of having low-Q cavities? For instance, in [Androvitsaneas, Petros, et al. "Efficient quantum photonic phase shift in a low Q-factor regime." ACS Photonics 6.2 (2019): 429-435] a low-quality cavity is used to achieve phase shifts as large as $2\pi/3$.
- 3- Please comment on the quantitative effect of the Purcell enhancement in the process, such as the achievable phase shifts and points on the Poincare sphere in general, purity of the reflected photonic states, and other relevant parameters.
- 4- As mentioned in the "Principle of the experiments" section, a few T magnetic field is used (2T, 1.3T, 1.7 T in different presented data) to build the Zeeman splitting and selective interaction with the QD energy levels. So, the authors need to clarify the comment in the abstract regarding the possibility of using this configuration "with zero or low magnetic field". Does this mean to suggest that the magnetic field is not required to induce spin-selective polarization rotations? Has the experiment been conducted for zero magnetic fields as well? There are some brief discussions in the supplementary document about this, which I suggest having further explained and possibly moved to the main text.
- 5- In Fig. 2(a), what is the reason to get larger I^{avg}_V at the resonance point for the larger magnetic field compared to smaller values of B? Also, please explain why the resonant behavior is observed at the detunings corresponding to $\omega_{\text{laser}} - \omega_{\text{cav}} = \omega_{\text{QD}}^{\text{up}}$ and not at $\omega_{\text{laser}} = \omega_{\text{QD}}^{\text{up}}$
- 6- Here the damping rates of the cavity for H and V modes are in proximity with the separation of resonance frequencies for these modes. Could the authors comment on the dependence of the spin-selective Kerr rotations in this scheme on the ratio of the damping to the mode separation? Have numerical simulations been conducted as well?
- 7- How many times in the measurement interval of 0.1 seconds for the intensities does the co-tunneling process for the electron in the QD happen? This strictly gauges the degree of validity for equation 1.
- 8- How is the self-consistent fitting done to extract P_{up} or P_{c} in Fig 3? Is P_{c} merely a fitting parameter in numerical simulations or has this parameter been experimentally extracted?
- 9- The underlying reason for having spin-selective phase shifts is not very clearly explained in the manuscript. There is a brief discussion before the "results" section in this regard; however, this needs to be explained much more clearly in the manuscript, especially in

connection with the role of photon polarization (right-handed and left-handed as shown in Fig. 1) in the ground states to trion states population transfer and bringing the laser into resonance with the ω_{QD} .

10- In Fig. 4(b) the numerical result on the Poincare sphere for $B=1.69\text{T}$ is very different from the results for $B=1.7\text{T}$ (Fig. 3(b) and Fig. 4(a)), why?

11- The difference between the dashed and solid lines cannot be really seen in Fig. 4(b). I suggest modifying the plot to make the distinction more apparent.

Reviewer #3 (Remarks to the Author):

The manuscript by E. Mehdi et al. presented a study of the manipulation of the polarisation of the reflected light from a cavity-QED device consisting of a single QD and a photonic cavity with electrical contacts. When the driving laser is in resonance with the QD transition, the polarisation of the reflected light changes. This change in polarisation was explained and modelled as an effect of the interference between the empty cavity reflection and the quantum resonance fluorescence. The modelling also gave an extrapolated polarisation rotation hypothetically induced by a pure spin-up QD state and the extrapolated Stokes vector (S_{up}) indeed shows a giant rotation at a certain laser frequency. Furthermore, the authors presented a similar polarisation rotation when the QD resonance shifts with the magnetic field and simulations of the impact of environmental noise on the reflected polarisation.

The authors motivated their study with a fascinating scope of producing “a stable, controllable, spin-dependent photon state, in a deterministic way” and used “Controlling photon polarisation with a single spin” as the title of the manuscript. However, after reviewing the manuscript and the supplementary materials, the reviewer thinks some claims are not well supported by the results or the relevant discussions. It is not clear how this work would actually benefit the research of quantum gates between incoming photons and stationary qubits. Several specific questions are as follows:

1. In order to support the claim in the title, “Controlling photon polarisation with a single spin”, the ability to initialise the spin state is critical. However, spin initialisation is not

presented or discussed here, even though polarisation rotations induced by a pure spin-up state was used to demonstrate the giant rotations.

2. Another important parameter is the efficiency of the spin-photon interface. An analysis of the efficiency would provide a more complete picture of the results. The intensities observed at different magnetic fields in Fig. 2a seem to differ, so it would be interesting to figure out whether the efficiency changes with cavity-laser detuning or with the magnetic field? Since the efficiency also affect the overall fidelity of a spin-photon interface, the fidelity should be clearly defined before presenting “a fidelity of 90.5 %” on line 439.

3. Although a single photon detector was equipped, this experiment was not set up in a single photon regime and averaged intensity was measured. Is there a particular reason for this?

4. The spin flip rates and lifetimes are key to understand the spin dynamics of the QD spin states. Currently, some rates were chosen to be drastically different for simulation purposes. This could be confusing for readers and a proper introduction to any existing knowledge about these rates would be useful.

Response to the reviewers' comments

In this response the reviewers' comments are in italic, with green color. Our response text is in black, normal font. Our revised sentences are marked with indentations and quotation marks, using the red color for modified text and black color for the unmodified one.

Reviewer #1 (Remarks to the Author):

“A weak vertically polarized laser is reflected off a micropillar cavity with an embedded InGaAs quantum dot. The fully polarization resolved reflectivity spectra is measured in the vicinity of the electron spin-up negative trion optical transition. The device is electrically gated, and measurements in two charge states are taken: an empty state, and a statistical mix of up/down and empty. The resulting change in signal is analyzed to infer the rotation of the photon polarization conditioned on the electron spin state. The reported rotations of more than $\pi/2$, and close to π are large – particularly when one considers that a single spin is involved, and the device is a few μm in size. Potentially, such a result is significant as evidence for the technological promise of spin-photon devices, and related quantum optics devices. However, there are some issues, which are discussed below.”

We thank Reviewer 1 for his/her in-depth reading and numerous suggestions, together with his/her positive assessment of the results' significance. As discussed below, we have happily taken into account all the comments to provide a strongly-revised version of our manuscript.

“1. (abstract) For me, the impression created on first reading the abstract is that an experiment where the electron spin is initialized is reported. Later, when I discovered that this was not the case, I was disappointed. A good abstract should manage expectations, but more importantly should not be open to interpretation. The term “..rotation amplitudes..” lacks precision. Are you referring to a conditional or absolute rotation? As I understand, rotation amplitudes of $\pm\pi/2$ and π are theoretical targets, and in an experimental paper what the reader wants to know is what do you measure? How close to that ideal do you get? Only one input polarization is reported, and the sentence “We find that the polarization state...” needs to be a bit more precisely worded. Are you claiming control over two rotation axes?”

We fully understand how this first impression was created by the previous version: the relevant sentence of the abstract now reads:

“[...] deduce the output polarisation Stokes vector conditioned by a **specific spin state, without spin initialization**. [...]”.

We note that the same clarification has been also added in our revised introduction, at L114.

As regards the target states, we also clarified that they were experimentally approached and give the corresponding extrapolated fidelities (also detailed in the main text), to show “how close to the ideal” we could get. Our phrasing takes care in expressing the fact that a fidelity is measured with respect to some ideal state, which is different from the notion of “rotation angle” (see also our response to item 8). The corresponding sentence of the abstract now reads:

“[...] We experimentally **approach polarisation states conditionally rotated by $\pi/2$, π , and $-\pi/2$** in the Poincaré sphere – **desired for applications based on spin-polarisation mapping and spin-**

mediated gates – with extrapolated fidelities of $(97\pm 3)\%$, $(85\pm 10)\%$, and $(92\pm 5)\%$, respectively. [...]”.

Also, we indeed had to clarify the sentence “*We find that the polarization state can be controlled in most of the Poincaré sphere...*”. We also refer to our response to item 10, which is strongly related to this point. The revised sentence in the abstract now reads:

“[...] We find that an enhanced light-matter coupling, together with limited cavity birefringence and reduced spectral fluctuations, allow targeting most states of the Poincaré sphere, with a control both in longitude and latitude, starting from a fixed incoming state [...]”.

All these changes in the abstract are consistently reflected in other changes in the main text, most of them discussed in the following parts of our response.

“2. (abstract, L14) “semiconductor quantum dot” This should be more specific, eg InAs. For the expert this conveys a lot of information about wavelength range etc. The non-expert will still understand the dot is a semiconductor. The generality of the results is independent of the vagueness of the language used.”

This suggestion now appears in the revised abstract.

“3. (intro, L46) The point of the third paragraph is unclear.”

We do realize that, indeed. The introduction has been strongly revisited (see also our response to the next items), and this third paragraph has been removed, with one of its sentences rewritten and included in the previous (second) paragraph. This sentence now reads:

“[...] In the optical domain, in particular, potential spin-photon interfaces have been explored with a number of solid-state emitters and cavity structures [22, 23], as well as various encodings, including polarisation, path, and time-bin [10-14]. [...]”.

“4. (intro, L53-L65) This is an important paragraph since it explains the desired functionality of the device. It is hard to understand. The first few lines are probably not a sentence. The sentence (L57) needs to be articulated more precisely, and maybe in terms that directly relate to the presented experiments. Sentence beginning, L62 should come before L60. The birefringence, and magnetic fields are high compared to what? The experiments are at 1-2 Tesla which is quite strong. I would consider shifting L60 to another paragraph, see 5.”

We thank the reviewer for all these suggestions. The paragraph is now more clearly and explicitly intended to explain the device functionality. Its first sentence is also clarified, and the discussion of the birefringence and of the magnetic field is shifted to another paragraph, as suggested (see next item 5). The paragraph now reads:

“[...] Polarisation encoding, in this respect, has the advantage of providing straightforward 1-qubit gates and measurements, as well as conceptually-simple protocols for various spin-photon and multi-photon gates [24-27]. A key objective is to produce a perfect spin-polarisation mapping : starting from a fixed incoming photon state, $|\Psi_{in}\rangle$, and depending upon a spin state $|\uparrow\rangle$ or $|\downarrow\rangle$, an ideal device would deterministically produce states of orthogonal polarisations, namely $|\Psi_{out}\rangle = |\Psi_{\uparrow}\rangle$ or $|\Psi_{\downarrow}\rangle$ with $\langle\Psi_{\uparrow}|\Psi_{\downarrow}\rangle=0$. [...]”.

“5. (intro, L66-L82) The discussion of moderate vs high birefringence is fragmented. Sentence beginning L72 is not obviously consistent with previous statement on L63. The advantages of a low birefringence device compared to high birefringence is not explained. I would consider breaking into two paragraphs. The first motivating the use of micropillars as a general spin-photon interface as paragraph 3, paragraph 4 discussing the desired device function, and paragraph 5 explaining why low birefringence is advantageous.”

Following this useful advice, we indeed rewrote all this part of the introduction by splitting in various paragraphs. There is now a separate paragraph explaining, first, the principle of the approaches based on both high-birefringent cavities and high-magnetic fields. This was the opportunity for us to make explicit the definition of these two notions, as suggested in item 4. Then, we included another paragraph to explain the other desirable approach, based on moderate birefringence (which is thereby defined). These two paragraphs now read:

“In this respect, most realizations have been pioneered using both **high magnetic fields and highly-birefringent cavities [7,10, 11], thus suppressing any spectral overlap between optical transitions and orthogonally-polarised cavity modes.** In such a case, only one transition, and only one of the cavity polarisation eigenmodes, can be excited by a given input. For a perfect device, this allows exploiting the π phase-shift induced by the excited transition to implement, ideally, a conditional π polarisation rotation [28].

Alternatively, a promising strategy is to use cavity-QED devices with moderate birefringence, i.e. exhibiting a **spectral overlap between orthogonally-polarised cavity modes [29-31].** In such a configuration, perfect spin-polarisation mapping can also be obtained at zero magnetic field, through opposite, $\pm \pi/2$ rotations in the Poincaré sphere for the states $|\Psi_{\uparrow}\rangle$ and $|\Psi_{\downarrow}\rangle$ [32]. This ensures compatibility with a variety of protocols based on deterministic quantum gates [24,33] and deterministic entanglement between multiple photons [25,26].”

We also refer to our answer to item 11, and the corresponding changes in the main text and Supplemental Materials, where the advantage of low birefringence is much more explicitly explained.

“6. (intro L80) “Yet, until now, the spin induced polarization have remained limited by optical losses...noise”. Looking at the actual data in figs 2, 3, this is still the case. A data manipulation is needed to remove the noise due to variations in the charge and spin state.”

We realize that this sentence was confusing, as it superposed different concepts and did not define the word “noise”. Here we meant spectral fluctuations (which tend to blur the conditional phase/polarisation shifts), rather than jumps between the ground states (which do not diminish the conditional phase/polarisation shift for a *given* ground state). We thus modified our sentence to:

“Yet, until now, **conditional** spin-induced rotations have remained limited **in angle, due to optical losses [32] and/or detrimental spectral fluctuations [32,34,35].**”

A number of other modifications were included to clarify the notion of “noise”: they are described in related points below, in particular item 9.

“7. (intro, L103) What is the Q-factor of the device?”

This is now indicated in the Methods of the revised version, at L627:

“The corresponding quality factor slightly differs for the two polarizations, with $Q_H=8650$ and $Q_V=8300$, respectively (see Supplemental section 7).”

“8. (intro, sentence beginning L102). The sentence should be more precise. Also, “full pi-rotations.” This is a bold claim, and probably not defensible. The rotation angle that is inferred should be reported, this is impressive enough.”

We fully agree that the sentence had to be clarified, and that “full π rotations” is a misleading wording (especially since “full” could be understood as “ideal”, as would be the case for a pure state). We also refer to our response to item 1, i.e. the way we modified the abstract to fairly express what we achieved.

Regarding our introduction, the corresponding sentence is now divided in two sentences:

“The Purcell enhancement provided by the high-Q cavity, and a strong reduction of spectral fluctuations compared to previous works [32,34,35], allow reaching giant polarisation rotations. This includes highly-desired configurations such as $\pm \pi/2$ and π rotations, though a degradation of the polarization purity, down to around 70%, is observed at the larger angles.”

We choose such a wording, as we find it misleading to attribute an angle different from π to describe the observed/predicted behavior in the Poincaré sphere. Of course, we do understand that an angle could have been derived from one Stokes parameter, through some simplified formula, such as “angle= $\arccos(-s_{HV})$ ”, as done in previous works. Yet, such an equation strongly suggests to the readers a wrong image, where pure polarisation states are obtained, whose projection on the HV axis indicates the “angle”.

Actually, this question highlights an originality of our work, unrelated to technological performance, and which should participate to the scientific impact of our manuscript. Unavoidably, the complete “3-axes” approach, where the output vector is fully measured, brings a qualitatively different perspective that requires a different way to discuss phenomena. To help understanding this point, we made a number of additional changes:

- First, we rewrote the corresponding sentence of the introduction (see L104), where the approach of most previous works is commented:

“[...] spin-induced polarisation rotations have mostly been measured via intensity contrasts in a given basis. This is equivalent to a single-axis projection in the Poincaré sphere, which is sufficient to deduce a fidelity to a specific target, yet does not give access to the actual polarisation states produced in the experiments. [...]”

- Also in the introduction (see L116), we rewrote the sentence describing our claim:

“In agreement with theoretical simulations, we find that residual spectral fluctuations do not limit the achievable latitude and longitude angles for the output state, yet limit its polarisation purity.”

- in the “Discussion” section, we explicitly take an example of π rotation in longitude with degraded purity, referring to the Methods section for rigorous definitions.

“[...] the QD-laser detuning can be set such that the Stokes vector points towards $|H\rangle$, with $s_{HV}^\uparrow=0.81$ and $s_{DA}^\uparrow = s_{RL}^\uparrow = 0$. In such a case the Stokes vector has experienced a π rotation in longitude while keeping its latitude at 0, the limitation being entirely due to the polarisation purity degraded down to 0.81 (see Methods).”

- Finally, in the Methods section (L674), we added a proper general way of deducing longitude and latitude angles from complete tomography:

“[...] Its latitude θ , defined as the angle between the Stokes vector and the equatorial (HDVA) plane, is then geometrically expressed as $\theta = \arcsin\left(s_{RL}/\sqrt{s_{HV}^2 + s_{DA}^2 + s_{RL}^2}\right)$. Its longitude ϕ , defined as the angle between the $|H\rangle$ state and the projection of the Stokes vector in the equatorial plane, is expressed as $\phi = \text{sgn}(s_{DA}) \arccos\left(s_{HV}/\sqrt{s_{HV}^2 + s_{DA}^2}\right)$.”

“9. (intro, L114) What is the nature of the “residual noise” considered? Also L189.”

This is related to item 6, as the notion of noise had to be clarified in multiple instances. Regarding the sentence mentioned here (previous L114, now L116), we explicitly replaced the notion of “noise” by the specific wording “spectral fluctuations”. Regarding the previous line L189, we also clarified why such spectral fluctuations play an important role:

“[...] a pure state $|\Psi_\uparrow\rangle$ requires stable reflection coefficients, and thus a stable transition energy $\omega_{\text{QD}}^\uparrow$. In the presence of spectral fluctuations, a rotation of polarisation is still expected for the conditional Stokes vector \vec{S}_\uparrow , yet with a degraded polarisation purity, [...]”

In addition, it was indeed important to discuss the nature of these spectral fluctuations, and we thus added the following paragraph:

“In general, spectral fluctuations can have various causes, including electrostatically-induced fluctuations as well as hyperfine interaction between the confined electron/hole and the 10^4 - 10^5 nuclear spins in the quantum dot [43]. Actually, the standard deviation $\sigma_{\text{SF}} = 0.5 \mu\text{eV}$ can be almost entirely explained by hyperfine interaction. If that were strictly the case, one would have $\sigma_{\text{HI}}^{(e)} = 2 \sigma_{\text{SF}} = 1 \mu\text{eV}$, with $\sigma_{\text{HI}}^{(e)}$ the standard deviation of the Zeeman splitting mainly induced by the electron-nuclei interaction. Such a value of $\sigma_{\text{HI}}^{(e)}$ is in agreement with the ones observed in similar, strongly-annealed quantum dots [44], and only slightly larger than the value $\sigma_{\text{HI}}^{(e)} = 0.8 \mu\text{eV}$ used to reproduce other experiments in a different device from our group [45].”

“10. (intro, L116) “...any orientation..can be reached.” Another bold claim. To be continued...”

We realize that indeed the claim was bold, and it actually corresponds to a true mistake that we made. We had already checked, of course, that all longitudes and latitudes can be reached. We were wrongly thinking that our simulations would naturally cover all possible orientations if more detunings (or, equivalently in Fig. 4b, more magnetic fields) were explored, as preliminary results seemed to indicate.

However, when going through all simulations at all potential detunings, it appears that there always remains a small region corresponding to orientations, i.e. *pairs* of latitude and longitude,

that *cannot* be reached. This is now discussed in the Supplemental section 7, accompanying extended data also suggested by Reviewer 1 (item 14). The corresponding region is best seen in the new figure S11 of the Supplemental Materials (reproduced below), where two opposite sides of the Poincaré sphere are shown. In this figure the cumulated simulations are displayed as orange lines covering most of the Poincaré sphere, except for a small forbidden region in black. It was indeed crucial to spot this mistake and we are very thankful to Reviewer 1 for his numerous questions on the subject, in this item and in following ones.

Fortunately, the existence of such forbidden region does not diminish the interest of our approach, since it corresponds to a subset of *small* (both in latitude and longitude) rotations with respect to the incoming state V, and only in a single “quadrant” (positive rotations in longitude combined with some negative rotations in latitude). We had not looked into that specific region since it is approached only for large detunings. As such, it does not include any of the directly useful states that we can wish to realize: π rotation is directly useful at high magnetic field (state H), $\pi/2$ rotations (all states on the DRAL meridian) can be used at zero field, while all intermediate rotations between $\pi/2$ and π are available to use for intermediate fields.

In addition to all the related changes and additions we made to the Supplemental Materials, we also modified accordingly our abstract (see the corresponding change in our response to item 1), as well as our introduction:

“[...] We finally show that, by a proper set of detunings, **most orientations** of the output Stokes vector \vec{S}_1 can be reached. [...]”,

and also our “Discussion” section:

- at L457:

“[...] These experimental results indicate the possibility to generate **most polarisation states in the Poincaré sphere** through a proper setting of the experimental parameters [...]”

- at L475:

“To illustrate **the diversity of possible output states**, we also show in Fig. 4b simulations [...]”

- and at L482:

“[...] In the absence of **spectral fluctuations**, the polarisation of the reflected photons can reach **most states** at the surface of the Poincaré sphere, **through a wide range of combinations in latitude and longitude** (see Supplemental section 7). [...]”

“11. (PoE, L176-L191) My feeling is the reader needs a physically intuitive picture of what determines the axis of rotation, so they can understand that in principle one could access the entire Poincare sphere. This may help them understand why the birefringence is important.”

We thank Reviewer 1 for this suggestion, which is also at the core of many important questions from Reviewer 2’s report. Strong changes in the manuscript have been made to help this crucial understanding. **There is now a whole section 2 in the Supplemental Materials, discussing the reflectivity coefficients and polarisation rotations.** This section includes explicit notations and (new) analytical expressions as a function of the device parameters. Such analytical approach is based on the semiclassical approximation, yet we numerically show it to be exact in the low-power regime and in the absence of noise/fluctuations. We also explain why, in presence of slow fluctuations (either spectral fluctuations or jumps between states, such as those induced by co-tunneling), the result of exact numerical calculations can simply be seen as the averaging of results deduced from the semiclassical approximation.

In addition, we modified extensively the main text, so that the reader can get quite a complete picture, without having to look at the analytical expressions in the Supplemental Materials. We avoided mentioning “two axes of rotation”, since there would be no possible analytical expression describing such “two axes”. Instead we use the HV axis as a natural reference (since $|\Psi_{in}\rangle = |V\rangle$) and show that our protocol allows us to vary the polarization state both in terms of projection along this axis, and in terms of rotation around this axis:

“[...] As detailed in Supplemental section 2, such rotations can be interpreted in the so-called semiclassical approximation, which is valid in the low-power regime (negligibly-populated trion states), and when neglecting all sources of fluctuations. In such a case \vec{S}_\uparrow corresponds to a pure polarisation $|\Psi_\uparrow\rangle$, with:

$$|\Psi_\uparrow\rangle = \frac{r_{V\rightarrow H}^\uparrow |H\rangle + r_{V\rightarrow V}^\uparrow |V\rangle}{\sqrt{|r_{V\rightarrow H}^\uparrow|^2 + |r_{V\rightarrow V}^\uparrow|^2}} \quad (1)$$

where $r_{V\rightarrow H}^\uparrow$ and $r_{V\rightarrow V}^\uparrow$ denote complex reflection coefficients, depending on the device parameters and on the various detunings between the laser, the $|\uparrow\rangle - |\downarrow\uparrow\rangle$ transition, and the two cavity modes. Their expressions are detailed in Supplemental section 2. They respectively govern the H-polarised and V-polarised contributions to the reflected output field, in the case where $|\Psi_{in}\rangle = |V\rangle$. [...]

“[...] Importantly, both $r_{V\rightarrow V}^\uparrow$ and $r_{V\rightarrow H}^\uparrow$ depend on two experimentally-tunable parameters, ω_{laser} and ω_{QD}^\uparrow , that can be independently varied. This provides the two required degrees of freedom to control the position of $|\Psi_\uparrow\rangle$ in the Poincaré sphere: the modulus (respectively, the phase) of the ratio $r_{V\rightarrow H}^\uparrow/r_{V\rightarrow V}^\uparrow$ governs the projection of the Stokes vector on the HV axis (respectively, its relative orientation around the HV axis). [...]”

The introduced notations strongly helped us discussing various other situations in the main revised text and in the revised Supplemental Materials. For example, regarding the conditions to reach various target states, at L460:

“[...] the rotated polarisation state, given by Eq. (1) in absence of fluctuations, can be controlled by varying both coefficients $r_{V\rightarrow V}^\uparrow$ and $r_{V\rightarrow H}^\uparrow$ through the experimentally-controlled parameters ω_{laser} and ω_{QD}^\uparrow . By using magnetic fields of 1.3 T, 1.7 T, and 2.1 T, we respectively targeted to

reach, at specific QD-laser detunings, an ideal polarisation state $|A\rangle$ (i.e. $r_{V\rightarrow V}^\uparrow = -r_{V\rightarrow H}^\uparrow$), $|H\rangle$ (i.e. $r_{V\rightarrow V}^\uparrow = 0$), and $|D\rangle$ (i.e. $r_{V\rightarrow V}^\uparrow = -r_{V\rightarrow H}^\uparrow$). [...]"

Also, as suggested by Reviewers 1 and 2, this helped us clarifying what is the effect of a moderate birefringence, and why it is desirable, specifically at low/moderate field.

- at L502:

"[...] This is especially promising for reaching perfect spin-polarisation mapping, i.e. $\langle \Psi_\uparrow | \Psi_\downarrow \rangle = 0$, at zero magnetic field. Indeed, the cross-polarising coefficients carry a spin-dependent sign, while the co-polarising coefficients do not (see Supplemental section 2). Thus, when $\omega_{QD}^\uparrow = \omega_{QD}^\downarrow$, $r_{V\rightarrow H}^\uparrow = -r_{V\rightarrow H}^\downarrow$, while $r_{V\rightarrow V}^\uparrow = r_{V\rightarrow V}^\downarrow$. As an example, if the QD transition energy at zero field could have been tuned to $\omega_{QD}^\uparrow = \omega_{QD}^\downarrow = \omega_{QD}^{cav} + 14.1 \mu\text{eV}$, the spin states $|\uparrow\rangle$ and $|\downarrow\rangle$ would almost have been mapped to the opposite polarisation states $|\Psi_\uparrow\rangle = |A\rangle$ and $|\Psi_\downarrow\rangle = |D\rangle$. Conversely, in a highly-birefringent device, such a feature would be impossible due to $r_{V\rightarrow H}^{\uparrow/\downarrow} = r_{H\rightarrow V}^{\uparrow/\downarrow} = 0$. Indeed, in absence of spectral overlap between H and V modes, it becomes impossible to both excite the QD through one mode and force it to emit into the other, far-detuned mode (see Supplemental sections 2 and 6). [...]"

- at L531:

"[...] (iv) A moderate birefringence, allowing Purcell-enhanced emission in both polarisations. The condition of moderate birefringence, in particular, is essential to allow converting light polarisation from one mode to the other, i.e. reach significant values for $r_{H\rightarrow V}$ and $r_{V\rightarrow H}$ (see Supplemental sections 2 and 6). [...]"

- and in the concluding paragraph at L572:

"Finally, we note that this control of the polarisation states, in moderately birefringent devices, provides a general approach to implement perfect spin-polarisation mapping at any magnetic field. Starting from a trivial input state such as $|\Psi_{in}\rangle = |V\rangle$, one can indeed maximize the distance between $|\Psi_\uparrow\rangle$ and $|\Psi_\downarrow\rangle$, and eventually reach the desired condition $\langle \Psi_\uparrow | \Psi_\downarrow \rangle = 0$, equivalent to $(r_{V\rightarrow H}^\uparrow)^* r_{V\rightarrow H}^\downarrow + (r_{V\rightarrow V}^\uparrow)^* r_{V\rightarrow V}^\downarrow = 0$. [...]"

"[...] With high magnetic fields, using highly-birefringent cavities [7, 10,12, 50] may also allow the engineering of perfect spin-polarisation mapping, yet this requires using non-trivial input states to compensate for the negligible coefficients $r_{H\rightarrow V}$ and $r_{V\rightarrow H}$ (see Supplemental section 6). This renders all the more necessary to include the tomography approach in future experiments with polarisation-encoded interfaces, to adapt the quantum gates and protocols to each specific configuration, and reach high-fidelity operations even with imperfect devices."

"12. (results,L332-I341) According to the "here" paragraph the main claims rest on a quantitative measurement of the rotation angle of the photon polarization conditioned on the spin of the QD. This measurement is indirect, and is inferred from a fit to model. To be convinced, the reader needs to understand the key points of the fitting procedure, and to be confident of the validity of the method. As a minimum there needs to be a convincing summary in the main text of this procedure, and this is absent. When I take a look at supplementary section 11, there is no pertinent information provided. It should not be necessary to read an 11-page supplement to understand the full horrors of the underlying argument. "

“The magnitude of the polarization rotation angle is expressed in the data as the depth of the dip at the QD-resonance. This depends on three fit parameters: P_c , the probability of the QD being occupied in the appropriate spin state, the Cooperativity, and a parameter σ representing a standard deviation in an unspecified quantity due to “environmental noise”. The other parameters are fixed by the empty cavity measurement. The fit is over-specified, and there is no unique fitting solution, see table 2 of supplement. The fitting solution is a surface in the parameter space, and the range of values is set by placing limits on g (i.e. the Cooperativity) and σ . These limits are not set by an independent measurement of σ or the Cooperativity, indeed there is no argument given for the values that are fixed. In effect, this is not a measurement. The values for the rotation angle are a choice, and this is really uncool. Please clarify the data processing.”

“13. A potential issue is optical pumping of the spin. This is not mentioned in the main text, but is discussed in supplement sec. 2. (p4, “no initialisation”) There is simply an assertion that the system operates in a particular regime where optical pumping is not relevant. The assertion is not supported by either measurements of the relevant timescales, or by references to measurements of typical timescales in the literature. “

We discuss items 12 and 13 simultaneously, as our answers and modifications regarding both points are strongly connected, and are also connected to the other Reviewers’ comments. We thank Reviewer 1 for suggesting numerous ways in which our parameter discussion and our main assertions had to be drastically clarified. **We strongly rewrote our explanations and discussions, in the main text but also through a full rewriting of most of the Supplemental sections, including additional data and fits.** In particular, we could strongly simplify the reading of the Supplemental Materials by discussing the parameter fits, already in the Supplemental section 3, after the useful theoretical elements have been introduced. We realized that many complexities, in our previous version of the Supplemental Materials, were unnecessary and could even misleadingly convince the reader that many parameters were kept unknown.

Before discussing our changes, we would like to briefly explain our initial choice of presenting only the intensity fits, thinking that our explanation of the (maximal) 6% uncertainty on P_c (and thus, the maximal 3% uncertainty on P_\uparrow) was convincing enough. Actually we performed other experiments, especially using single detected photons to project the system in a given spin/charge state, subsequently measuring this state through a second photon. We initially left these aspects out of the discussion, to preserve the novelty of future papers¹. Yet we understand that at least part of our additional analyses has to be included in the manuscript. We now discuss our changes in the main text, and in parallel **we also refer to our fully-revisited Supplemental sections 2 to 5.**

First, we modified the “Principle of the Experiments” section to explicitly give the information regarding the electron escape time in our device, and point to the (new) Supplemental section 5 where the justifying data can be found. Discussions regarding the impact of this co-tunneling escape time are included in this same paragraph, also pointing to the (entirely rewritten) Supplemental section 4 where these aspects are fully discussed:

¹ We plan to submit at least one additional manuscript with data from the exact same device, in particular in Voigt configuration where we deduce both spin coherence (related to fluctuations of the Zeeman splitting, dominating σ_{SF}), and charge/spin lifetime (dominated by cotunneling). All the data have already been shown to fit our simulations when taking the same parameters as in our manuscript (with a voltage-dependent P_c , maximized at 0.94, as in the present case).

“As detailed in the Supplemental section 5, the device under study operates in a rapid co-tunneling regime, where a trapped electron escapes the quantum dot in typically 4 ns, and is directly replaced by another electron from the Fermi sea. Even in such conditions, the radiative transitions can still be considered stable-enough to provide a well-defined, state-dependent optical response, since their Purcell-enhanced emission time is around 200 ps [...]

[...] Notably, the co-tunneling regime prevents from initializing the spin by optical spin pumping, as each co-tunneling event implies a loss of spin memory, leading to $P_{\uparrow}=P_{\downarrow}=P_c/2$, with P_c the charge occupation probability (see Supplemental section 4)”

Such an introduction of new material does not increase the complexity of the parameter discussion: the latter is now clarified and explained earlier on in the Supplemental Materials (section 3). For such discussion we use the fact that, consistently with exact numerical resolution, all the effects can be interpreted in terms of the reflectivity coefficients, with an additional averaging over intensities with some probability laws (where the additional parameters appear).

The fitting procedure also had to be clarified in the main text, which we started by discussing the unambiguously-extracted parameters:

“[...] First, all cavity parameters have been unambiguously extracted by fitting experimental data obtained in the empty-cavity regime (see extended data in Supplemental section 7). [...]

[...] Then, most of the quantum dot parameters are unambiguously extracted, by fitting the average intensity measurements $I_{H/V}^{avg}$, in various conditions. This includes the parameters governing the magnetic field response, and thus ω_{QD}^{\uparrow} , deduced from the resonance frequencies measured at various magnetic fields. This also includes the QD-cavity coupling constant $g = (15 \pm 1) \mu\text{eV}$ and the spontaneous decay rate $\gamma_{sp} = (0.35 \pm 0.05) \mu\text{eV}$, describing the emission in all spatial modes other than the two fundamental cavity modes H and V. These two parameters are quite precisely estimated, even in presence of other uncertain parameters, thanks to the fact that they directly influence the amplitude, shape and width of the QD-induced optical response. [...]

The fact that the light-matter coupling g is almost unambiguously extracted (its $1\mu\text{eV}$ uncertainty has a minor effect), as well as γ_{sp} , strongly helps simplifying the discussion of these two remaining parameters in the main text, at L295:

“Finally, the two remaining parameters of our model are related to the expected sources of noise. First, spectral fluctuations are considered, and described by a Gaussian distribution of the transition energy ω_{QD}^{\uparrow} , with a standard deviation σ_{SF} . In addition, a non-unity value of the charge occupation probability P_c is also considered. As discussed in Supplemental section 3, fitting the measured intensities alone shows that these parameters are constrained by a line in the parameter space, bounded by extremal points ($P_{c,min} = 0.88$; $\sigma_{SF,min} = 0 \mu\text{eV}$) and ($P_{c,max} = 1$; $\sigma_{SF,max} = 0.8 \mu\text{eV}$).

With these two bounds in mind, we can consider that $P_c = 0.94 \pm 0.06$, yet reminding that the uncertainty of 6% is taken as a conservative estimate. Indeed, the value $\sigma_{SF,min} = 0 \mu\text{eV}$ (and thus the corresponding extremum $P_{c,min} = 0.88$) is unrealistic, as spectral fluctuations are expected at least due to hyperfine interaction. As regards the value $P_{c,max} = 1$, (and thus the implied extremum $\sigma_{SF,max} = 0.8 \mu\text{eV}$) it is also unrealistic in our device, due to the presence of co-tunneling, i.e. charge fluctuations. Overall, our best estimate is truly at $P_{c,best} = 0.94$ which, through the fits in Fig. 2, corresponds to the associated best estimate $\sigma_{SF,best} = 0.5 \mu\text{eV}$.

This value of $P_{c,best}$ also fits additional data discussed in Supplemental section 5. Such measurements, including two-photon coincidences measured as a function of delay and applied voltage, illustrate how the electron escape/capture times sharply depend on the voltage bias. They also show that, for the optimal applied voltage of -0.63 V, a good fit of all available data is obtained with respective escape and capture times of 4 ns and 250 ps, corresponding to $P_c=0.94$ and $P_{\uparrow}=P_{\downarrow}=P_c/2=0.47$."

We also rewrote the part discussing the uncertainty on P_{\uparrow} and its impact on the extrapolation process, see at L386:

"[...] The third panel shows the extrapolated intensities I_X^{\uparrow} , as deduced from the measured intensities I_X^{cav} and I_X^{avg} with Eq. (2), and from a self-consistent fit using $P_{\uparrow}=0.47$. Taking into account the extremal bounds on the parameter P_c , though such bounds are unrealistic, one can associate a maximal uncertainty of ± 0.03 to this best estimate of P_{\uparrow} : the impact of such uncertainty on the extrapolation process is discussed in Supplemental section 3."

Finally, we mention in the main text that this uncertainty on P_{\uparrow} is taken into account in the uncertainty with which we estimate our experimental fidelities, please see item 14 below.

"14. In fig 4(a), extracted data is presented to argue that a phase-shift of +/-pi/2 can be achieved, i.e. a D/A state can be attained. However, whilst 3d plots look nice, it is difficult looking at fig 4(a), particularly in the case of D, to see that the desired effect is achieved. In (b), a different perspective is shown which is clearer, but in a demonstration the focus is on the measured results, not the theory curves. Perhaps it would be good to reproduce the data of 3D plots in the supplement, using 2D plots of Stokes-vector components vs the control parameter (detuning?)."

We thank our reviewer for his suggestion: we added at L435 a link to the Supplemental section 7, where Fig. S10 does contain the 2D plots of Stokes vector components:

"[...] The results of the extrapolation processes are displayed in the Poincaré sphere, in Fig. 4a, for the three magnetic fields previously explored (see also the corresponding Stokes components in Supplemental section 7). [...]"

We also clarified with which experimental fidelities our target states were approached (this is also indicated in the abstract, see our response to item 1):

"[...] These three targets were experimentally approached with fidelities of $(97\pm 3)\%$, $(85\pm 10)\%$, and $(92\pm 5)\%$, respectively. In these estimations, the fidelity of a Stokes vector \vec{S}_{\uparrow} to an ideal target \vec{S}_{target} is computed through $F = \frac{1}{2}(1 + \vec{S}_{\uparrow} \cdot \vec{S}_{target})$, and the uncertainty takes into account the effect of the ± 0.03 uncertainty on our best estimate for P_{\uparrow} , used in Eq. (2). [...]"

"15. In fig. 4, it looks as if the polarization rotates about an axis lying in the AHAD plane, which is controlled via the B-field – perhaps by tuning the cavity/optical transition detuning, although this is unclear. It looks as if there is a limit to how close this axis can be tilted towards A or D, limiting access to the H-state, i.e. pi-shift. This kind of undermines the claim of pi-shift made in the "here.." paragraph, see 7."

This remark is strongly connected to items 10 and 11, and thus **we refer to our corresponding answers and the list of associated changes, in the main text and in the Supplemental Materials, including the new Fig. S11 showing all potential output states.** As discussed above the possibility to reach various

states is due to the possibility to change both the amplitude and phase of the reflection coefficients $r_{V \rightarrow H}^{\uparrow}$ and $r_{V \rightarrow V}^{\uparrow}$. Indeed, the ratio $r_{V \rightarrow H}^{\uparrow}/r_{V \rightarrow V}^{\uparrow}$ governs the projection of the state on the HV axis, while its phase governs its relative orientation around the HV axis. As also discussed in our response to item 11, the conditions for the mentioned states are easily written (L465):

“[...] polarisation state |A⟩ (i.e. $r_{V \rightarrow H}^{\uparrow} = r_{V \rightarrow V}^{\uparrow}$), |H⟩ (i.e. $r_{V \rightarrow V}^{\uparrow} = 0$), and |D⟩ (i.e. $r_{V \rightarrow H}^{\uparrow} = -r_{V \rightarrow V}^{\uparrow}$) [...].”

Interestingly, mathematically finding state |H⟩ is the easiest part, as it just consists in reaching $r_{V \rightarrow V}^{\uparrow} = 0$ through a destructive interference between the directly-reflected laser and the co-polarised resonance fluorescence (see also analytical formulae in our new Supplemental section 2). Therefore, claiming the possibility of a π rotation angle is almost trivial theoretically, the big physical difficulty being to reach a good polarization purity, i.e. a truly stable destructive interference, with no detrimental fluctuations.

“16. (Methods) Sufficient information to reproduce the experiments should be provided, possibly in the supplement. If the information can be found in refs, then a clear statement such as “Full details of the device, and experimental setup can be found in ref. “ should be made. In particular, quantitative information on the doping, and dimensions of the micropillar are absent. Likewise, the optics is sketchy. The polarization properties will be important here. What NPBS do you use? How do you calibrate/compensate for birefringence of setup? Is the light focused by a room temperature objective through a strained cryostat window, or is a cold objective used? How is the mode-matching achieved? What is the NA of the micropillar?”

Here also we thank the reviewer for the detailed suggestions, we realize how many important details were indeed missing. We now refer more adequately to the Methods section in our main text, and we completed the Methods section with the additional information. In particular:

- at L596, regarding the sample structure, doping, and dimensions:

“Full details on the device fabrication can be found in Refs [36, 37], though a different QD-pillar device is used in the present work. In particular, the sample [...].

[...] To electrically contact the structure, the bottom mirror (Si-doped) presents a gradual doping from $2 \cdot 10^{18} \text{cm}^{-3}$ down to $1 \cdot 10^{18} \text{cm}^{-3}$. This level of doping is maintained in the first half of the cavity region, and is stopped only 25 nm before the QD layer, which creates a tunnel barrier between the quantum dot and the Fermi sea. The top mirror is C-doped with increasing doping level, from zero to $2 \cdot 10^{19} \text{cm}^{-3}$ at the surface. [...].

[...] Each device is formed of a central micropillar (around 3 μm diameter) connected through four ridges to a large circular frame, [...].”

- at L634, regarding the optical coupling inside the cryostat, including the information on mode matching and numerical aperture:

“[...]. Light is focused and collected using a cold aspheric lens (4 mm focal length) within the cryostat. The diameter of the free space beam incoming on the aspheric lens is adjusted so that the numerical aperture of the focused beam matches that of the pillar device (N.A. ≈ 0.2). Using an optimisation and characterisation method detailed in Ref. [30], near-perfect mode matching can be approached, leading to $\approx 95\%$ efficiency for the coupling of incoming photons into the cavity. [...].”

- at L652, regarding the additional information about polarization measurement, compensation and calibration:

“[...]. In addition to the main optical elements sketched in Fig. 1c, additional motorized waveplates are used to compensate the polarisation rotations induced by the optical setup and the cryostat window, both for the incoming and reflected beams. The non-polarising beam splitter has a reflectivity around 90% for both polarisations, allowing to approximate its effect by a unitary polarisation rotation that is included in the optical compensation. In the overall, the polarisation tomography setup successively measures the average intensities I_X^{avg} in the six different polarisation bases $X = H, V, D, A, R$ and L , where all polarisations are defined at the entrance of the microcavity. [...]”

“17. Ref. [32] is in Phys. Rev. applied. Check the refs.”

This is now corrected, thanks.

“To summarize the report. The device is state of the art, and the raw data of fig. 2, 3(measured) clearly shows a giant polarization rotation conditioned on the QD spin. This is impressive. However, a paper needs to be measured against the claims made. According to the abstract, “We experimentally demonstrate rotation amplitudes such as $\pi/2$ and π in the Poincare sphere, as required for applications based on spin-polarisation mapping and spin-mediated photon-photon gates.” Ideally, such claims would be supported by direct measurements. By inspection, the “extrapolated data” does not pass through H, and thus claim of π rotation is simply not true. Probably it is not strictly true even in theory. To accept the validity of the claim of $\pi/2$ rotation, the reader needs to understand the data processing/manipulation, including underlying assumptions, and uniqueness of fit. A favourable endorsement will largely be decided by the strength of response to point 12.”

We are thankful to Reviewer 1, in addition to all his/her suggestions, for the positive assessment regarding our device and the fact that our raw data “clearly show giant polarisation rotations conditioned by a single QD spin”, which is “impressive”. We fully understand that several claims had to be clarified, and we believe that our revised sentence in the abstract (already mentioned in item 1) provides the best and most rigorous way to rephrase the main claim discussed here:

“[...] We experimentally approach polarisation states conditionally rotated by $\pi/2$, π , and $-\pi/2$ in the Poincaré sphere – desired for applications based on spin-polarisation mapping and spin-mediated gates – with extrapolated fidelities of $(97 \pm 3)\%$, $(85 \pm 10)\%$, and $(92 \pm 5)\%$, respectively. [...]”.

We hope also that our revised discussion of the fitting parameters and methods, completed with additional data (see in particular our response to items 12 and 13, and corresponding changes), will be viewed as an actually positive aspect of our work. We fully understand that this was a crucial point due to our extrapolation technique. In the revised version, in addition to the discussion and supplementary figure showing the effect of the uncertainty on P_{\uparrow} , we have also explicitly mentioned how this uncertainty impacts the extrapolated fidelities (see our response to item 14).

Finally, we would like to give a last insight regarding the statement “the claim of π rotation is simply not true” (if the data does not pass through H). This sentence is correct in the sense of an ideal π rotation not being achieved. We hope however that our answer to item 8 will have shown the importance of distinguishing between a limited fidelity and an incomplete rotation. This is all

the more important in our work which, contrary to previous ones, went through the process of carefully calibrating and measuring tomographic data along three measurement bases. Interestingly, the pioneering work with atoms, published in Nature 508, 241 (2014), used a different approach which allowed claiming a conditional pi phase shift in spite of numerous sources of imperfections. By measuring an interference pattern with and without an atom, this work could show that *“The phase of the reflected light is shifted by $(1.1 \pm 0.1)\pi$ relative to the case with no atom, and the visibilities of the oscillation with ϕ_V are $44\% \pm 2\%$ and $39 \pm 2\%$ [...]”*. The language is different, as no tomography is performed, but this sentence shows quite well that the problem lies not in the limited angle: it is related to the quality of the interference that can be achieved with the produced state. In the end, this is quite similar to our situation where reaching state H would have required a perfectly stable, destructive interference (see our response to item 14).

“Similar questions can be asked of the title. If the electron is not initialized, there is no control over the spin state so the title is not accurate, and needs to change.”

We fully understand this comment, also mentioned in the report of reviewer 3. The kind of control we achieve is the possibility to modify the conditional output polarization within a continuum of possible states, i.e. most of the Poincaré sphere. Yet, the previous title could also be interpreted as a claim for a spin-based optical gate, where a spin up or down is used as a binary switch to get one of two possible polarization outputs. We believe the interpretation will be clearer now that we have changed our title to:

“Controlling **the giant photon polarization rotations induced by** a single spin”

“Overall, the writing is imprecise, a greater care over the accuracy of statements, and more attention to communicating key points is needed. Whilst the dots are often there, the narrative does not join the dots, and construct full and effective arguments. For example, there is no clear explanation of what determines the angle or axis of polarization rotation, or why weak birefringence is desirable. The paper would benefit from a “brief history” of the device evolution, so the reader can understand where the current device fits. A quantitative benchmarking of rotation angles achieved in prior works would also be beneficial to place the current work in context. “

Thanks to all the useful suggestions we do believe that our rewriting of the main text (and Supplemental Materials) now allows “joining the dots” and “communicating key points”, with a great care taken regarding the “accuracy of statements”. Regarding the angle or axis of rotation, as well as the importance of birefringence, we think this is now much better addressed, as discussed in particular in our response to item 11. The quantitative benchmarking has also been added, please see below our response to item 1 or Reviewer 2’s report.

In summary, we are very thankful to Reviewer 1 for his/her careful reading and all his/her suggestions. We hope Reviewer 1 will share our view that our new version has been significantly improved and, in its strongly revised form, deserves publication in Nature Communications.

“Reviewer #2 (Remarks to the Author):”

“In this manuscript, E. Mehdi and coauthors employed a structure composed of a singly-charged semiconductor quantum dot embedded in a high-Q pillar-based microcavity, to show spin-selective rotations of the polarization of the reflected photons, done in a deterministic way and possible to operate in configurations with zero or low magnetic field. Building deterministic light-matter interfaces, in particular with solid-state spins as stationary qubits, is an important current topic in the realm of optical quantum information science and hence the focus of the manuscript is of relevance and importance in the field.”

“Before consideration for publication, there are multiple points/questions that need to be clarified:”

We thank Reviewer 2 for his/her positive assessment regarding the focus of our manuscript, and for all his/her numerous and useful suggestions. We believe that the revised manuscript is significantly improved, and that the points and questions of Reviewer 2 have indeed been clarified, as discussed below.

“1- It is crucial to properly present the exact distinction between the current work and the recently published work in the area. In particular, authors need to address the difference between their work and previous works such as [Androvitsaneas, Petros, et al. "Efficient quantum photonic phase shift in a low Q-factor regime." ACS Photonics 6.2 (2019): 429-435] and [Wells, L. M., et al. "Photon phase shift at the few-photon level and optical switching by a quantum dot in a microcavity." Physical Review Applied 11.6 (2019): 061001.] and [Antón, Carlos, et al. "Tomography of the optical polarization rotation induced by a single quantum dot in a cavity." Optica 4.11 (2017): 1326-1332.] which have also employed negatively charged QDs in micropillar cavities to realize spin-photon interfaces, in more detail.”

Indeed, this distinction with previous works was only partially discussed, though there are strong quantitative and qualitative differences.

First, we stress that the paper by Antón *et al.* did only mention neutral quantum dots, instead of negatively charged ones. The results were qualitatively different due to the absence of a spin, and thus the absence of *conditional* polarization rotation (the observed polarization rotation, up to 20° only, is an effect solely due to the quantum dot fine structure splitting). To make this distinction clearer, we added the following words at L108:

“We use polarisation tomography [39], this time applied to a charged quantum dot-microcavity device, to fully characterise the state of the reflected photons in the Poincaré sphere.”

and we also explicitly mentioned the use of a neutral quantum dot in this reference, at L205:

“In the presence of spectral fluctuations, a rotation of polarisation is still expected for the conditional Stokes vector \vec{S}_\uparrow , yet with a degraded polarisation purity, as in the experiment performed in Ref. [39] with a neutral quantum dot.”

Turning to the reported works with spin-based structures, we first highlight the fact that a crucial difference with them is *qualitative*, due to the fact that we use a complete tomography approach. **Regarding this aspect, we refer to our response to item 8 of Reviewer 1’s report, where various changes are mentioned with the goal of clarifying this originality of our work.** In addition,

regarding the technological performance of these spin-based devices, there were also a number of limitations in all previous works, which we start discussing in the introduction, at L81:

“[...] until now, the spin-induced polarisation rotations have remained limited in angle, due to optical losses [32] and/or detrimental spectral fluctuations [32, 34, 35]. As such, measurements of polarisation rotations above $\pi/2$ could only be obtained in a regime of strong post-selection [35] [...]”,

and at L101:

“[...] The Purcell enhancement provided by the high-Q cavity, and a strong reduction of the spectral fluctuations compared to previous works [32, 34, 35], allow reaching giant polarisation rotations. [...]”,

Finally, to better explain the various conditions to get large polarization shifts (also in response to item 3), we now detail them in the “Discussion” section, at L520:

“[...] the cavity-QED device presented here actually meets the four crucial conditions to be able to generate most polarisation states in the Poincaré sphere: (i) a large-enough output-coupling efficiency η_{top} , to allow the QD resonance fluorescence to strongly interfere with the directly-reflected light [32] (ii) a drastic reduction of spectral fluctuations, to limit the QD inhomogeneous broadening and thus preserve the purity of the output polarization states (iii) a cooperativity large-enough to broaden the QD-transition homogeneous linewidth significantly above the residual spectral fluctuations, and (iv) a moderate birefringence, allowing Purcell-enhanced emission in both polarisations. The condition of moderate birefringence, in particular, is essential to allow converting light polarisation, i.e. reach significant values for $r_{\text{H}\rightarrow\text{V}}$ and $r_{\text{V}\rightarrow\text{H}}$. Still, the other three conditions explain why giant and stable rotations had never been observed in low-birefringence devices, either due to output coupling efficiencies below 50% [32], limited cooperativities [34, 35], and/or significant spectral fluctuations [32,35].

To overcome the effect of slow spectral fluctuations, a post-selection approach has been introduced in Ref. [35], yet this technique selects a small portion of the experimental data, and would fail to compensate for fast hyperfine-induced fluctuations, in the MHz range [43]. [...]”

“2- How critical is the quality factor in achieving the targeted phase shift and what are the consequences of having low-Q cavities? For instance, in [Androvitsaneas, Petros, et al. "Efficient quantum photonic phase shift in a low Q-factor regime." ACS Photonics 6.2 (2019): 429-435] a low-quality cavity is used to achieve phase shifts as large as $2\pi/3$.”

“3- Please comment on the quantitative effect of the Purcell enhancement in the process, such as the achievable phase shifts and points on the Poincare sphere in general, purity of the reflected photonic states, and other relevant parameters.”

We answer simultaneously to both items 2 and 3, since the quality factor and the Purcell enhancement are related quantities. **In the revised manuscript, we now provide analytical formulae (see our response to item 11 of Reviewer 1’s report) which show how the conditional polarization rotations depend on the device cooperativity, itself proportional to the Purcell enhancement, and to the quality factor as well).** However, these rotations also depend on the device output coupling and birefringence, while their purity directly depends on the amount of

spectral fluctuations (inhomogeneous broadening) versus the QD homogeneous linewidth. **We refer to our response to the previous item 1, which gives a list of important conditions, including cooperativity i.e. Purcell enhancement, which is crucial to broaden the QD-transition homogeneous linewidth above the residual spectral fluctuations.**

Notably, this list of conditions illustrates why Androvitsaneas et al. could not achieve a $2\pi/3$ phase shift in absence of strong post-selection (see our response to item 1 above). In the low-Q cavity, there is no significant broadening of the QD homogeneous linewidth, which remains very sensitive to inhomogeneous broadening by spectral fluctuations. This is the reason why they had to post-select on the random moments where the homogeneous QD spectrum happened to resonate with the laser energy, and produce a large rotation/phase shift. This does not correspond to the idea of a stable, controlled polarization rotation with high purity, as is the final goal of the present paper. To better emphasize the importance of robustness with respect to spectral fluctuations, we also added in the Discussion section, at L551:

“Higher-purity polarisation states will also be obtained through increased Purcell enhancements, for both polarisations, to maximize the robustness of both reflection coefficients for $r_{V \rightarrow V}$ and $r_{V \rightarrow H}$, with respect to remaining fluctuations.”

Finally, as regards the achievable points in the Poincaré sphere, they are now better discussed thanks to the extended data provided in the Supplemental section 7: please see in particular the additional figure in our response to item 8 of Reviewer 1’s report.

“4- As mentioned in the “Principle of the experiments” section, a few T magnetic field is used (2T, 1.3T, 1.7 T in different presented data) to build the Zeeman splitting and selective interaction with the QD energy levels. So, the authors need to clarify the comment in the abstract regarding the possibility of using this configuration “with zero or low magnetic field”. Does this mean to suggest that the magnetic field is not required to induce spin-selective polarization rotations? Has the experiment been conducted for zero magnetic fields as well? There are some brief discussions in the supplementary document about this, which I suggest having further explained and possibly moved to the main text.”

We do confirm that a magnetic field is not required to induce spin-selective polarization rotations, which was the reason why $\pm 6^\circ$ polarization rotations could be observed in our previous work (with a sub-optimal device and only a single-axis measurement). We now make it more explicit in the revised introduction:

“[...] perfect spin-polarisation mapping can also be obtained at zero magnetic field, through opposite, $\pm \pi/2$ rotations in the Poincaré sphere for the states $|\Psi_\uparrow\rangle$ and $|\Psi_\downarrow\rangle$ [32].”

Also, we thank Reviewer 2 for the advice of moving discussions of spin-selective rotations to the main text, as well as discussions related to the operation at zero or low field. **We refer to our answer to item 11 of Reviewer 1’s report, which includes a number of related changes introduced in the main text. We also refer to the new Supplemental section 2** which is intended to clarify the behavior of the reflection coefficients, through their analytical formulae. In summary, regarding the possibility to work at zero magnetic field, it is directly related to the fact that the polarization-converting coefficients carry a spin-dependent sign such that, when $\omega_{QD}^\uparrow = \omega_{QD}^\downarrow$, $r_{V \rightarrow H}^\uparrow = -r_{V \rightarrow H}^\downarrow$. Though this equality would not be strictly maintained at different magnetic fields, the possibility to play with $r_{V \rightarrow H}^\uparrow$ and $r_{V \rightarrow H}^\downarrow$ (in addition to playing with $r_{V \rightarrow V}^\uparrow$ and $r_{V \rightarrow V}^\downarrow$) provides all the required degrees of freedom to find perfect spin-photon mapping at any magnetic field. In highly-birefringent devices, this would certainly not be the case, due to both $r_{V \rightarrow H}^\uparrow$ and $r_{V \rightarrow H}^\downarrow$ tending towards zero.

“5- In Fig. 2(a), what is the reason to get larger I^{avg}_V at the resonance point for the larger magnetic field compared to smaller values of B ? Also, please explain why the resonant behavior is observed at the detunings corresponding to $\omega_{\text{laser}} - \omega_{\text{cav}} = \omega_{\text{QD}}$ and not at $\omega_{\text{laser}} = \omega_{\text{QD}}$ ”

First, we apologize for the confusion regarding the resonant behavior, which does appear at $\omega_{\text{laser}} = \omega_{\text{QD}}$. We removed the confusing notation ω_{QD} in Fig. 2a, and instead completed the caption with the following sentence:

“[...] In each panel, the vertical dashed line highlights the resonance condition for which ω_{laser} equals ω_{QD} , the energy of the $|\uparrow\rangle - |\uparrow\downarrow\rangle$ transition, which is shifted by the applied magnetic field (see Fig. 1b) [...]”

Regarding I^{avg}_V , it gathers contributions from I^{cav}_X (directly-reflected laser on the empty cavity) and I^{\uparrow}_X (governed by the interference between the directly-reflected laser and the co-polarized QD emission). All these intensities can be computed from coefficients like $|\Gamma_{V \rightarrow V}|^2$, as is now introduced in the revised version (see in particular the new Equation 1 and Supplemental section 2). As also better explained in the revised version, the empty-cavity response is a Lorentzian function, which increases towards unity when the laser is tuned out of resonance with the cavity mode. This is exactly what happens when the magnetic field is increased: ω_{QD} and thus the relevant range of ω_{laser} are shifted out of resonance from the V cavity mode, and the contribution of the empty-cavity increases, increasing I_V . **Since this question regarding the V-polarised intensity impacts the device efficiency, we also refer to our response to item 2 of Reviewer 3’s report, and the corresponding changes at L556.**

“6- Here the damping rates of the cavity for H and V modes are in proximity with the separation of resonance frequencies for these modes. Could the authors comment on the dependence of the spin-selective Kerr rotations in this scheme on the ratio of the damping to the mode separation? Have numerical simulations been conducted as well?”

This question about what we call “moderate birefringence” was indeed an important point, that we clarified through a number of changes. **We refer to our response to item 11 of Reviewer 1’s report, and in particular all the changes in the main text, where the desirability of moderate birefringence is discussed. We also refer to the Supplemental sections 2 and also 6 (which specifically discusses the high-birefringence limit).**

We also performed many numerical simulations, some of them added to the revised Supplemental Materials, so that the reader can better visualize the achievable states, covering most of the Poincaré sphere (see in particular our response to item 8 of Reviewer 1’s report). However, we decided to leave the simulations as a function of birefringence for a future theory paper. Indeed, it requires a large amount of simulations to illustrate the various impacts of birefringence: not only the achievable states, but also the robustness of the reflection coefficients (and thus the purity of the states in presence of fluctuations), and the efficiency of the spin-photon interface (i.e. the probability to actually reflect the incoming photons). Actually, low/moderate birefringence is always desirable for the robustness and the efficiency of the present scheme, while the description of all the achievable states stays non-trivial even in limiting cases (such as zero birefringence, infinite cooperativity, perfect output coupling, etc...). Our revised manuscript thus focuses more

on the qualitative discussion, as seen with our response to item 11 of Reviewer 1's report. Our goal in this revised version is that the reader gets the big picture without focusing on the non-trivial Poincaré sphere representations.

"7- How many times in the measurement interval of 0.1 seconds for the intensities does the co-tunneling process for the electron in the QD happen? This strictly gauges the degree of validity for equation 1."

Indeed it was important to justify Eq. (1), which is now Eq. (2) of the revised version, together with the validity of the other critical equation $P_{\uparrow}=P_{\downarrow}=P_c/2$. All these equations are related to the fact that the electron escape time is only 4 ns, which means that there are more than 10^7 escape/capture processes during the measurement interval. **We refer to our discussion of items 12 and 13 from Reviewer 1's report, and to the related changes which were performed in the main text (in particular at L163, L318) and in the Supplemental Materials (especially sections 4 and 5) to introduce, clarify and justify these claims,**

Specifically, regarding the comparison with the detector time, we included the following sentence in the related paragraph, at L171 (just after introducing the co-tunneling and the 4ns electron escape time):

"[...] when integrating counts for 0.1s on the single photon detector, one measures the average intensities I_x^{avg} with contributions from the three possible ground states $|\uparrow\rangle$, $|\downarrow\rangle$ or $|\emptyset\rangle$, with respective probabilities P_{\uparrow} , P_{\downarrow} and P_{\emptyset} . [...]"

"8- How is the self-consistent fitting done to extract P_{\uparrow} or P_c in Fig 3? Is P_c merely a fitting parameter in numerical simulations or has this parameter been experimentally extracted?"

This is an important point which also raised questions and comments from Reviewer 1. In summary, fitting the intensities alone does only provide bounds on the potential values of P_c and P_{\uparrow} , leading to a claim that P_c is in the interval $[0.88; 1]$, and correspondingly P_{\uparrow} in the interval $[0.44-0.5]$. However, both bounds are unrealistic (as is now better explained in the manuscript) and additional measurements in the new Supplemental section 5 confirm that our best estimates ($P_c=0.94$ and thus $P_{\uparrow}=0.47$) correspond to the maximally-achieved (voltage-dependant) occupation probability, with fitted electron escape and capture times of 4ns and 250 ps, respectively. **Since this discussion of parameters has been entirely rewritten, we also refer here to our discussion of items 12 and 13 from Reviewer 1's report, (see in particular all changes performed in the main text, especially in the paragraphs starting at L295 and L306, and the strongly-revisited Supplemental Materials).**

"9- The underlying reason for having spin-selective phase shifts is not very clearly explained in the manuscript. There is a brief discussion before the "results" section in this regard; however, this needs to be explained much more clearly in the manuscript, especially in connection with the role of photon polarization (right-handed and left-handed as shown in Fig. 1) in the ground states to trion states population transfer and bringing the laser into resonance with the ω_{QD}^{\uparrow} ."

We fully agree and we thank Reviewer 2 for pointing out this difficulty that we had to solve. Indeed, the selection rules are based on circular polarisations, as reminded by Reviewer 2, while the cavity

birefringence and the input reference state (corresponding to zero rotation) are along H and V polarisations. In the end we solved the issue by converting all our theoretical analysis to the H/V polarization, while keeping sure that all formulae in the H/V basis did exactly fit the numerical results, as well as our former analysis in the circular basis (in Arnold et al, Nature Communications 6, 6236 (2015)).

Regarding all the changes made in the main text, and the new Supplemental section 2, we refer to our response to item 11 of Reviewer 1's report, where several changes are introduced which are related to this point (see in particular paragraphs starting at L180 and L499).

"10- In Fig. 4(b) the numerical result on the Poincare sphere for B=1.69T is very different from the results for B=1.7 T (Fig. 3(b) and Fig. 4(a)), why?"

The numerical results at 1.69 and 1.7T are actually extremely close, the value of 1.69T being chosen so that the output state passes more precisely through the desired state $|H\rangle$ (in absence of environmental noise). Yet it is indeed true that the results seem very different due to different perspectives, and we are thankful to Reviewer 2 for noticing it. To prevent confusion, we added a sentence to the figure caption:

"[...] Note that these simulations are viewed from the top of the Poincaré sphere, i.e. from a different perspective than in previous figures."

"11- The difference between the dashed and solid lines cannot be really seen in Fig. 4(b). I suggest modifying the plot to make the distinction more apparent."

We do agree and we modified Fig. 4b to invert solid and dashed lines, while also slightly thickening them for clarity.

Reviewer #3 (Remarks to the Author):

"The manuscript by E. Mehdi et al. presented a study of the manipulation of the polarisation of the reflected light from a cavity-QED device consisting of a single QD and a photonic cavity with electrical contacts. When the driving laser is in resonance with the QD transition, the polarisation of the reflected light changes. This change in polarisation was explained and modelled as an effect of the interference between the empty cavity reflection and the quantum resonance fluorescence. The modelling also gave an extrapolated polarisation rotation hypothetically induced by a pure spin-up QD state and the extrapolated Stokes vector (S-up) indeed shows a giant rotation at a certain laser frequency. Furthermore, the authors presented a similar polarisation rotation when the QD resonance shifts with the magnetic field and simulations of the impact of environmental noise on the reflected polarisation."

"The authors motivated their study with a fascinating scope of producing "a stable, controllable, spin-dependent photon state, in a deterministic way" and used "Controlling photon polarisation with a single spin" as the title of the manuscript. However, after reviewing the manuscript and the supplementary materials, the reviewer thinks some claims are not well supported by the results or the relevant discussions. It is not clear how this work would actually benefit the research of quantum gates between incoming photons and stationary qubits. Several specific questions are as follows:"

We are happy that Reviewer 3 shares our interest for the “fascinating scope” of producing stable, controllable, spin-dependent photon states, and we are thankful for his/her remarks. Indeed we realized that some claims had to be rewritten and clarified, while others were not entirely discussed. As a starting point, **we first refer to our response to Reviewers 1 and 2, and to the corresponding changes made in the revised version.** In particular, we hope the revised manuscript will help convincing Reviewer 3 that this work can benefit the research towards quantum gates in various ways:

- Technologically, our work demonstrates the potential of electrically-contacted, pillar-based interfaces, yet should also incentivize further implementations of low-birefringence interfaces.
- Experimentally, the tomography approach unambiguously describes the complete output state, and may thus serve as a reference for future characterizations of quantum interfaces.
- Theoretically, our general approach allows *adapting* the quantum gates and protocols, to reach high-fidelity operations in spite of technological and experimental constraints.

Below, we describe all the changes which have been specifically made in view of Reviewer 3’s suggestions and questions.

“1. In order to support the claim in the title, “Controlling photon polarisation with a single spin”, the ability to initialise the spin state is critical. However, spin initialisation is not presented or discussed here, even though polarisation rotations induced by a pure spin-up state was used to demonstrate the giant rotations.”

We fully understand this comment which is also raised by Reviewer 1. First of all, it was indeed important to clarify our title, changing it to:

“Controlling **the giant photon polarization rotations induced by** a single spin”

We believe such a revised title better fits the idea that the photon polarization state can be modified through giant (and experimentally-controllable) polarization rotations, which is the scope of our manuscript.

In addition, we hope our revised version better explains that spin initialization is not considered here (it is now stated already in the abstract and introduction) and that it is actually impossible to initialize the spin in our device. Indeed, as is now explained already in the “Principle of the experiments” section, the dominant co-tunneling processes erase any spin memory after a few nanoseconds, forcing it back to the average probabilities $P_{\uparrow}=P_{\downarrow}=P_c/2$. **In support to these statements, we refer to our discussion of items 12 and 13 in Reviewer 1’s report, including all the changes performed in the main text, and the revisited Supplemental Materials with added data showing co-tunneling.**

“2. Another important parameter is the efficiency of the spin-photon interface. An analysis of the efficiency would provide a more complete picture of the results. The intensities observed at different magnetic fields in Fig. 2a seem to differ, so it would be interesting to figure out whether the efficiency changes with cavity-laser detuning or with the magnetic field? Since the efficiency also affect the overall fidelity of a spin-photon interface, the fidelity should be clearly defined before presenting “a fidelity of 90.5 %” on line 439.”

We are thankful for this important comment, which raises a missing point of our previous version. The fidelity was only defined in the Methods section, while the efficiency was not discussed.

First, we now define the fidelity in the main text, in addition to displaying the experimental fidelity for all three magnetic fields (see at L468):

“[...] These three targets were experimentally approached with fidelities of $(97\pm 3)\%$, $(85\pm 10)\%$, and $(92\pm 5)\%$, respectively. In these estimations, the fidelity of a Stokes vector \vec{S}_\uparrow to an ideal target \vec{S}_{target} is computed through $F = \frac{1}{2}(1 + \vec{S}_\uparrow \cdot \vec{S}_{\text{target}})$, and the uncertainty takes into account the effect of the ± 0.03 uncertainty on our best estimate for P_\uparrow , used in Eq. (2). [...]”

We also added an entire paragraph in the “Discussion” section, at L556, to introduce the notion of efficiency and clarify its origin, dependence, and potential optimization:

“To develop truly optimal devices, one also needs to increase the efficiency of the spin-photon interface, i.e. the probability to successfully reflect the incoming photons, which is equivalent to the total normalized intensity $I_H + I_V$. The latter is limited, in particular, due to our imperfect $\eta_{\text{top}} = 0.635$, which allows photons to escape in other directions than through the top mirror. As seen from the values of I_H and I_V in Fig. 2a, the present device has an efficiency of around 35% at 1.7 T, and around 60% at 2.1 T, yet dropping down to around 15% at 1.3 T. This dependence is mainly due to the varying detunings $\omega_{\text{QD}}^\uparrow - \omega_{\text{cav,H}}$ and $\omega_{\text{QD}}^\uparrow - \omega_{\text{cav,V}}$, which govern $|r_{V\rightarrow H}^\uparrow|^2$ and $|r_{V\rightarrow V}^\uparrow|^2$, and thus I_H and I_V . In addition to improving η_{top} , optimizing the efficiency may also require finding an optimal value of the mode splitting $\omega_{\text{cav,H}} - \omega_{\text{cav,V}}$, to allow maximizing both I_H and I_V simultaneously.”

“3. Although a single photon detector was equipped, this experiment was not set up in a single photon regime and averaged intensity was measured. Is there a particular reason for this?”

4. The spin flip rates and lifetimes are key to understand the spin dynamics of the QD spin states. Currently, some rates were chosen to be drastically different for simulation purposes. This could be confusing for readers and a proper introduction to any existing knowledge about these rates would be useful.”

We respond to items 3 and 4 simultaneously since the actual reason for using single-photon detectors was to perform additional measurements, crucial to deduce the spin dynamics (actually governed by charge dynamics, i.e. co-tunneling), and now included in the new Supplemental section 5. This use of single-photon detection is now commented in the Methods section, at L665:

“[...] The intensity measurements are performed through single-photon detection, which is also useful to extract additional data such as photon coincidences and second-order intensity correlations, as in Supplemental section 5. [...]”

As regards item 4, we fully agree that the corresponding section was very confusing. As Reviewer 3 correctly understood, it is for simulation purposes that we have the habit of playing with drastically different rates in our numerical toolbox: this allows simulating, for example, a perfectly-initialized spin (to simulate I_X^\uparrow). But such details were unnecessary and we rewrote the corresponding Supplemental section 4. In particular, we modified the illustrating figure S3, to highlight the dominant (co-tunneling) processes, and we added a paragraph summarizing our knowledge about all these timescales (see page 8 of the Supplemental Materials):

“[...] in our device in the co-tunneling regime, the typical escape time τ_{esc} is at best of several nanoseconds, and the typical capture time of the order of 200 ps (see next Supplemental section). This is by far shorter than all the expected spin-flip times $\tau_{\text{SF},|i\rangle\rightarrow|f\rangle}$, for all initial and final spin states $|i\rangle$ and $|f\rangle$. Indeed, the hyperfine interaction is screened by the applied magnetic field [7]: the lifetime of an electron spin in similar devices would be expected to be well above the millisecond timescale at our high magnetic fields [8], and the expected lifetime of the hole spin is also well above the microsecond timescale, as soon as the applied field reaches a few tens of milliTeslas [9]. Therefore, our numerical simulations are dominated by co-tunneling processes [...]”

This co-tunneling regime is at the heart of many questions, already discussed in this Response Letter. **In particular, we also refer to our discussion of items 12 and 13 in Reviewer 1’s report, including all the changes performed in the main text, and the strongly-revisited Supplemental Materials with added data showing co-tunneling.**

REVIEWER COMMENTS

Reviewer #1 (Remarks to the Author):

1. (title) In the survey, 2/3 readers flagged the title “Controlling photon polarisation with a single spin” as making claims not supported by the evidence, since the spin state is random. The new title “Controlling the giant polarization rotations induced by a single spin” is at best ambiguous. A title should not require a footnote to be understood. I expect the current title will also result in communication errors.

In a quantum tech context, “control” could be interpreted as a short-hand for quantum control. The authors argue that the control is the tuning of the magnitude and axis of the polarization rotation imparted on the photon by the spin. However, one of these control knobs is the frequency of the laser, so that does not fly. The other is the magnetic field, but no experiments where the B-field is scanned are shown to demonstrate a fine level of control.

The important thing about the title is the message that will be received by the reader. Making contrived arguments to justify a title is missing the point.

“Electrical gating of giant polarization rotations induced by a single spin”, might work.

2. (point 12) Regarding the reader understanding the key points of the analysis.

I am looking at an 11 page paper, 18 page supplement, and a 23 page response. It reminds me of the Mark Twain quote “I do not have time to write a short letter, so I will write you a long letter”. The point is that good writing is concise, and takes longer to craft. The main text could probably fit into 4 pages. The discussion of the analysis is hard to follow because it is so long.

If I follow what is going on, there is an over-specification of the fitting problem, resulting in a non-unique fit. This results in a fairly large error in the inferred rotation angles, and is captured in the error bars of the fidelities. So I see this now as mostly a presentational issue,

with some implications for significance of the work. This is an era of high fidelity gates, and the interest is in errors that are small compared with the accuracy of the characterisation method presented.

3. On page 4 of response, the authors state in bold font. “Actually, this question highlights an originality of our work, unrelated to technological performance, and which should participate to the scientific impact of our manuscript. Unavoidably, the complete “3-axes” approach, where the output vector is fully measured, brings a qualitatively different perspective that requires a different way to discuss phenomena. To help understanding this point, we made a number of additional changes:”

In the abstract, fidelities for rotations about one axis are quoted, ie. σ_x , σ_z gates. What about σ_y gate?

Personally, I think (without doing detailed literature search) that the main result is the demonstration of large polarization rotation induced by single spin in an electrically contacted device. To the best of my knowledge prior reports of large polarization rotations in single electron dots have not been contacted. There is a technical challenge to be overcome in terms of doping degrading cavity Q-factor etc. The work shows that large rotations can be achieved with the diode structure and that there is a noise advantage that leads to large rotations being observed even in a time-averaged measurement.

Reviewer #2 (Remarks to the Author):

The authors have done extensive edits to some sections of the main text and the supplementary material. I believe both the manuscript and the supplementary material are in better shape now, however, there still exists statements that are not well supported or discussed and some mistakes and typos. The remaining points/issues are discussed below:

1- How is the claimed reduction of spectral fluctuations, compared to the previous work, experimentally quantified in this work? It is claimed that the reasons for getting higher polarization rotations, compared to previous works, is the reduction of loss as well as the reduction of spectral fluctuations in this work. However, it is not mentioned how these

parameters are exactly improved (techniques used) here, nor a proper quantitative comparison is made with the previously reported values.

2- I recommend removing I_x^{avg} from the label of Fig 3a, noting that only the left plot corresponds to the "avg" value, while the middle and right panels show " c_{av} " and " $|\text{up}\rangle$ " intensities

3- The view angle for the sphere in Fig. 3b panel two is not specified. I recommend either making it either completely plotted in DA-VH plane or maybe a 3D visualization. With this visualization, the twist of the trajectory from the VH-RL plane is not shown.

4- There is a mistake in the caption of Fig. 4b. The dashed line most probably corresponds to the simulation without the environment noise and the solid lines are the results with the noise. This can also be seen by noting that only the solid lines have some points with non-unity purity and the dashed lines always have almost unity purity.

5- On page 8 line 605, I believe there is a notation typo. The level of Si doping should be 2×10^{18} and 1×10^{18} , right? Same on line 610

6- On page 8 line 656 "The non-polarising beam splitter has a reflectivity around 90% for both polarisations, allowing to approximate its effect by a unitary polarisation rotation that is included in the optical compensation" needs to be clarified. This statement is very confusing. How does a non-polarizing beam splitter result in polarization rotation here? And how is the compensation exactly done?

7- On page 9 line 676, the last term in the square root function should be s_{RL}^2 and not s_{DA}^2 .

8- Zero magnetic fields may also be used in these spin-photon interfaces, as also stated in line 503. What was the reason for not trying zero magnetic fields experimentally and testing $|A\rangle$ state in the output for example for spin-up? Is the main purpose of the non-zero magnetic field here to screen the hyperfine interaction, such that the case of the zero

magnetic fields results in very noisy experimental results with a huge reduction in the purities and fidelities?

9- what does "exact numerical calculations" mean in this context? It has been referred to but not specified. Does it refer to the solutions to master equations (in the supplementary material) with or without semiclassical approximations? Please clarify.

10- The issue that I raised in the previous review regarding the selection rules is still confusing to the reader of the manuscript. The selection rules are for circularly polarized lights between the ground and excited trion states for the solid-state qubit considered, while Fig.1 clearly shows it for H/V polarization. Although some attempts have been made to clarify these in the supplementary material, these discrepancies and lack of proper explanation of the exact method used, especially in the main text, make it very hard for the reader to capture some of the main points of the work. There are certain important points/details, such as this, that need to be clarified especially within the main text (at least briefly) to avoid confusing the reader.

11- It is assumed that the electron spin is always either spin up or spin down, and this assumption carries out through all the extrapolations of the data. However, the electron can be in a superposition state. In this case, how can the weighted sum of the conditional intensities to get to the average intensity (equation 2) be justified? Qubits in superposition are extremely important in the implementation of quantum algorithms and computations and this point is not handled at all in the text. In short, what happens when we have spin in the superposition? and what is the reason for not considering it here?

12- In supplementary material page 2: where are the 2nd and 3rd equations of this page for the input and output field operator been used exactly and what is the precise connection between these equations and equations starting the next section of the supplementary material?

13- Supplementary material page 12, first paragraph: "One can then use a simple input state such as [...]" Why isn't it $|H\rangle$ and $|V\rangle$ input state and is instead $|D\rangle$? Please clarify.

As mentioned before, there has been a clear improvement from the previous version of the manuscript in terms of laying out the reasoning behind some statements and conveying the main points. Also, the experimental results are indeed important showing very large polarization rotations. However, the above points still need to be properly addressed.

Reviewer #3 (Remarks to the Author):

In the revised manuscript, substantial improvements have been implemented in response to the reviewers' feedback and comments. The authors have presented the claims and findings in a more precise manner, and they have provided a more meticulous and comprehensive explanation of the simulations and relevant processes. Although these revisions have addressed several concerns, the reviewer still believes that a crucial issue has not been adequately resolved.

Specifically, the authors listed three aspects of how this work can benefit the research towards (any kinds of) quantum gates, however, the original comment of Reviewer #3, was referring to “quantum gates between incoming photons and stationary qubits” as quoted from the manuscript. To clarify, the reviewer finds it challenging to envision the achievement of stationary qubits in the present device, without spin initialisation or even a discussion about future possibilities of spin initialisation in an improved structure. Consequently, it remains unclear how this work would actually benefit the research of quantum gates between incoming photons and stationary qubits.

The authors also stressed that spin initialisation is not considered here. However, the ultimate goals or applications they listed in the introduction still require a specific spin state or a spin qubit, and these also give the impression that the extrapolated giant photon polarisation rotation is primarily useful for an initialised or at least a long-lived spin state. As a result, the reviewer believes it's crucial to discuss the potential for spin initialisation and suggests explicitly acknowledging the absence of spin initialisation in the present work when making comparisons to other relevant studies.

Preliminary note

We thank all our reviewers for their numerous, renewed questions and comments, which we fully took into account to provide an improved and clarified version.

In addition to our specific answers and modifications, we **include an “Appendix” section at the end of this Response Letter**, providing a more detailed discussion of Refs 35 (Androvitsaneas *et al.*, ACS Photonics 2019) and 36 (Wells *et al.*, Physical Review Applied 2019). This Appendix is intended to help our reviewers in the comparison with our own work. The main points of this comparison will in any case appear, in a summarized version, in our answers below.

Reviewer #1

“1. (title) In the survey, 2/3 readers flagged the title “Controlling photon polarisation with a single spin” as making claims not supported by the evidence, since the spin state is random. The new title “Controlling the giant polarisation rotations induced by a single spin” is at best ambiguous. A title should not require a footnote to be understood. I expect the current title will also result in communication errors.

In a quantum tech context, “control” could be interpreted as a short-hand for quantum control. The authors argue that the control is the tuning of the magnitude and axis of the polarisation rotation imparted on the photon by the spin. However, one of these control knobs is the frequency of the laser, so that does not fly. The other is the magnetic field, but no experiments where the B-field is scanned are shown to demonstrate a fine level of control.

The important thing about the title is the message that will be received by the reader. Making contrived arguments to justify a title is missing the point.

“Electrical gating of giant polarisation rotations induced by a single spin”, might work.”“

We understand the point of the reviewer, which is the fact that “control” could still be interpreted, in spite of our precautions in the title, abstract, introduction, etc, as “quantum control”. We understand the need to remove any remaining ambiguity.

The title we now propose is: **“Giant optical polarisation rotations induced by a single quantum dot spin”**. Indeed, as will be discussed below in point 3, the previous works (Androvitsaneas *et al.*, ACS Photonics 2019 [35] and Wells *et al.*, Physical Review Applied 2019 [36]) have not provided demonstrations of true rotations of the Stokes vector. These works miss the required additional measurements, and deal with too large inhomogeneous broadenings (much larger than the lifetime-limited linewidth), that do not allow the coherent superposition of polarisation components (see also our response to point 3 and in the Appendix at the end of our Response Letter).

Also, we feel “Electrical gating” could evoke an electrical pulse sequence, which is not the case. “Giant polarisation rotations induced by a single spin in an electrically-contacted spin-photon interface” is accurate, but too long. Also, we consider electrical contact an enabling factor, but our work is not “technology-focused”: **our objective is also to provide proper definitions and standards to understand, measure, simulate, and discuss the physics of polarisation-encoded quantum nodes** (see also our responses below).

“2. (point 12) Regarding the reader understanding the key points of the analysis.

I am looking at an 11 page paper, 18 page supplement, and a 23 page response. It reminds me of the Mark Twain quote "I do not have time to write a short letter, so I will write you a long letter". The point is that good writing is concise, and takes longer to craft. The main text could probably fit into 4 pages. The discussion of the analysis is hard to follow because it is so long.

We are conscious of that fact: **we thus shortened and clarified our discussions so that the main text is shorter and, including the Methods section, finishes at page 8. We did so while, at the same time, ensuring to include all the insertions required by the present and previous reports from all reviewers.** We point out that around 45 points were raised in total, most of them divided in several sub-items, requiring around 100 insertions (an important fraction of whom were required in the main text).

Regarding the fact that the main text could "*probably fit into 4 pages*": we actually chose Nature Communications for this manuscript as we believed the Letter format was not appropriate to provide proper definitions and standards, and communicate all the associated physics. Concise writing was used by previous papers on the subject (Androvitsaneas *et al.* [35], Wells *et al.* [36]), yet with problematic definitions of "phase shift", inherent to the use of single-axis measurements (see also point 3 and Appendix).

In the end, it is clear that our 3-axis tomography approach takes time to introduce and discuss. This is especially true as, in contrast with the previous works, we went through the effort of not only measuring, but also simulating, and fitting, every presented curve. We did not find any previous reference addressing the physics involved in these discussions, neither experimentally nor theoretically.

We tried our best to communicate this physics, and we thank our reviewers for their efforts and suggestions that help us to do so. We actually believe their numerous questions showed the interest as well as the novelty of our approach, and the necessity to properly put it on paper.

If I follow what is going on, there is an over-specification of the fitting problem, resulting in a non-unique fit. This results in a fairly large error in the inferred rotation angles, and is captured in the error bars of the fidelities. So I see this now as mostly a presentational issue, with some implications for significance of the work. This is an era of high fidelity gates, and the interest is in errors that are small compared with the accuracy of the characterisation method presented."

We agree with Reviewer 1 regarding the uncertainties in the inferred polarisation states and fidelities. The additional uncertainty is a direct physical consequence of the observed co-tunneling in our device, limiting P_c to 94% instead of 100% (please also see our response to Reviewer 3 regarding the perspectives for suppressing this co-tunneling).

In the previous version we had chosen maximal confidence intervals for P_c , obtained by fitting the measured intensities alone (not considering additional data such as cross-correlations). However, what most researchers and official institutions actually use is the standard error. **In the revised version, we explicitly refer to the use of additional data to strengthen our estimation, and also explicitly choose the standard error.** This ensures our uncertainties can be fairly compared to that of other works. **This also allows shortening and clarifying our main text to solve the "presentational issue"** (the corresponding information being transferred to the Supplemental Materials, and clarified). In particular, our sentences at L297 now read:

“[...] As discussed in Supplemental sections 3 and 5, the best fit is obtained for $P_c=0.94 \pm 0.03$, together with $\sigma_{SF} = (0.5 \pm 0.2) \mu\text{eV}$, where the uncertainties correspond to the standard error. This set of parameters is the one that allows fitting all the measured data in Fig. 2, but also additional data, in the form of two-photon coincidences measured as a function of delay. Such measurements (see Supplemental section 5) show that, for the optimal applied voltage of -0.63 V, a good fit of all available data is obtained with respective escape and capture times of 4 ns and 250 ps, indeed corresponding to $P_c=0.94$.”

This also simplifies the discussion at L366:

“[...] The third panel shows the extrapolated intensities I_X^\uparrow , as deduced from the measured intensities I_X^{avg} and I_X^{cav} with Eq. (2), using $P_\uparrow = \frac{P_c}{2} = 0.47 \pm 0.015$. [...]”

The fidelities now read, at L437 (and also in the abstract):

“[...] These targets were experimentally approached with fidelities of $(97 \pm 1) \%$, $(84 \pm 7) \%$, and $(90 \pm 8) \%$, respectively (see Methods) [...]”

Finally, in the Methods section, the uncertainty estimation for fidelities is now explicit, and expressed in terms of standard errors:

“[...] The fidelity of a Stokes vector \vec{S} to an ideal target \vec{S}_{target} is computed through $F = \frac{1}{2} (1 + \vec{S} \cdot \vec{S}_{\text{target}})$. For each fidelity the displayed standard error is computed through uncertainty propagation, considering a $\pm 3 \%$ relative standard error on each measured intensity, as well as the estimated ± 0.015 standard error on P_\uparrow .”

We note that the 3% relative standard error on the four intensity measurements clearly provides the largest source of error on the estimated fidelities. For example, had we neglected the standard error on P_\uparrow one would still have obtained a standard error of $\pm 6 \%$ (instead of $\pm 7 \%$) on the fidelity to the H state.

(As a side note, we also partially agree with Reviewer 3 that this an “era of high-fidelity gates”. Yet this is less the case for gates produced using solid-state emitters, and, even more challenging, the gates to be developed using *receiving* nodes. These are the hardest since they require lifetime-limited linewidths and low-loss as well as high cavity-QED couplings.)

“3. On page 4 of response, the authors state in bold font. “Actually, this question highlights an originality of our work, unrelated to technological performance, and which should participate to the scientific impact of our manuscript. Unavoidably, the complete “3-axes” approach, where the output vector is fully measured, brings a qualitatively different perspective that requires a different way to discuss phenomena. To help understanding this point, we made a number of additional changes:” In the abstract, fidelities for rotations about one axis are quoted, ie. sigma_x, sigma_z gates. What about sigma_y gate?”

In our work we have not optimized sigma-y gates (i.e. producing states $|R\rangle$ or $|L\rangle$ starting from the input state $|V\rangle$). We nonetheless note that, for an appropriate value of the QD-laser detuning, the data acquired at 1.7T show the production of a reasonable fidelity of $(90 \pm 5)\%$ with respect to state $|R\rangle$. Yet this fidelity would be higher with an optimized value of the magnetic field, at 1.5T. This is best seen in Fig. S10 of the Supplemental materials, where we see that the Stokes coefficient

s_{RL} approaches unity at 1.5T. From this figure the predicted fidelity to state $|R\rangle$, at the appropriate detuning, is 98%. **This additional information is now added to the Supplemental Materials.**

As Reviewer 1 probably has already in mind, this question is indeed key for a fair comparison of our work to Wells *et al.* [35] and Androvitsaneas *et al* [36] (see also Appendix). Both works aimed at *reversing* polarisation from some input state $|P\rangle$ to the opposite target state $|\bar{P}\rangle$ (i.e. sigma-z gate). To produce any other state, performing a sigma-x and a sigma-y gate, a *controlled* and *coherent* superposition such as $\alpha|P\rangle + \beta|\bar{P}\rangle$ is required, with a *stable* relative phase. This stable phase is exactly what we demonstrate, in particular at 1.35T, with an extrapolated fidelity of 97% with respect to state $|A\rangle = (|H\rangle - |V\rangle)/\sqrt{2}$ (starting from the input state $|V\rangle$).

No stable phase can be produced, however, when the detuning between the laser and the emitter freely evolves due to inhomogeneous broadening (large spectral wandering): mostly *incoherent* superpositions can be obtained in that case. The intensity contrast in [35,36] is measured along the $P\bar{P}$ axis, which is well adapted to characterize the efficiency of the intended sigma-z gate, but is also, by construction, insensitive to these phase fluctuations between $|P\rangle$ and $|\bar{P}\rangle$. The intensity contrast measured along another axis, transverse to $P\bar{P}$, would be close to zero – due to phase blurring – even if their spin could have been initialized (actually in these previous works the spin was not initialized either, see also Appendix at the end of our Response Letter).

This distinction between actual rotations of the Stokes vector (in the sense of varying longitude or latitude), **compared to depolarisation** (Stokes vector stuck to the $P\bar{P}$ axis) **thus lies in the appearance of non-zero transverse components of the Stokes vector**, which we fully demonstrate. This led us to clarifications of our main text:

- At L86, we modified the introduction with the following sentences:

“[...] spin-induced rotations have remained limited in angle, due to optical losses [33] and inhomogeneous broadenings [33-36]. In particular, inhomogeneous broadenings much larger than the homogeneous linewidths lead the output states to fluctuate all around the Poincaré sphere, resulting in potentially strong depolarisations, though post-selection can be used to partially mitigate the impact of fluctuations [35].

Interestingly, in the variety of polarisation-based experiments [7,10,11,30,33-36], spin-induced polarisation rotations have mostly been measured via intensity contrasts in a single polarisation basis. This is equivalent to a single-axis projection in the Poincaré sphere, which can only give limited information. A typical difficulty is that a measurement axis can be well adapted to measure some fidelity, with respect to an ideal target [35,36], yet at the same time prevent from distinguishing between depolarisation effects and actual polarisation rotations.”

- At L212, the sentence commenting on the effect of spectral fluctuations is modified to better explain the impact of inhomogeneous broadening on depolarisation. Note that the 2016 reference by Androvitsaneas *et al.*, added as Ref. 34, does measure polarisation along a transverse axis and thus provides a true indication of rotation angle, yet limited to 6° due to the large inhomogeneous broadening (see also Appendix).

“[...] a pure state $|\Psi_T\rangle$ requires stable reflection coefficients, and thus a stable, lifetime-limited transition energy ω_{QD}^\dagger . Conversely, inhomogeneous broadenings [33-36] result in

the instability of the reflection coefficients, in phase and/or amplitude. This can lead to various degrees of depolarisation (as already observed in Ref. [40] with a neutral quantum dot) and potentially severe limitations regarding the averaged rotation angle [34]”

- Finally, at L497, the quantitative comparison of the inhomogeneous broadenings and homogeneous linewidths is given (also in Response to item 1 of Reviewer 2’s report):

“[...] giant and stable rotations had never been observed in low-birefringence devices, either due to output coupling efficiencies below 50% [33] and/or **inhomogeneous broadenings [33-36]**. As an example, for the two devices respectively used in Refs [34-35] and in Ref. [36], the inhomogeneous full-widths at half-maximum are of the order of 4.5 μeV and 10 μeV , respectively, to be compared with respective homogeneous linewidths of 0.8 μeV and 1.4 μeV . In our work, the inhomogeneous broadening remains limited since $\sigma_{\text{SF}} = 0.5 \mu\text{eV}$ thanks to the efficient evacuation of fluctuating carriers through the electrical contacts. In addition, the homogeneous linewidth is increased to 3.5 μeV : this is obtained thanks to an improved Purcell enhancement, due to a higher number of Bragg mirror pairs, yet at the cost of a sub-optimal output coupling efficiency $\eta_{\text{top}} = 0.635$ (as compared to $\eta_{\text{top}} > 0.9$ in Refs. [34-35] and [36]). We also note that in the previous references [33-36], as in the present work, the spin has not been initialised.”

“Personally, I think (without doing detailed literature search) that the main result is the demonstration of large polarisation rotation induced by single spin in an electrically contacted device. To the best of my knowledge prior reports of large polarisation rotations in single electron dots have not been contacted. There is a technical challenge to be overcome in terms of doping degrading cavity Q-factor etc. The work shows that large rotations can be achieved with the diode structure and that there is a noise advantage that leads to large rotations being observed even in a time-averaged measurement.”

We thank Reviewer 1 for his/her comments, which all helped improving our paper and clarify message and claims. We agree with this analysis, yet reminding that the “prior reports of large polarisation rotations” were not showing rotations (in the sense of a rotating angle of the Stokes vector, thus non-zero transverse components). We thus firmly believe that our work brings enough novelty to warrant publication in Nature Communications. We also refer our Reviewer 1 to the reports of Reviewers 2 and 3, and to the corresponding changes further improving our manuscript.

Reviewer #2

“The authors have done extensive edits to some sections of the main text and the supplementary material. I believe both the manuscript and the supplementary material are in better shape now, however, there still exists statements that are not well supported or discussed and some mistakes and typos. The remaining points/issues are discussed below:”

“1- How is the claimed reduction of spectral fluctuations, compared to the previous work, experimentally quantified in this work? It is claimed that the reasons for getting higher polarisation

rotations, compared to previous works, is the reduction of loss as well as the reduction of spectral fluctuations in this work. However, it is not mentioned how these parameters are exactly improved (techniques used) here, nor a proper quantitative comparison is made with the previously reported values.”

We apologize for our lack of clarity regarding this point. Indeed, reducing spectral fluctuations decreases the output polarisation state’s fluctuations, resulting in much less depolarisation, and larger rotations (see also our response to item 3 of Reviewer 1’s report, as well as the Appendix at the end of our Response Letter). This is quantified by the inhomogeneous broadening of the quantum dot linewidth, compared to the homogeneous linewidth. In our work, the inhomogeneous broadening is quantified by its standard deviation $\sigma_{\text{SF}} = 0.5 \mu\text{eV}$, which is much lower compared to previous works (Wells *et al.*, PR Applied 2019 [35], and Androvitsaneas *et al.*, ACS Photonics 2019 [36]). This is obtained thanks to the electrical contacting of the sample, allowing the evacuation of fluctuating carriers through the electrical contacts. In parallel, the reduction of optical losses in the cavity, compared to our previous work (Arnold *et al.*, Nat. Commun. 2015 [33]) was achieved by optimizing the number of Bragg mirror pairs, chosen to maximize the Purcell enhancement while improving the output coupling efficiency.

The following changes have been introduced to clarify these points, starting at L86:

“[...] spin-induced rotations have remained limited in angle, due to optical losses [33] and/or inhomogeneous broadenings [33-36]. In particular, inhomogeneous broadenings much larger than the homogeneous linewidths lead the output states to fluctuate all around the Poincaré sphere, resulting in potentially strong depolarisations, though post-selection can be used to partially mitigate the impact of fluctuations [35].

And at L497:

“[...] giant and stable rotations had never been observed in low-birefringence devices, either due to output coupling efficiencies below 50% [33] and/or inhomogeneous broadenings [33-36]. As an example, for the two devices respectively used in Refs [34-35] and in Ref. [36], the inhomogeneous full-widths at half-maximum are of the order of 4.5 μeV and 10 μeV , respectively, to be compared with respective homogeneous linewidths of 0.8 μeV and 1.4 μeV . In our work, the inhomogeneous broadening remains limited since $\sigma_{\text{SF}} = 0.5 \mu\text{eV}$ thanks to the efficient evacuation of fluctuating carriers through the electrical contacts. In addition, the homogeneous linewidth is increased to 3.5 μeV : this is obtained thanks to an improved Purcell enhancement, due to a higher number of Bragg mirror pairs, yet at the cost of a sub-optimal output coupling efficiency $\eta_{\text{top}} = 0.635$ (as compared to $\eta_{\text{top}} > 0.9$ in Refs. [34-35] and [36]). [...]”

“2- I recommend removing I_x^{avg} from the label of Fig 3a, noting that only the left plot corresponds to the “avg” value, while the middle and right panels show “cav” and “|up>” intensities”

This suggestion now appears in the Fig 3a.

“3- The view angle for the sphere in Fig. 3b panel two is not specified. I recommend either making it either completely plotted in DA-VH plane or maybe a 3D visualisation. With this visualisation, the twist of the trajectory from the VH-RL plane is not shown.”

Indeed the viewing angles chosen in Fig. 3b are both non-trivial, though other choices also lead to difficult visualisations. In our revised version we followed this very useful advice from Reviewer 2, of providing a 3D visualisation, in the form of a supplementary GIF file. For Fig. 3b this is now indicated in the revised caption:

“Extrapolated Stokes vector \vec{S}_\uparrow as a function of $\omega_{\text{laser}} - \omega_{\text{QD}}^\uparrow$ (see colorscale), viewed from two different angles in the Poincaré sphere (circles : extrapolated from experimental data ; lines : numerical simulations). See also supplementary GIF file for a complete 3D visualisation.”

In this revised version we now also provide a similar 3D visualisation for all our plots in the Poincaré spheres, accessible as supplementary files.

We chose GIF files as a quite universal way to share the 3D visualisation (our figures are plotted with Matlab and sharing the .fig file could limit the visualisation due to licence unavailability and/or incompatibility between Matlab software releases).

“4- There is a mistake in the caption of Fig. 4b. The dashed line most probably corresponds to the simulation without the environment noise and the solid lines are the results with the noise. This can also be seen by noting that only the solid lines have some points with non-unity purity and the dashed lines always have almost unity purity.”

This caption of Fig. 4b is now corrected.

“5- On page 8 line 605, I believe there is a notation typo. The level of Si doping should be 2×10^{18} and 1×10^{18} , right? Same on line 610”

The notation typo is now corrected.

“6- On page 8 line 656 “The non-polarising beam splitter has a reflectivity around 90% for both polarisations, allowing to approximate its effect by a unitary polarisation rotation that is included in the optical compensation” needs to be clarified. This statement is very confusing. How does a non-polarizing beam splitter result in polarisation rotation here? And how is the compensation exactly done?”

This interesting question is actually a nice illustration that polarisation rotations should also be discussed in terms of phase, in addition to intensity (we refer here to our response to the 3rd point of Reviewer 1). By definition, a non-polarising beam splitter (NPBS) ensures that the *intensity* is divided independently of the polarisation (in our case the NPBS reflectivity is of 90% for both eigenaxes of the NPBS). However, such NPBS still generally presents a polarisation-dependant phase, which induces potentially undesired transformations. This rotation can however be described by a simple unitary polarisation rotation (as the reflected intensity is the same for both polarisations). Note that mirrors as well as optical fibers also generate phase differences along the path, i.e. unitary polarisation rotations.

The concept of polarisation compensation then consists in using an additional pair of half-wave and quarter-wave plates, in order to counter-balance the global unitary transformation produced by all the optical elements. This additional pair of wave plates is calibrated simply by ensuring that an

initially H-polarised light is completely cross-polarised, when setting the polarisation tomography setup to measure the orthogonal polarisation (V).

The statement now reads at line 627 as the following:

“ [...] The non-polarising beam splitter has a reflectivity around 90% for both polarisations, ensuring that the intensity is divided independently of the polarisation. This does not exclude the possibility of a phase difference between polarisations, inducing a unitary polarisation rotation that is included in the optical compensation. [...]”

“7- On page 9 line 676, the last term in the square root function should be s_{RL}^2 and not s_{DA}^2 .”

The correction is now made in the main text.

“8- Zero magnetic fields may also be used in these spin-photon interfaces, as also stated in line 503. What was the reason for not trying zero magnetic fields experimentally and testing $|A\rangle$ state in the output for example for spin-up? Is the main purpose of the non-zero magnetic field here to screen the hyperfine interaction, such that the case of the zero magnetic fields results in very noisy experimental results with a huge reduction in the purities and fidelities?”

We thank Reviewer 2 for raising this interesting point. In this work, the main goal of using a high magnetic field is to separate in energy the quantum dot transitions so the laser only excites one transition at $\omega_{\text{QD}}^{\uparrow}$, while the other at $\omega_{\text{QD}}^{\downarrow}$ is too far detuned to yield any polarisation rotation. At zero magnetic field, where $\omega_{\text{QD}}^{\uparrow} = \omega_{\text{QD}}^{\downarrow}$, the experiments would not be specifically more noisy, but indeed we would not have been able to distinguish the two transitions, due to energy degeneracy, resulting in mainly depolarisation (absence of transverse components of the Stokes vector). This would also be the case at a low magnetic field (say 30 mT), which indeed allows screening the hyperfine interaction, yet would still lead to $\omega_{\text{QD}}^{\uparrow} \approx \omega_{\text{QD}}^{\downarrow}$.

However, a more trivial reason why our set of experiments did not include low magnetic fields is that the amplitude of polarisation rotation would have been very small in any case. Indeed, large polarisation rotations require *small* detunings between the $|\uparrow\rangle - |\uparrow\downarrow\rangle$ transition energy and the cavity modes energies, $\omega_{\text{QD}}^{\uparrow} - \omega_{\text{cav},V}$ and $\omega_{\text{QD}}^{\uparrow} - \omega_{\text{cav},H}$. In our device, the large magnetic fields allow controlling these detunings, by properly setting the value of $\omega_{\text{QD}}^{\uparrow}$ in between $\omega_{\text{cav},V}$ and $\omega_{\text{cav},H}$. Giant polarisation rotations cannot be reached at zero magnetic field with our device: the quantum dot is too far-detuned from the cavity modes, as shown with the reflectivity spectra of the studied device below.

In order to concisely mention this difficulty, we introduced the following changes at L470:

“Thus, when $\omega_{\text{QD}}^{\uparrow} = \omega_{\text{QD}}^{\downarrow}$ (at zero magnetic field), $r_{V \rightarrow H}^{\uparrow} = -r_{V \rightarrow H}^{\downarrow}$, while $r_{V \rightarrow V}^{\uparrow} = r_{V \rightarrow V}^{\downarrow}$. With our device, it was impossible to tune the QD transition energy at zero field so that $\omega_{\text{QD}}^{\uparrow} = \omega_{\text{QD}}^{\downarrow} = \omega_{\text{cav},V} + 14.1 \mu\text{eV}$. Had it been the case, the spin states $|\uparrow\rangle$ and $|\downarrow\rangle$ would almost have been mapped to the opposite polarisation states $|\Psi_{\uparrow}\rangle = |D\rangle$ and $|\Psi_{\downarrow}\rangle = |A\rangle$. [...]”

“9- what does “exact numerical calculations” mean in this context? It has been referred to but not specified. Does it refer to the solutions to master equations (in the supplementary material) with or without semiclassical approximations? Please clarify.”

Indeed, what we call “exact numerical simulations” refers to the numerical solving of the master equation, as presented in the supplementary materials. Such an approach is independent from the analytical one (for which we relied on the semiclassical approximation, see also our response to points 10 and 12).

We understand that a potential confusion could arise, since we first use such numerical simulations in section 2, to verify the analytical formulae obtained with the semiclassical approximation. It is only afterwards that we introduce the larger potential of numerical simulations (taking into account processes such as jumps between the two ground states or the two excited states, escape/capture mechanisms, and inhomogeneous broadening induced by spectral fluctuations), as described in section 4.

To avoid any confusion in the supplementary material, we clarified the end of the supplementary section 2:

“The above calculations are useful to understand the physics of spin-photon interfaces, thanks to the analytical description of the pure states $|\psi_{\uparrow/\downarrow/\emptyset}\rangle$, obtained in the linear low-power regime and in absence of fluctuations. We now introduce an independent approach based on numerical simulations performed with the Quantum Optics Toolbox [5]. This consists in numerically solving the master equation in the stationary regime, without any approximation, in order to compute [...]”

[...]

As a first verification, we use this approach to check that numerical predictions confirm the validity of the analytical formulae, as illustrated in Fig. S1. In the numerical simulations for this figure, most parameters are kept identical to those in the main text, except that $\sigma_{\text{SF}} = 0$ (no spectral fluctuations) and that the system is numerically forced to stay in the two-level subspace of states $|\uparrow\rangle$ and $|\uparrow\downarrow\uparrow\rangle$. This allows the absence of fluctuations, which, in addition to the choice of a low-power excitation (so that $P(|\uparrow\downarrow\uparrow\rangle) \ll 1$), ensures compatibility with the validity domain of the semiclassical approximation. The numerically-simulated intensities [...].

The numerical approach is actually more general, as it can provide an exact numerical solving of the master equation, independently of any approximation. We use it to compute time-dependent quantities, to take into account spectral fluctuations, and to fully describe co-tunneling processes: this will be described in Supplemental section 4. [...]"

A clarification was also added at the beginning of Supplemental section 4:

"We now describe how exact numerical simulations are performed in presence of fluctuations, independently of any approximation."

"10- The issue that I raised in the previous review regarding the selection rules is still confusing to the reader of the manuscript. The selection rules are for circularly polarized lights between the ground and excited trion states for the solid-state qubit considered, while Fig.1 clearly shows it for H/V polarisation. Although some attempts have been made to clarify these in the supplementary material, these discrepancies and lack of proper explanation of the exact method used, especially in the main text, make it very hard for the reader to capture some of the main points of the work. There are certain important points/details, such as this, that need to be clarified especially within the main text (at least briefly) to avoid confusing the reader."

Indeed, this point needed clarification. The circular right and left polarisations (R and L) are the proper eigenbases to describe the QD, as given by the selection rules. On the other hand, the proper eigenbases to describe the cavity correspond to the H and V polarisations, not only because we excite V , but also since the cavity leads to different Purcell enhancements for the H - and V -polarised components (that participate to polarisations R and L). This directly translates into the optical Bloch equations used to derive the polarisation states, as is now more clearly detailed in the Supplemental Materials (see also our response to point 12). Here we focus only on the changes made in the main text, to clarify the usage of H/V polarisations while the selection rules refer to R and L :

- At L145:

"[...] Furthermore, the transition energy $\omega_{\text{QD}}^{\uparrow}$ is varied, with the applied magnetic field, in the vicinity of the two cavity mode resonances $\omega_{\text{cav,H}}$ and $\omega_{\text{cav,V}}$ as displayed in Fig. 1b. This ensures that the $|\uparrow\rangle - |\uparrow\downarrow\uparrow\rangle$ transition, at $\omega_{\text{QD}}^{\uparrow}$, benefits from an efficient Purcell enhancement in both cavity eigenmodes, yet with generally different Purcell factors. These eigenmodes are defined as "horizontally" (H) and "vertically" (V) polarised"

- At L155:

"[...] The incoming polarisation, with a Stokes vector denoted \vec{S}_{in} , is adjusted to match one of the two cavity eigenaxes, defining state $|V\rangle$. This allows avoiding cavity-induced polarisation rotation [30] while exciting the desired transition, as $|V\rangle = i(|L\rangle - |R\rangle)/\sqrt{2}$

with R and L the right-handed and left-handed circular polarisations corresponding to the selection rules in Fig. 1a. [...]"

- At L192:

"[...] In this approximation, one can solve the optical Bloch equations for the cavity operators describing the two eigenmodes (H - and V -polarised), and for the QD operators describing the two electron-trion transitions (R - and L -polarised). [...]"

Please see also our response to point 12 and the corresponding changes in the Supplemental section 2, where more details can be found regarding the latter sentence.

"11- It is assumed that the electron spin is always either spin up or spin down, and this assumption carries out through all the extrapolations of the data. However, the electron can be in a superposition state. In this case, how can the weighted sum of the conditional intensities to get to the average intensity (equation 2) be justified? Qubits in superposition are extremely important in the implementation of quantum algorithms and computations and this point is not handled at all in the text. In short, what happens when we have spin in the superposition? and what is the reason for not considering it here?"

We thank Reviewer 2 for raising this important point, which was indeed missing. Working with a spin prepared in a superposed state will allow reaching an entangled state such as, ideally:

$$\frac{|\uparrow\rangle \otimes |\psi_{\uparrow}\rangle + |\downarrow\rangle \otimes |\psi_{\downarrow}\rangle}{\sqrt{2}}$$

Such state will be maximally entangled if the condition $\langle\psi_{\uparrow}|\psi_{\downarrow}\rangle = 0$ is reached. Having the spin in a superposed state will be one of the future objectives of this work. However, as discussed in the supplementary section 5, the electron leaves the quantum dot within 4ns due to the presence of co-tunneling mechanisms. This prevents the initialisation of the spin, but also limits the preparation of a superposed state, yielding a totally *incoherent* superposition (with populations $\rho_{\uparrow\uparrow} = \rho_{\downarrow\downarrow}$ and no coherent terms: $\rho_{\uparrow\downarrow} = \rho_{\downarrow\uparrow} = 0$)

- To introduce this missing notion, we first completed the introduction with an additional sentence, at L54:

"[...] A key objective is to produce a perfect spin-polarisation mapping : starting from a fixed incoming photon state, $|\psi_{in}\rangle$, and depending upon a spin state $|\uparrow\rangle$ or $|\downarrow\rangle$, an ideal device would deterministically produce states of orthogonal polarisations, namely $|\psi_{out}\rangle = |\psi_{\uparrow}\rangle$ or $|\psi_{\downarrow}\rangle$ with $\langle\psi_{\uparrow}|\psi_{\downarrow}\rangle = 0$. This in turn would allow producing maximally-entangled spin-photon states of the form $(|\uparrow\rangle \otimes |\psi_{\uparrow}\rangle + |\downarrow\rangle \otimes |\psi_{\downarrow}\rangle)/\sqrt{2}$, through the interaction between an incoming photon and a coherent spin superposition, such as $(|\uparrow\rangle + |\downarrow\rangle)/\sqrt{2}$."

- Regarding the weighted sum of conditional intensities, we now justify it at the end of Supplemental section 2:

"[...] We note that such equations are valid only for classical probability distributions, i.e. incoherent superpositions of the three possible states, as expected in presence of incoherent relaxation mechanisms (e.g. co-tunneling). In contrast, for a photon

interacting with a coherent spin superposition, the system would instead be described by an entangled spin-photon state, after the interaction.”

- We also completed the Discussions section, at L522, to discuss the perspectives for the generation of spin superpositions (the same paragraph is also written in response to modifications requested by Reviewer 3):

“[...] A more scalable approach will be to combine electrical control, as performed here, with techniques recently developed that allow drastic reductions of hyperfine-induced fluctuations [48]. To allow implementing spin initialisation, control and superposition, it will also be crucial to prevent co-tunneling, e.g. by widening the tunnel barrier between the QD layer and the n-doped Fermi sea. GaAs/AlGaAs quantum dots could also be used as the embedded stationary qubit, allowing to reach spin coherence times above 100 μ s through dynamical decoupling [49]. Such improvements will allow taking advantage of spin-induced polarisation rotation with the spin prepared in highly-coherent superposition states, in order, for instance, to deterministically generate entanglement between a spin and multiple photons [20]. [...]”

“12- In supplementary material page 2: where are the 2nd and 3rd equations of this page for the input and output field operator been used exactly and what is the precise connection between these equations and equations starting the next section of the supplementary material?”

Indeed we realise that our explanations in Supplemental section 2 were lacking important information (this is also related to item 10, where the separate role of cavity quantities and QD quantities could not be understood from the previous version).

To clarify the discussion, we first numbered the 2nd and 3rd equations of page 2 of the supplementary material. These equations give the definition of the input field operator when the input field is considered as coherent, and the relation between the input and output field operators in the input-output formalism.

To clarify the connection between these equations and the reflection coefficients, we completed the Supplemental section 2 to give all the calculation steps:

“[...] To derive analytical formulae, one can solve the optical Bloch equations in the stationary regime for the QD and cavity operators, namely $\langle \hat{a}_{H/V} \dot{} \rangle = 0$ and $\langle \hat{\sigma}_{R/L} \dot{} \rangle = 0$. Note that we choose different polarisation bases for the quantum dot operators and the cavity ones. The circular right and left polarisations (R and L) are indeed the proper basis to describe the QD, as they are given by the selection rules in the studied configuration. On the other hand, the H and V eigenmodes correspond to the proper basis to describe the cavity.

The optical Bloch equations are found by multiplying all terms of the master equation by $\hat{a}_{H/V}$ ($\hat{\sigma}_{R/L}$ respectively) on the left, and by taking the trace. One obtains:

$$\langle \hat{a}_H \dot{} \rangle = -i(\omega_{\text{cav},H} - \omega_{\text{laser}})\langle \hat{a}_H \rangle - \sqrt{\kappa_{\text{top},H/V}} \langle \hat{b}_{\text{in},H/V} \rangle - \frac{\kappa_{H/V}}{2} \langle \hat{a}_{H/V} \rangle - \frac{g}{\sqrt{2}} (\langle \hat{\sigma}_R \rangle + \langle \hat{\sigma}_L \rangle)$$

$$\langle \hat{a}_V \rangle = -i(\omega_{\text{cav},V} - \omega_{\text{laser}})\langle \hat{a}_V \rangle - \sqrt{\kappa_{\text{top},H/V}} \langle \hat{b}_{\text{in},H/V} \rangle - \frac{\kappa_{H/V}}{2} \langle \hat{a}_{H/V} \rangle - \frac{ig}{\sqrt{2}} (\langle \hat{\sigma}_L \rangle - \langle \hat{\sigma}_R \rangle)$$

together with:

$$\begin{aligned} \langle \dot{\hat{\sigma}}_{R/L} \rangle &= -i(\omega_{\text{QD}}^{\uparrow/\downarrow} - \omega_{\text{laser}})\langle \hat{\sigma}_{R/L} \rangle - \frac{\gamma_{\text{sp}}}{2} \langle \hat{\sigma}_{R/L} \rangle \\ &\quad - \frac{g}{\sqrt{2}} \left(\langle \hat{\sigma}_{R/L}^\dagger \hat{\sigma}_{R/L} \hat{a}_H \rangle - \langle \hat{\sigma}_{R/L} \hat{\sigma}_{R/L}^\dagger \hat{a}_H \rangle \pm i \langle \hat{\sigma}_{R/L}^\dagger \hat{\sigma}_{R/L} \hat{a}_V \rangle \mp i \langle \hat{\sigma}_{R/L} \hat{\sigma}_{R/L}^\dagger \hat{a}_V \rangle \right) \end{aligned}$$

The semiclassical approximation, in such a case, consists in replacing the average value of product operators, such as $\langle \hat{A}\hat{B} \rangle$, by the product of average values $\langle \hat{A} \rangle \langle \hat{B} \rangle$ [2]. For a spin perfectly initialised in the state $|\uparrow\rangle$, and in the low-power limit, $P(|\uparrow\rangle) = \langle \hat{\sigma}_{R/L} \hat{\sigma}_{R/L}^\dagger \rangle = 1$, all other state populations being zero. Simplified equations are then obtained:

$$\begin{aligned} \langle \dot{\hat{\sigma}}_R \rangle_\uparrow &= -i(\omega_{\text{QD}}^\uparrow - \omega_{\text{laser}})\langle \hat{\sigma}_R \rangle_\uparrow - \frac{\gamma_{\text{sp}}}{2} \langle \hat{\sigma}_R \rangle_\uparrow + \frac{g}{\sqrt{2}} (\langle \hat{a}_H \rangle + i \langle \hat{a}_V \rangle) \\ \langle \dot{\hat{\sigma}}_L \rangle_\uparrow &= -i(\omega_{\text{QD}}^\uparrow - \omega_{\text{laser}})\langle \hat{\sigma}_L \rangle_\uparrow - \frac{\gamma_{\text{sp}}}{2} \langle \hat{\sigma}_L \rangle_\uparrow \end{aligned}$$

Conversely, if the spin is initialized in state $|\downarrow\rangle$:

$$\begin{aligned} \langle \dot{\hat{\sigma}}_R \rangle_\downarrow &= -i(\omega_{\text{QD}}^\uparrow - \omega_{\text{laser}})\langle \hat{\sigma}_R \rangle_\downarrow - \frac{\gamma_{\text{sp}}}{2} \langle \hat{\sigma}_R \rangle_\downarrow \\ \langle \dot{\hat{\sigma}}_L \rangle_\downarrow &= -i(\omega_{\text{QD}}^\uparrow - \omega_{\text{laser}})\langle \hat{\sigma}_L \rangle_\downarrow - \frac{\gamma_{\text{sp}}}{2} \langle \hat{\sigma}_L \rangle_\downarrow + \frac{g}{\sqrt{2}} (\langle \hat{a}_H \rangle - i \langle \hat{a}_V \rangle) \end{aligned}$$

Solving the above equations in the stationary regime, one can deduce the average values $\langle \hat{a}_H \rangle_{\uparrow/\downarrow}$. Finally, one can derive the relations between the input and output fields [4], using their definition given by (2) and (3). One obtains:

$$\begin{aligned} b_{\text{out},H}^{\uparrow/\downarrow} &= r_{H \rightarrow H}^{\uparrow/\downarrow} b_{\text{in},H} + r_{V \rightarrow H}^{\uparrow/\downarrow} b_{\text{in},V} \\ b_{\text{out},V}^{\uparrow/\downarrow} &= r_{H \rightarrow V}^{\uparrow/\downarrow} b_{\text{in},H} + r_{V \rightarrow V}^{\uparrow/\downarrow} b_{\text{in},V}. \quad [\dots] \quad " \end{aligned}$$

"13- Supplementary material page 12, first paragraph: "One can then use a simple input state such as [...]" Why isn't it $|H\rangle$ and $|V\rangle$ input state and is instead $|D\rangle$? Please clarify."

The section 6 of the supplementary material focuses on a spin-photon interface in the limit of highly-birefringent cavity. In this case, exciting one of the cavity eigenaxes (H or V) doesn't lead to polarisation rotation and only an input polarisation state corresponding to a superposition of $|H\rangle$ and $|V\rangle$ should be considered.

- To clarify this, we first corrected the typo on the value of $r_{H \rightarrow V} = 0$ in the supplementary material. The zero values for both $r_{H \rightarrow V}$ and $r_{V \rightarrow H}$, in the high-birefringence limit, are indeed the reason for the absence of polarisation rotation if a cavity eigenaxis is used. We also clarified our sentence:

“In such a case, it is impossible to obtain any polarisation rotation by exciting along one of the cavity eigenaxes: **exciting the device with V-polarised (resp. H-polarised) light only provides V-polarised (resp. H-polarised) output.** [...]”

- In addition, we improved the ensuing discussion in our Supplemental Materials. This implied, first, giving the complete definitions of $|\psi_{\uparrow}\rangle$ and $|\psi_{\downarrow}\rangle$ (including denominators which are relevant to the discussion):

$$|\psi_{\uparrow}\rangle = \frac{\alpha |H\rangle + \beta r_{V \rightarrow V}^{\uparrow} |V\rangle}{\sqrt{|\alpha|^2 + |\beta r_{V \rightarrow V}^{\uparrow}|^2}} \quad \text{and} \quad |\psi_{\downarrow}\rangle = \frac{\alpha |H\rangle + \beta r_{V \rightarrow V}^{\downarrow} |V\rangle}{\sqrt{|\alpha|^2 + |\beta r_{V \rightarrow V}^{\downarrow}|^2}}$$

$$\langle \psi_{\uparrow} | \psi_{\downarrow} \rangle = \frac{|\alpha|^2 + (1 - |\alpha|^2)(r_{V \rightarrow V}^{\uparrow})^* r_{V \rightarrow V}^{\downarrow}}{\sqrt{|\alpha|^2 + |\beta r_{V \rightarrow V}^{\uparrow}|^2} \sqrt{|\alpha|^2 + |\beta r_{V \rightarrow V}^{\downarrow}|^2}}$$

The following changes were then also introduced:

“We note, in addition, that smaller values of α lead to smaller probabilities for a photon to be reflected when the spin is in the $|\uparrow\rangle$ state. The optimal configuration is thus to work close to the threshold α_{opt} , i.e. the highest value of α allowing perfect mapping:

$$|\alpha_{\text{opt}}| = \sqrt{\frac{(1+C_V-2\eta_{\text{top},V})(2\eta_{\text{top},V}-1)}{2\eta_{\text{top},V}(2+C_V-2\eta_{\text{top},V})}},$$

which is generally bounded by $0 \leq |\alpha_{\text{opt}}| < 1/\sqrt{2}$. Working close to this threshold allows maximizing the denominator of the scalar product $\langle \psi_{\uparrow} | \psi_{\downarrow} \rangle$, hence the probability for a photon to be actually reflected (either in state $|\psi_{\uparrow}\rangle$ or $|\psi_{\downarrow}\rangle$)”

- Finally, regarding the limit case of a perfect device (which corresponds to the Duan-Kimble protocol):

“[...], allowing $|\alpha_{\text{opt}}|$ to reach $1/\sqrt{2}$. This is only in this specific limit that one should use a simple input state such as $|D\rangle = (|H\rangle + |V\rangle)/\sqrt{2}$, to obtain perfect spin photon mapping at resonance with $|\psi_{\uparrow}\rangle = |D\rangle$ and $|\psi_{\downarrow}\rangle = |A\rangle$, as in the pioneering Duan-Kimble protocol [15] [...]”

“As mentioned before, there has been a clear improvement from the previous version of the manuscript in terms of laying out the reasoning behind some statements and conveying the main points. Also, the experimental results are indeed important showing very large polarisation rotations. However, the above points still need to be properly addressed.”

We thank Reviewer 2 for all the time and effort, and his/her detailed suggestions, which strongly helped to further improve the manuscript. For further information we also refer to our response to Reviewers 1 and 3, and to the Appendix at the end of our Response Letter. We believe the manuscript is now in good shape to justify publication in Nature Communications.

“Reviewer #3

“In the revised manuscript, substantial improvements have been implemented in response to the reviewers' feedback and comments. The authors have presented the claims and findings in a more precise manner, and they have provided a more meticulous and comprehensive explanation of the simulations and relevant processes. Although these revisions have addressed several concerns, the reviewer still believes that a crucial issue has not been adequately resolved.

Specifically, the authors listed three aspects of how this work can benefit the research towards (any kinds of) quantum gates, however, the original comment of Reviewer #3, was referring to “quantum gates between incoming photons and stationary qubits” as quoted from the manuscript. To clarify, the reviewer finds it challenging to envision the achievement of stationary qubits in the present device, without spin initialisation or even a discussion about future possibilities of spin initialisation in an improved structure. Consequently, it remains unclear how this work would actually benefit the research of quantum gates between incoming photons and stationary qubits.”

The authors also stressed that spin initialisation is not considered here. However, the ultimate goals or applications they listed in the introduction still require a specific spin state or a spin qubit, and these also give the impression that the extrapolated giant photon polarisation rotation is primarily useful for an initialised or at least a long-lived spin state. As a result, the reviewer believes it's crucial to discuss the potential for spin initialisation and suggests explicitly acknowledging the absence of spin initialisation in the present work when making comparisons to other relevant studies.”

We fully agree, and we are sorry that we did not include this aspect in our first version of the manuscript. Due to fast co-tunneling (4ns escape time), spin initialisation happened to be unfeasible with our device. Yet, such fast co-tunneling was the direct consequence of an initial choice: a 25nm undoped barrier, between the n-doped Fermi sea and the QD layer. This 25nm distance is known to be adapted to standard InAs quantum dots, which are only few nanometers high. But our work shows it is not adapted in our case: our QDs being annealed at 850°C, gallium and indium interdiffuse and lead to both larger QD heights and lower energy confinement for the QD states. Now that this has been identified, we are starting the preparation of devices with suppressed co-tunneling. As soon as slower capture/escape rates will be measured, a range of applications will become possible, including spin initialisation but also, of course, spin-photon entanglement (this is also related to point 11 of Reviewer 2's report).

As regards the comparison with Wells *et al.* (Phys Rev. Applied 2019) [35] and Androvitsaneas *et al.* (ACS Photonics 2019) [36] we realise that we also forgot to mention that they do not show spin initialisation either. As discussed in more detail in the Appendix at the end of this Response Letter, in Ref. [36] it is only the post-selection which allows selecting time bins for which the spin was in the proper state (in addition to selecting time bins for which the QD happened to be on resonance with the incoming laser). Also, in Ref. [35], the fact that a “control pulse” is used should not be read as quantum control, nor spin initialisation: the control pulse is used to turn *off* the interaction with the quantum dot (just as we use voltage control to remove the charge from our quantum dot).

To include the discussion about spin initialisation and the future technological perspectives (not only for spin initialisation but also for spin coherence and the possibilities that this will offer), we modified the Discussions section, starting at L512:

“[...] We also note that in the references [33-36], as in the present work, the spin has not been initialised.

To mitigate the effects of slow spectral fluctuations and non-initialised spin, a post-selection approach has been introduced in Ref. [35], yet such technique cannot compensate for fast hyperfine-induced fluctuations [44]. A more scalable approach will be to combine electrical control, as performed here, with techniques recently developed that allow drastic reductions of hyperfine-induced fluctuations [48]. To allow implementing spin initialisation and control, it will also be crucial to prevent co-tunneling, e.g. by widening the tunnel barrier between the QD layer and the n-doped Fermi sea. GaAs/AlGaAs quantum dots could also be used as the embedded stationary qubit, allowing to reach spin coherence times above 100 μs through dynamical decoupling [49]. Such improvements will allow taking advantage of spin-induced polarisation rotation with the spin prepared in highly-coherent superposition states, in order, for instance, to deterministically generate entanglement between a spin and multiple photons [25]. [...]"

We thank Reviewer 3 for his/her suggestions which, together with the comments and questions of Reviewers 1 and 2, helped us to strongly improve and clarify our manuscript. Further information can also be found in the Appendix below.

Appendix

We provide here an Appendix where we discuss specificities of the previous references [35,36] to help better contrasting them to our work.

The first part of this analysis has already been mentioned in our response to Reviewer 1: we remind that what is called a "phase shift angle", in [35,36] is the arccosine of a contrast between two intensity points, along a specific polarisation $P\bar{P}$ (with $|P\rangle$ the input state and $|\bar{P}\rangle$ the opposite, target state). Measuring along such a polarisation axis is indeed well adapted to characterize the fidelity to a sigma-z gate (*reversing* polarisation from $|P\rangle$ to $|\bar{P}\rangle$): ideally, the measured contrast $(I_P - I_{\bar{P}})/(I_P + I_{\bar{P}})$ would go from +1 to -1, in a spin-dependent way. This would be the signature of a perfect sigma-z gate, i.e. a perfect " π phase shift". In presence of non-unity contrasts, however, measuring only $(I_P - I_{\bar{P}})/(I_P + I_{\bar{P}})$ does not allow distinguishing between limited rotation angles and depolarisation processes.

To demonstrate rotations, i.e. changes of orientation of the Stokes vector in latitude and/or longitude, one would have to demonstrate non-zero *transverse* components with respect to the $P\bar{P}$ axis. Such transverse components, however, can only appear if coherent superpositions are obtained, such as $\alpha|P\rangle + \beta|\bar{P}\rangle$. We do demonstrate such superpositions with our target states $|D\rangle$ and $|A\rangle$, starting from input state $|V\rangle$. Yet, with large inhomogeneous broadenings ($\sim 10 \mu\text{eV}$ in Ref. [36], and $\sim 4.5 \mu\text{eV}$ in Ref. [35], compared to $\sim 0.8 \mu\text{eV}$ and $\sim 1.4 \mu\text{eV}$ homogeneous linewidths), one predicts mostly *depolarisation*, i.e. a trajectory of the Stokes vector going in a straight line through the center of the sphere, remaining along the $P\bar{P}$ axis. Due to the quasi-absence of transverse components, the amount of "true rotation" (i.e. "change of orientation") can only remain very small.

In addition to these premises, there are differences between the previous references [35,36]:

- An interesting specificity of Ref. [36], by Wells *et al*, is the use of a control pulse. Yet, this presence of a control pulse should not be interpreted as spin initialisation. In absence of control pulse, there is indeed a 50% occupation probability for the desired spin state, similarly to the situation in our work. The control pulse is used in [36] to turn *off* the interaction with the QD, by making this 50% occupation probability decrease (ideally towards zero), instead of

increasing it towards 100%. Such control pulse therefore plays a role similar to our voltage control, which we used to *remove* the charge, and thus the spin, from the quantum dot.

In this context, Wells et al. use the state $|R\rangle$ as input state, which corresponds to $P\bar{P} = RL$. In absence of control pulse, their measured contrast in the RL basis is around 0.1 (this is extracted from Fig. 3b, taking into account the different scales on the left and right axes). This contrast can be translated to the angle $\arccos(0.1) \approx 85^\circ$, but it is also compatible with an almost complete depolarisation (due to large inhomogeneous broadenings, together with unpolarised spin). In absence of additional measurements (along HV and/or DA polarisation axes), it is impossible to make a distinction between rotation and depolarisation, and it is also impossible to estimate how much rotation is induced, not by the quantum dot spin, but by the birefringent cavity alone. Indeed, in presence of a control pulse (turning off the interaction), the measured contrast in the RL basis (Fig. 3c) is around 0.55, which is associated to a large angle: $\arccos(0.55) \approx 55^\circ$. There is thus a strong possibility that a large cavity-induced rotation, solely due to birefringence (if the device is not exactly excited along a cavity eigenaxis) participates to this 55° angle. This in turn leads to a large uncertainty on the amplitude of the true QD-induced effects. This is inherent to single-axis measurements which do not allow separating rotation from depolarisation.

- As regards Androvitsaneas *et al.* [35], it also has an interesting specificity: the use of an innovative post-selection technique to address the problem of large spectral wandering (and, at the same time, the fact that the spin is not initialised). The input state being $|V\rangle$ (and therefore $P\bar{P} = VH$), Androvitsaneas *et al.* used two photodiodes to monitor intensities I_V and I_H and select, among many time bins, the small fraction of those which lead to the smallest values of I_V and highest values of I_H . Such values occur whenever the wandering QD transition happens to be precisely in resonance with the laser (and when the non-initialised spin happens to be in the desired state).

In addition to this post-selection, two additional photodiodes, measuring the reflected light in the same polarisation basis (HV), are used to deduce a phase shift defined as $\arccos[(I_V - I_H) / (I_V + I_H)]$. Such angle, by construction, can be large when I_V is post-selected to be small, and I_H post-selected to be large. The best post-selected contrast obtained is around -0.5, which does indicate a temporary change of orientation of the instantaneous Stokes vector. Yet, this -0.5 post-selected contrast, instead of -1, is probably a signature of a remaining 50% depolarisation, due to remaining inhomogeneous broadening, even after post-selection is applied. Indeed, the transverse components (along DA or RL axes) are expected to be zero in these experimental conditions (where the QD is, in average, in resonance with the laser, but still varies from blue-detuned to red detuned due to remaining inhomogeneous broadening).

To distinguish depolarisation from true rotation, additional measurements would have been needed, for example in the DA basis. Such measurements in the DA basis have indeed been reported by the same team using the same device, without post-selection (Androvitsaneas et al, PRB 2016, now Ref. [34]). In this former paper, the same authors had considered a different definition of the phase shift: $\arcsin[(I_D - I_A) / 2\sqrt{I_H I_V}]$. According to such definition, a maximal phase shift of 6° was demonstrated with this same device, limited by the inhomogeneous broadening. This angle being measured using a transverse axis does give information about the rotation in longitude, and thus can be effectively compared with the rotation amplitudes reported in our work.

REVIEWERS' COMMENTS

Reviewer #1 (Remarks to the Author):

The conditional polarization rotation induced by a single quantum dot spin embedded in an electrically contacted micropillar is characterized. The electrical contacts stabilize the charge environment suppressing the fluctuations in the exciton optical transition energy by a factor of 9 compared to prior art [34-36], and is close to the limit due to fluctuations caused by nuclear spin environment. More importantly, the inhomogeneous broadening is reduced to be far below the homogeneous broadening resulting in a stable device. This is evidenced by a large rotation that can be observed in a time-ensemble average measurement, without the need for post-selection methods to mitigate against low frequency fluctuations. This is significant because a stable device is needed for scalability.

Hence I recommend publication.

However, I do so through gritted teeth. The first two versions of the manuscript were unpublishable due to title/claims unsupported by data. This was a total waste of everyones time. I get a sense that the abstract rides a bit too close to the line, and advice is not being listened to, and this is to the detriment of the reader. I note that the discussion crucial to understanding the significance of the work is on p7, 2nd col. It follows a long winded discussion of a measurement method the authors have already reported on in refs. [31,40]. The reader needs to perform a feat of endurance to reach the key take home messages. The paragraph is unclear on the difference in device design that leads to improved stability, and provides no quantitative information on the polarization rotations achieved in prior art to provide clear context for the resulting improvement in device performance.

Reviewer #2 (Remarks to the Author):

The authors have put a great effort into revising the manuscript for the second time, including changing the title of the manuscript, which should have been done, as emphasized by reviewers during the previous round of the review. The manuscript and supplementary

material are in a much better shape from different aspects, including the added clarification and explanations to multiple sections of both the main text and the supplementary material. Also, adding the GIF files certainly improved the quality of the data presentation and visualization of the data by the readers. The extra appendix section that the authors added to the end of their response also was helpful in presenting a comparison with previous works done on the field and emphasizing their new contributions, especially regarding the analysis of the rotations on the full Poincare sphere. The authors also clarified the circumstances regarding spin initialization and the technical difficulties and their future efforts toward reducing co-tunneling effects and possible spin initializations for quantum spin-photon interface applications. I want to emphasize again the point that was also pointed out by the first reviewer during the previous round of review regarding mentioning stationary qubits in the text. From the authors' response to the review report, it is clear that they understand and agree that there is no stationary matter or spin qubit in their experiment as there is a fast co-tunneling process happening without any spin initialization in between tunneling events. However, the abstract and introduction still, to some extent, advertise the immediate applicability of their result for spin-photon entanglement and having matter qubits interacting with photons. The fact that this is not an immediate result of this work becomes clear after reading the rest of the manuscript, and some parts of the supplementary material. I think it would have been useful to have the abstract and introduction also reflect that message at least briefly at the beginning. Also, having large spin-induced polarization rotation of photons with spins in quantum dots is clearly the main point and achievement of the work. Yet it seems that the modifications made to the geometry to improve the homogeneous linewidth have resulted in sub-optimal output coupling of only 0.635 resulting in low device efficiencies and reducing the throughput of the system and the deterministic entanglement of the spins and photons in the future applications of such devices. The authors have added discussions about the post-selection step to improve the performance, which I think remains to be tested in future efforts for the presented configuration. I like to add that I do not see low output coupling in their case as necessarily a negative point, as experiments are always expected to improve. In the end, I would like to emphasize again that the responses and modifications have certainly helped with delivering the message of the manuscript to the reader in a clearer way.

Reviewer #3 (Remarks to the Author):

The authors have revised the manuscript to address the comments. I would recommend the publication of the manuscript.

Reviewer #1 (Remarks to the Author):

The conditional polarization rotation induced by a single quantum dot spin embedded in an electrically contacted micropillar is characterized. The electrical contacts stabilize the charge environment suppressing the fluctuations in the exciton optical transition energy by a factor of 9 compared to prior art [34-36], and is close to the limit due to fluctuations caused by nuclear spin environment. More importantly, the inhomogeneous broadening is reduced to be far below the homogeneous broadening resulting in a stable device. This is evidenced by a large rotation that can be observed in a time-ensemble average measurement, without the need for post-selection methods to mitigate against low frequency fluctuations. This is significant because a stable device is needed for scalability.

Hence I recommend publication.

However, I do so through gritted teeth. The first two versions of the manuscript were unpublishable due to title/claims unsupported by data. This was a total waste of everyones time. I get a sense that the abstract rides a bit too close to the line, and advice is not being listened to, and this is to the detriment of the reader. I note that the discussion crucial to understanding the significance of the work is on p7, 2nd col. It follows a long winded discussion of a measurement method the authors have already reported on in refs. [31,40]. The reader needs to perform a feat of endurance to reach the key take home messages. The paragraph is unclear on the difference in device design that leads to improved stability, and provides no quantitative information on the polarization rotations achieved in prior art to provide clear context for the resulting improvement in device performance.

We thank Reviewer 1 for his/her patience and useful advice. We started from different views regarding the main message and the notion of control, but we fully understand that some readers could share the same concerns as Reviewer #1. In this new version we further improved our manuscript (especially abstract and introduction, as advised), also in light of the comments by Reviewer #2.

First, we modified our abstract with the following sentence:

“[...] **A complete tomography approach is introduced to extrapolate** the output polarisation Stokes vector, conditioned by a specific spin state, **in presence of spin and charge fluctuations** [...]”

This provides a first indication (completed in the introduction) that, not only the spin but also the charge can fluctuate during the experiment, and that an extrapolation is required. We also specified more clearly the notion of control in our work, by rewriting the following sentence:

“[...] We find that an enhanced light-matter coupling, together with limited cavity birefringence and reduced spectral fluctuations, allow targeting most **conditional** rotations in the Poincaré sphere, **with a control both in longitude and latitude.** [...]”

Regarding the “*quantitative information on polarization rotations achieved in prior art*”, we also understand that this information was only partially addressed in the introduction, which we completed to:

“[...] A typical difficulty is that a measurement axis can be well adapted to measure some fidelity, with respect to an ideal target [35,36], yet at the same time prevent from distinguishing between depolarisation effects and actual polarisation rotations. In Refs [33,34], conversely, actual rotations could be demonstrated using a well-adapted measurement basis, yet only reaching rotation angles up to 6°.”

We also modified the following introductory paragraph, to clarify the main message and claims (also in response to Reviewer #2). We more directly mention the presence of detrimental spin and charge fluctuations, remove the word “control” from the paragraph, and insist on the fact that the rotations are conditional and extrapolated. To improve the fact that “*that the discussion crucial to understanding the significance of the work is on p7, 2nd col*”, and, correspondingly that “*the reader needs to perform a fit of endurance to reach the key take home messages*”, we also directly give the key technological ingredients allowing to reach giant rotations:

“Here, we report on ~~the observation and control~~ of giant polarisation rotations induced by a single QD-embedded electron spin, deterministically coupled to an electrically-contacted pillar cavity (see Methods). Compared to previous works [33-36], the strong reduction of spectral fluctuations provided by the electrical contacts, and the increased Purcell enhancement, are key ingredients allowing us to reach giant polarisation rotations. This includes highly-desired configurations such as $\pm\pi/2$ and π conditional rotations in the Poincaré sphere, though a degradation of the polarisation purity, down to around 70%, is observed at large angles.

We use polarisation tomography [40], this time applied to a charged quantum dot-microcavity device, to fully characterise the state of the reflected photons in the Poincaré sphere. The possibility to add or remove the electron from the quantum dot allows us to extrapolate the conditional Stokes vector \vec{S}_\uparrow conditioned to a charged QD in the spin state $|\uparrow\rangle$ even in presence of detrimental spin and charge fluctuations. We finally show that, by a proper set of detunings, most orientations of the conditional output Stokes vector \vec{S}_\uparrow can be reached. [...]”

Finally, regarding “*the difference in device design that leads to improved stability*”, we now better refer to it at the corresponding paragraph (page 7, col 2):

“[...] In our work, the inhomogeneous broadening remains limited since $\sigma_{SF} = 0.5 \mu\text{eV}$, thanks to the efficient evacuation of fluctuating carriers through the electrical contacts (see Methods and Refs [37,38] for details on the device structure). [...]”

Reviewer #2 (Remarks to the Author):

The authors have put a great effort into revising the manuscript for the second time, including changing the title of the manuscript, which should have been done, as emphasized by reviewers during the previous round of the review. The manuscript and supplementary material are in a much better shape from different aspects, including the added clarification and explanations to multiple sections of both the main text and the supplementary material. Also, adding the GIF files certainly improved the quality of the data presentation and visualization of the data by the readers. The extra appendix section that the authors added to the end of their response also was helpful in presenting a comparison with previous works done on the field and emphasizing their new contributions, especially regarding the

analysis of the rotations on the full Poincare sphere. The authors also clarified the circumstances regarding spin initialization and the technical difficulties and their future efforts toward reducing co-tunneling effects and possible spin initializations for quantum spin-photon interface applications. I want to emphasize again the point that was also pointed out by the first reviewer during the previous round of review regarding mentioning stationary qubits in the text. From the authors' response to the review report, it is clear that they understand and agree that there is no stationary matter or spin qubit in their experiment as there is a fast co-tunneling process happening without any spin initialization in between tunneling events.

We thank Reviewer #2 for his/her positive assessment and all his/her previous comments, which helped us providing crucial revisions.

However, the abstract and introduction still, to some extent, advertise the immediate applicability of their result for spin-photon entanglement and having matter qubits interacting with photons. The fact that this is not an immediate result of this work becomes clear after reading the rest of the manuscript, and some parts of the supplementary material. I think it would have been useful to have the abstract and introduction also reflect that message at least briefly at the beginning.

We fully understand and we have accordingly revised our abstract and introduction: please see our response to Reviewer #1, where the corresponding changes have been detailed.

Also, having large spin-induced polarization rotation of photons with spins in quantum dots is clearly the main point and achievement of the work. Yet it seems that the modifications made to the geometry to improve the homogeneous linewidth have resulted in sub-optimal output coupling of only 0.635 resulting in low device efficiencies and reducing the throughput of the system and the deterministic entanglement of the spins and photons in the future applications of such devices. The authors have added discussions about the post-selection step to improve the performance, which I think remains to be tested in future efforts for the presented configuration. I like to add that I do not see low output coupling in their case as necessarily a negative point, as experiments are always expected to improve.

This is indeed an important point and we fully agree with Reviewer #2. We introduce this important discussion at the end of the corresponding paragraph :

“[...] We note that improved values of $\eta_{\text{top}}=0.9$ have already been obtained in similar structures [39]: this paves the way towards enhanced efficiencies, and potentially towards nearly-deterministic entanglement of incoming photons with a single spin.”

In the end, I would like to emphasize again that the responses and modifications have certainly helped with delivering the message of the manuscript to the reader in a clearer way.

Reviewer #3 (Remarks to the Author):

The authors have revised the manuscript to address the comments. I would recommend the publication of the manuscript.

We are very thankful to Reviewer #3 whose precious comments and questions allowed us to reach this point, and provide an improved manuscript.